# An engram of intentionally forgotten information

Sanne Ten Oever [1,2,3 ✉], Alexander T. Sack [3,4], Carina R. Oehrn [5,6] & Nikolai Axmacher [7,8 ✉]

Successful forgetting of unwanted memories is crucial for goal-directed behavior and mental wellbeing. While memory retention strengthens memory traces, it is unclear what happens to memory traces of events that are actively forgotten. Using intracranial EEG recordings from lateral temporal cortex, we find that memory traces for actively forgotten information are partially preserved and exhibit unique neural signatures. Memory traces of successfully remembered items show stronger encoding-retrieval similarity in gamma frequency patterns. By contrast, encoding-retrieval similarity of item-specific memory traces of actively forgotten items depend on activity at alpha/beta frequencies commonly associated with functional inhibition. Additional analyses revealed selective modification of item-specific patterns of connectivity and top-down information flow from dorsolateral prefrontal cortex to lateral temporal cortex in memory traces of intentionally forgotten items. These results suggest that intentional forgetting relies more on inhibitory top-down connections than intentional remembering, resulting in inhibitory memory traces with unique neural signatures and representational formats.

[1] Max Planck Institute for Psycholinguistics, Wundtlaan 1, 6525XD Nijmegen, The Netherlands. [2] Donders Centre for Cognitive Neuroimaging, Radboud University, Kapittelweg 29, 6525EN Nijmegen, The Netherlands. [3] Department of Cognitive Neuroscience, Faculty of Psychology and Neuroscience, Maastricht University, Oxfordlaan 55, 6229EV Maastricht, The Netherlands. [4] Department of Psychiatry and Neuropsychology, School for Mental Health and Neuroscience (MHeNs), Brain and Nerve Centre, Maastricht University Medical Centre+ (MUMC+), Debyelaan 25, 6229HX Maastricht, The Netherlands. [5] Department of Neurology, Philipps-University of Marburg, Biegenstraße 10, 35037 Marburg, Germany. [6] Center for Mind, Brain and Behavior (CMBB), Philipps-University Marburg, Biegenstraße 10, 35037 Marburg, Germany. [7] Department of Neuropsychology, Faculty of Psychology, Institute of Cognitive Neuroscience, Ruhr University Bochum, Universitätsstraße 150, 44801 Bochum, Germany. [8] State Key Laboratory of Cognitive Neuroscience and Learning and IDG/McGovern Institute for Brain Research, Beijing Normal University, 19 Xinjiekou Outer St, Beijing 100875, China. ✉email: sanne.tenoever@mpi.nl; nikolai.axmacher@ruhr-uni-bochum.de

Any sensory input triggers a cascade of neural responses that span across various brain regions and engages numerous excitatory and inhibitory circuits[1,2]. These responses can be considered the neural fingerprint of an item, its stimulus-specific neural representation. Stimulus-specific representations are uniquely shaped by the specific circumstances in which a stimulus is encoded into memory, such that some neuronal connections are strengthened and others are inhibited (Fig. 1)[3,4]. These modifications give rise to item- and context-specific memory traces, which carry both information about the identity of a stimulus and about the transformations that occurred during learning. Reactivating these memory traces thus allows one to remember specific events together with their learning context, the hallmark of episodic memory. Indeed, several studies showed that episodic memory retrieval depends on the reoccurrence of the item- and/or context-specific neural representations, a process that relies on close coordination between the hippocampus and the adjacent lateral temporal cortex (LTC)[5–9]. This effect is typically more pronounced for remembered compared to forgotten information[10,11] (Fig. 1A) and has been quantified via encoding-retrieval similarity (ERS). The increased ERS that was found for remembered events could come about due to processing along selectively modified sensory-semantic connections or due to the activation of the memory context due to recognition-related processes. Note that while in some studies the terms reactivation or reinstatement refer specifically to recognition-related reactivation, we here are neutral about the process that causes the ERS effects (sensory-semantic processing or active recognition memory). In contrast to remembering, incidental forgetting could result from a failure to modify connections during encoding, i.e., a lack of learning-related plasticity (Fig. 1D).

This passive process of forgetting may differ from the active mechanisms in play when one intentionally tries to forget information. In fact, although most studies on memory traces focus on remembering, humans also have the ability to intentionally forget information[12,13]. Indeed, when individual items are directly followed by instructions to forget them, they are less likely to be retrieved afterward than items that are followed by remembering instructions. Successful intentional forgetting instructions are accompanied by increased

activation of the dorsolateral prefrontal cortex (DLPFC) and reduced activity in the hippocampus[14,15]. DLPFC activation putatively reflects the recruitment of inhibitory control processes that exert a top–down influence on memory functions[16]. Note that these inhibitory processes concern functional inhibition, which likely comprises inhibitory as well as excitatory synaptic connections[17,18]. Indeed, intentional forgetting is accompanied by increases in alpha/beta (8–25 Hz) oscillations—a signature of active functional inhibition[19,20]—in both DLPFC[21] and hippocampus[22]. Thus, intentional forgetting seems to be an active and effortful inhibitory process rather than a mere consequence of reduced rehearsal[16,21,23].

It is unknown what happens to the memory trace of intentionally forgotten items. On one hand, one may hypothesize that active forgetting of newly encoded stimuli erases neuronal connections that support the memory traces of these items. In this scenario, successfully forgotten TBF items should show lower ERS as compared with (incidentally) remembered TBF items—similar to what is seen for forgotten vs. remembered TBR items (Fig. 1B). Alternatively, active forgetting may selectively modify functional inhibition patterns that are created during encoding as well. In this case, ERS effects would not be generally reduced but may rely specifically on these inhibition patterns (Fig. 1C). Conceptually, selective modification of inhibition in memory representations resembles the inhibitory memory traces that are built during extinction learning[24–26].

As described above, this inhibitory signature likely depends on alpha/beta oscillations[20,27] and may reflect inhibitory feedback from DLPFC to areas representing the memory trace[5–9]. Although the hippocampus has been proposed to act as a pointer or index towards stimulus-specific representations[28], these representations seem to rely on neocortical areas. For highly semantic material such as the words used in our study, the LTC is a key candidate for such areas, as it is among the most relevant brain regions for semantic representations[29,30]. In fact, previous studies have shown strong item-specific ERS effects in the LTC for correctly remembered information[8,31].

Here, we used intracranial EEG (iEEG) recordings from the DLPFC and LTC of epilepsy patients to investigate the item-specific ERS of words that were either cued to be forgotten or to be remembered. Our results demonstrate that successful intentional forgetting is owing to a selective modification of item-specific top–down connections, rather than a mere degradation of the memory traces.

## Results

The experiment consisted of an encoding and a retrieval phase in which written words were presented on a screen (Fig. 2A). In the encoding phase, each word was followed by a cue that instructed participants to either remember the word (to-be-remembered words: TBR) or to forget it (to-be-forgotten words: TBF). During the retrieval phase, participants were presented with all the words from the encoding phase (TBR and TBF) and an equal number of new words. Participants classified these words via button presses as either "old" (presented during the encoding phase) or "new" (not presented before). In the main analysis, we included 16 presurgical epilepsy patients (8 females; age: 41.3 ± 15.0, mean ± standard deviation) who were implanted with iEEG electrodes in LTC (Fig. 2C; red dots), the most relevant brain area for semantic representations[29,30]. Ten of these patients were implanted with electrodes in the DLPFC (Fig. 2C, blue dots). Representations in other neocortical areas and in the hippocampus were analyzed in post hoc analyses as well (see Supplementary Fig. 1 for all channel locations).

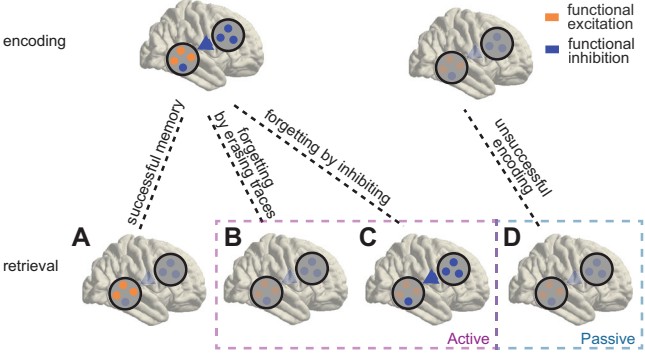

**Fig. 1 Reading memory traces via encoding-retrieval similarity.** During encoding, stimulus-specific patterns of functional inhibition and activation are modified in a context-dependent manner. These patterns reoccur during successful memory retrieval, resulting in high encoding-retrieval similarity (ERS; **A**). Active forgetting may result from either erasing the traces of previously modified connections (**B**; low ERS) or from selective modification of patterns of functional inhibition (**C**; high ERS, but for inhibition patterns). If encoding is not successful (i.e., memory traces are not modified) ERS values will be low (**D**).

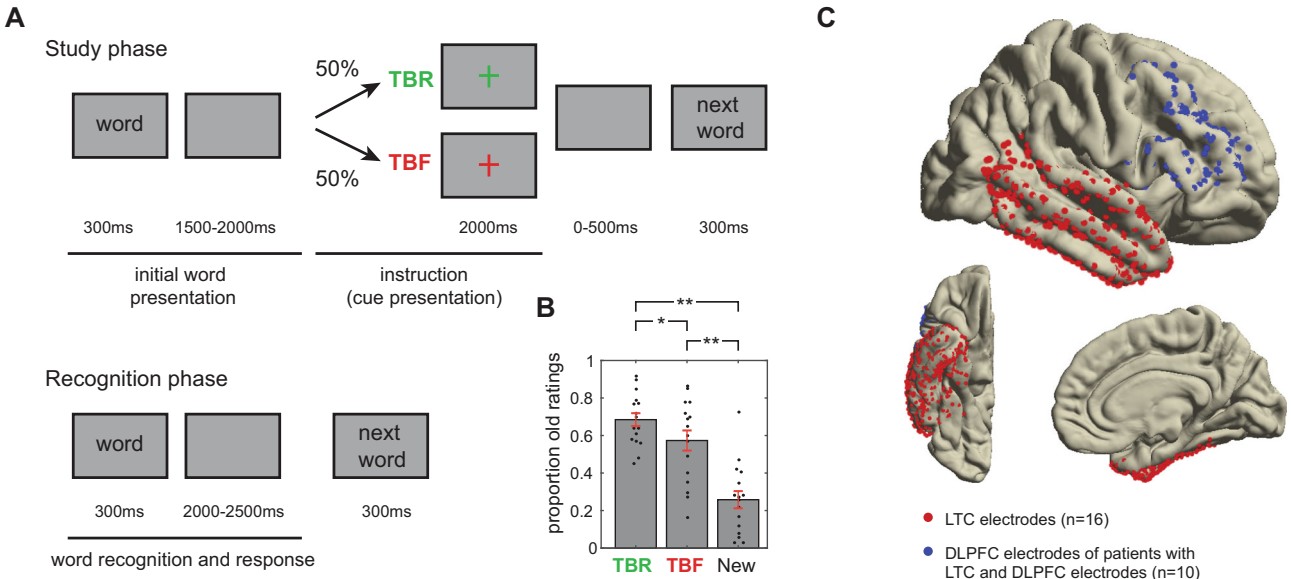

**Fig. 2 Experimental design and behavioral results. A** Experimental design: item-method-directed forgetting task. TBR: to-be-remembered; TBF: to-be-forgotten. **B** Behavioral results in patients with intracranial EEG electrodes in the lateral temporal cortex (LTC; $n = 16$; two-sided paired $t$-tests). Error bars indicate the standard error of the mean. Single and double asterisk(s) indicate significance at the 0.05 and 0.01 level, respectively (TBR-TBF: $p = 0.012$; for TBR and TBF versus new $p < 0.001$). **C** LTC and dorsolateral prefrontal cortex (DLPFC) electrodes. Red colors include the 16 patients with LTC electrodes. Blue electrodes indicate the DLPFC electrodes of the 10 patients with LTC and DLPFC electrodes. Source data are provided as a Source Data file.

**Behavioral results**. As expected, we found that the proportion of remembered words was higher among TBR than TBF items (TBR: $68.5 \pm 14.2\%$; TBF: $57.3 \pm 22.2\%$; $t(15) = 2.875$, $p = 0.012$; Fig. 2B). Nevertheless, the false alarm rates were lower than the rates of correctly memorized TBF items in all participants (i.e., TBF hits > false alarms new items; $t(15) = 8.22$, $p < 0.001$; "old" responses to new words: $25.8\% \pm 18.9\%$). Similarly, the d-prime for TBF words was significantly lower than for TBR words ($t(15) = 3.00$, $p = 0.009$; Supplementary Fig. 2).

We then performed a repeated-measures ANOVA with reaction times as a dependent variable and instruction and memory as factors. This analysis showed faster reaction times for the TBR compared with the TBF condition ($F(1,15) = 5.212$, $p = 0.037$) and faster reaction times for remembered compared with forgotten words ($F(1,15) = 8.362$, $p = 0.012$; see Supplementary Fig. 2). The interaction between instruction and memory showed a trend ($F(1,15) = 3.48$, $p = 0.082$), pointing towards slightly more pronounced reaction time differences for TBR than TBF items.

**Item-specific alpha/beta ERS effects for TBF words**. In order to investigate item-specific memory traces of TBR and TBF items, we analyzed ERS[8,11,31]. Item-specific neural representations consisted of distributed time-frequency patterns across all LTC channels during word presentation ($27.3 \pm 11.1$ channels; range, 7–45; total, 437 channels). We extracted power values between 650 ms before stimulus onset to 1.5 s post-stimulus, using Morlet wavelets for frequencies of 2–30 Hz (2–6 cycles, linearly increasing) and multi-tapers for frequencies of 30–150 Hz (dpss taper, 10 cycles, 0.5 cycles smoothing). Log-transformed power values were baseline corrected ($-0.3$ s to $-0.1$ s baseline window). ERS was calculated via Spearman's correlations across LTC channels and across time for each encoding-retrieval word pair, separately for each frequency bin (vectorizing channel x time data in a time window of 0.1–0.5 s, a total of nine non-overlapping time bins; after artifact correction $89.0 \pm 21.1$ and $89.4 \pm 20.7$ items had trials with no artifacts in the encoding and retrieval period for the TBR and TBF condition, respectively). Thus, ERS

values reflect correlations of frequency-specific power values in every encoding and retrieval trial, which are independent of the overall activity levels in these trials. For example, there can be pronounced correlations at relatively low overall (channel-averaged) activity levels; or low, negative, or absent correlations at high overall activity levels.

We first aimed to identify item-specific ERS, separately for TBR and TBF words. We thus contrasted averaged ERS values between all matching encoding-retrieval word pairs (within-item ERS) versus averaged ERS values between all non-matching word pairs (between-item ERS; Fig. 3A: dark orange vs. light orange). We corrected for multiple comparisons using cluster statistics with label-shuffled surrogates[32], separately for the low and high frequencies.

For TBR items, we only found a trend for significant item-specific ERS (i.e., higher within-item than between-item ERS) for the high frequencies (105–110 Hz; Fig. 3B, cluster statistics = 5.05, $p = 0.072$; max $t(15)$ value = 2.69). No cluster was found for the low frequencies. For TBF items, we observed the opposite pattern of results, with item-specific ERS only in the low-frequency analysis (16–21 Hz; cluster statistics = 19.61, $p = 0.008$; max $t(15)$ value = 3.96), but not in the high-frequency analysis (no cluster found). Directly contrasting item-specific (i.e., within-item vs. between-item) TBR with item-specific TBF indicated more pronounced low-frequency ERS for TBF vs. TBR items in the beta-frequency range (18–22 Hz; cluster statistics = $-13.43$, $p = 0.026$; min $t(15)$ value = $-3.33$). Controlling for the difference in the amount of remembered versus forgotten items between TBR and TBF items (by calculating first the ERS over forgotten and remembered items and then averaging) showed a similar overall pattern (Supplementary Fig. 3A). A direct comparison between the low-frequency effect for the TBF and the high-frequency trend for the TBR items showed an interaction between frequency and condition ($F(1,15) = 11.15$; $p = 0.004$).

In a post hoc analysis we also analyzed the same contrast in different brain regions (parietal cortex, DLPFC, and hippocampus; Supplementary Fig. 4). We found in the hippocampus a

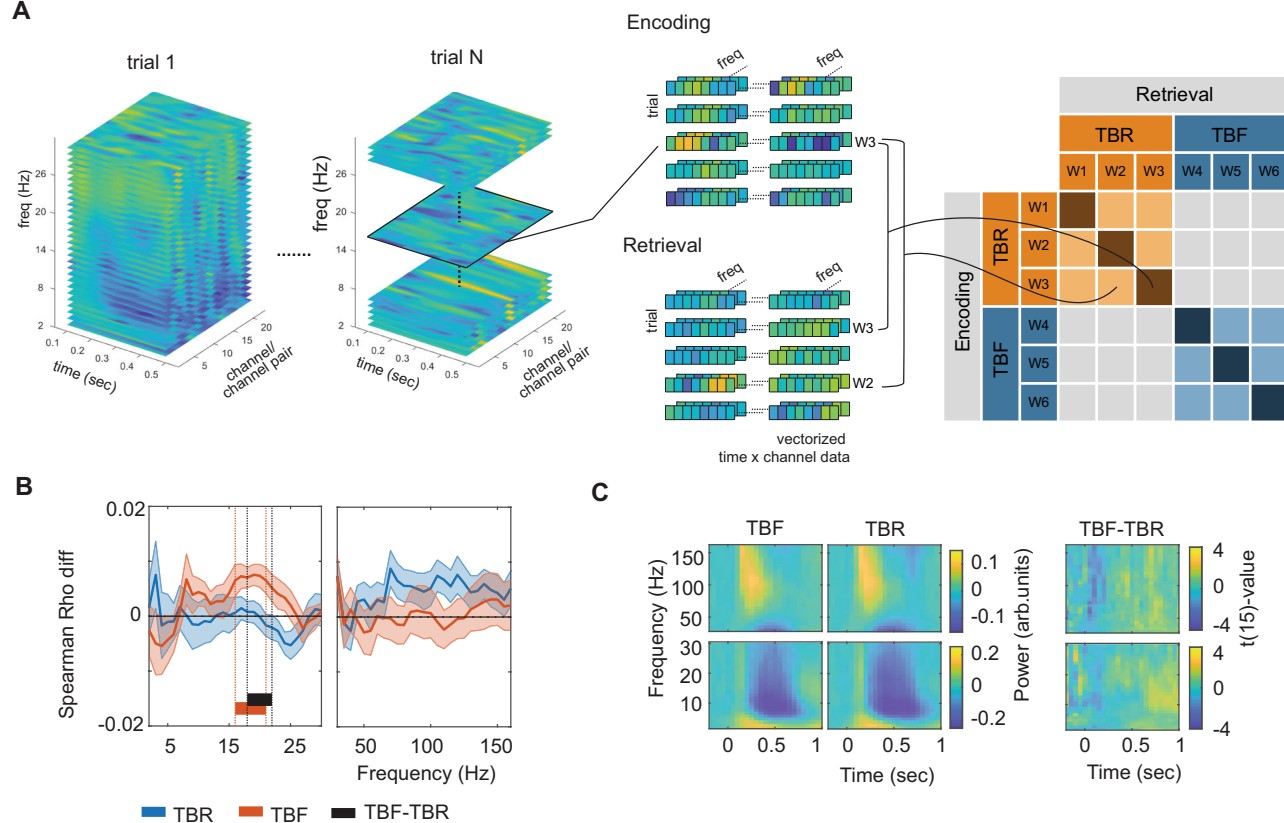

**Fig. 3 Voluntary forgetting modifies but does not remove item-specific memory traces. A** Encoding-retrieval similarity analysis pipeline. Time-frequency patterns were extracted from all LTC electrodes, and matrices of time points x channels (or channel pairs for coherence analyses) were built, separately for each frequency bin. Encoding-retrieval similarities were compared between same (diagonal) and different (off-diagonal) words (W) at encoding and retrieval in order to identify item-specific activations. We computed Spearman's correlations for to-be-remembered (TBR) and to-be-forgotten (TBF) words separately, and also separately for items that were actually remembered and forgotten (see Fig. 4). **B** Encoding-retrieval similarity of distributed activity patterns across lateral temporal cortex channels based on power for to-be-remembered (TBR) and to-be-forgotten (TBF) words, irrespective of subsequent memory. Shaded error bars indicate the standard error of the mean ($n = 16$). Blue, orange, and black lines indicate significant clusters for item-specific activation of TBR items, TBF items, and the difference between TBR versus TBF items, respectively. **C** Time-frequency responses for TBR and TBF items during retrieval. No significant differences were found. Source data are provided as a Source Data file.

significant item-specific ERS for TBF (135–160 Hz and 105–125 Hz; cluster statistics = 23.06 and 14.78; $p = 0.01$ and 0.023; max $t(15)$ value = 4.50), but not TBR items for high frequencies. The difference between TBF and TBR was also significant (145–160 Hz; cluster statistics = −10.79; $p = 0.03$; max $t(15)$ value = −3.18). However, the hippocampal effect was not memory-specific (see next section).

We additionally investigated ERS effects in LTC when encoding and retrieval times were shifted across time, considering periods between −0.2 sec and +0.8 sec (averaged over the frequencies that were found in the corresponding contrasts in the non-shifted analyses). Results are shown in Supplementary Fig. 3B.

These results demonstrate item-specific ERS effects of TBF items in the low-frequency range. In order to exclude that the differential ERS effects between TBR and TBF items were owing to overall power differences between conditions, we compared the time-frequency distributions of power values between TBR and TBF items during retrieval. We averaged across trials and electrodes in each subject and focused on the same time window as in the ERS analyses. We found no evidence for a difference (Fig. 3C; largest cluster low frequencies: cluster statistics = 50.26; $p = 0.176$; largest cluster high frequencies: cluster statistics = 35.02; $p = 0.232$). This strongly suggests that differential item-specific ERS effects of TBF and TBR items are not owing to overall power differences in the LTC.

**Functional relevance of item-specific ERS effects for memory.** We next investigated whether ERS differed between words that were actually remembered and those that were forgotten, separately for the TBR and TBF condition. For TBR words, item-specific ERS (within-item vs. between-item ERS) was only found for remembered words (Fig. 4A; 125–160 Hz and 70–80 Hz; cluster statistics = 19.62 and 8.06; $p = 0.006$ and $p = 0.04$; max $t(15)$ value = 2.70 and 2.99), but not for forgotten words (no cluster found). The difference between remembered and forgotten TBR words was significant (140–155 Hz; cluster statistics = 9.68, $p = 0.036$; max $t(15)$ value = 3.12). Importantly, this effect was not due to the larger number of remembered vs. forgotten TBR trials: when we randomly selected as many remembered trials as there were forgotten trials in each patient (repeated for 50 times and averaged), we observed a very similar result (cluster statistics for remembered words = 14.31, $p = 0.015$, 140–160 Hz). No effects were found for the low frequencies. Again, memory effects could not be explained by overall power differences, which were only found in a later time window (0.65–1.5 s) and at low frequencies (2–21 Hz; Fig. 4C; cluster statistics = −467.96, $p = 0.004$; min $t(15)$ value = −4.74). As this time window was not included in the ERS analysis, it could not have influenced the result. Moreover, all our contrasts consist of comparisons between within-item vs. between-item ERS values, and thus any overall effects (i.e., effects that are not item-specific) are accounted for.

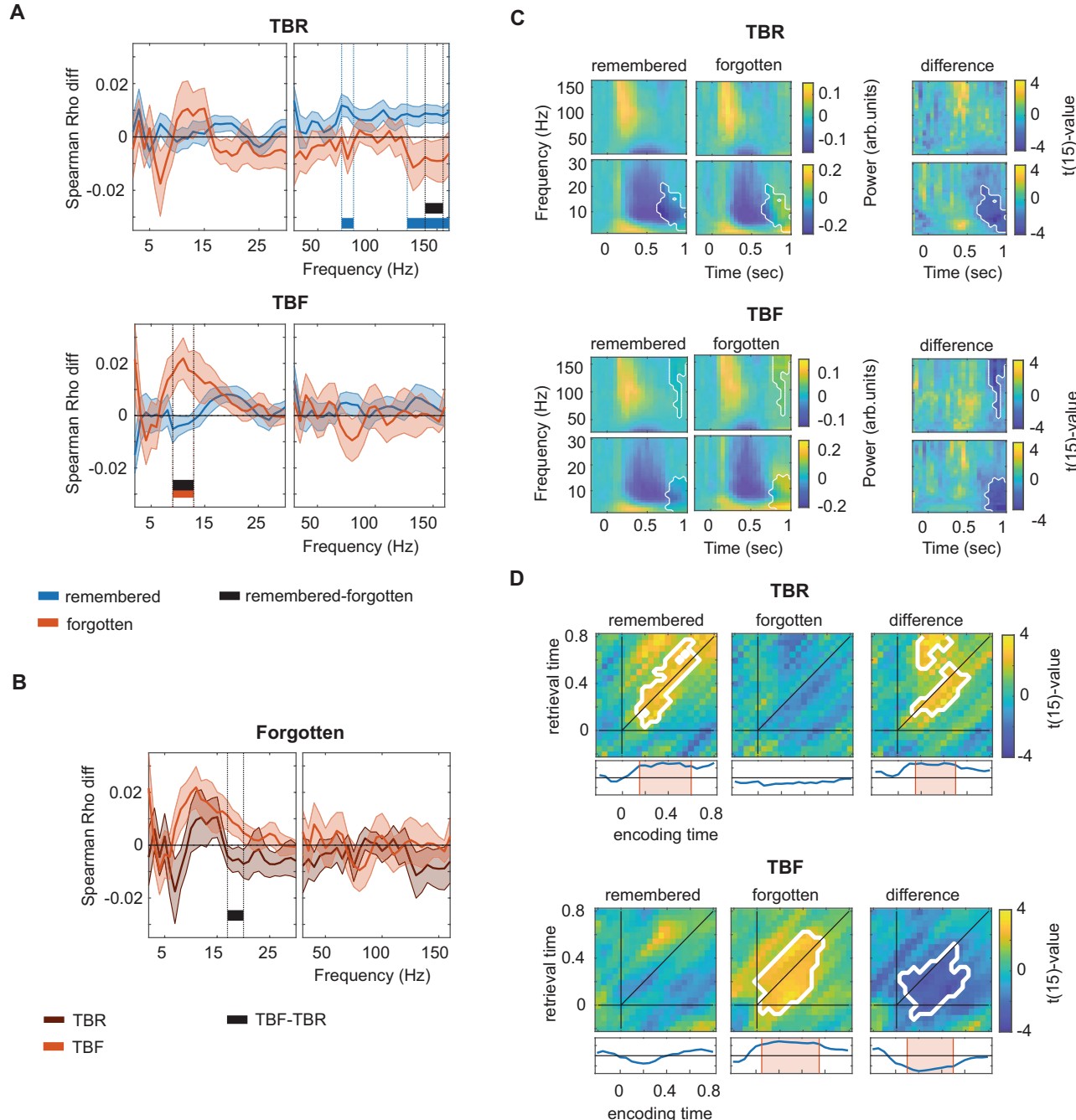

**Fig. 4 Differential encoding-retrieval similarity of remembered and forgotten items. A** Encoding-retrieval similarity for all to-be-remembered (TBR; top) and to-be-forgotten (TBF; bottom) items, separately for words that were later remembered (blue) or forgotten (orange) based on LTC power ($n = 16$). Blue, orange, and black lines indicate significant item-specific encoding-retrieval similarity of remembered items, forgotten items, and their difference, respectively. Shaded error bars indicate the standard error of the mean. The direct contrast between intentionally vs. incidentally forgotten items (i.e., TBR forgotten vs. TBF forgotten) is separately displayed (**B**). **C** Time-frequency responses during retrieval (averaged across all lateral temporal channels) for remembered and forgotten TBR (top) and TBF (bottom) words. Alpha/beta power was significantly higher for forgotten words in a late window (white outline). **D** Time-shifted encoding-retrieval similarity analyses showed similar patterns as the main analyses. The white outline indicates significance (cluster-corrected for multiple comparisons). Insets at the bottom represent the time windows at which effects are significant when encoding-retrieval similarity is based on the same encoding and retrieval time. Source data are provided as a Source Data file.

For TBF words, item-specific ERS effects of low-frequency patterns were specific for successfully forgotten items: Significant ERS was observed in the low-frequency analysis for forgotten, but not for remembered words (9–13 Hz; Fig. 4A; cluster statistics = 12.30, $p = 0.031$; max $t(15)$ value = 2.94). The difference between forgotten and remembered words was significant for the low-frequency analysis (9–13 Hz; cluster statistics = −12.89,

$p = 0.032$; min $t(15)$ value = −3.03). In addition, the ERS for the TBF forgotten words was higher than for the TBR forgotten words in the beta range (17–20 Hz; cluster statistics = −9.58; $p = 0.038$; max $t(15)$ value = −2.66). Again, results could not be explained by overall power, which differed only in a later time period (Fig. 4C; 0.75–1.45 s; 2–15 Hz; cluster statistics = −435.38, $p = 0.005$; min $t(15)$ value = −5.99 and 0.75–1.45 s; 75–160 Hz;

cluster statistics = −260.34, $p = 0.016$; min $t(15)$ value = −5.00; see Supplementary Fig. 5 for raw correlation traces).

The time-shifting ERS analyses averaged over the frequencies of the significant clusters showed similar patterns of increased ERS for TBR remembered vs. forgotten words (Fig. 4D; TBR main effect: cluster statistics = 145.73; $p = 0.014$; max $t(15)$ value = 3.31; remembered-forgotten difference: cluster statistics = 89.96, $p = 0.015$; max $t(15)$ value = 3.93) and for TBF forgotten vs. remembered words (Fig. 4D; cluster statistics = 248.47; $p = 0.019$; max $t(15)$ value = 3.68; remembered-forgotten difference: cluster statistics = −194.26, $p = 0.015$; min $t(15)$ value = −3.68). Note that the statistics of this analysis are likely inflated (for the on-diagonal clusters) as we pre-selected the frequency range based on the on-diagonal analyses.

Our data presented thus far show that successfully remembering information relies on ERS in the gamma range. They also demonstrate that intentionally forgotten information shows ERS effects as well, that they depend on item-specific distributions of alpha/beta power values, and that they are functionally relevant for successful forgetting. Interestingly, alpha/beta activity patterns during the presentation of an item that was later successfully forgotten were even more similar to patterns during a novel item than a different successfully forgotten TBF item at retrieval (Supplementary Fig. 6 and Supplementary Fig. 7 for cue-retrieval similarity).

We go on to investigate whether alpha/beta ERS effects of TBF items are related to top–down connections between DLPFC and LTC.

**Item-specific ERS effects between DLPFC and LTC**. The DLPFC is crucially involved in the voluntary control of forgetting[22,33–35] and exerts an inhibitory top–down influence over hippocampal memory functions[14,15,22]. DLPFC is also strongly connected to LTC to support, for example, language functionality[36] and memory[37,38]. However, it is still unknown whether the DLPFC merely suppresses the formation of stimulus-specific memory traces, or whether these interactions are incorporated into the memory traces. We thus investigated whether the patterns of DLPFC-LTC interactions that occur during encoding of individual items can also be found during retrieval of these items, and whether this effect is specific to TBF trials. We calculated ERS using frequency-specific coherence between all pairs of LTC and DLPFC channels as features (including AAL atlas values 7,11,13 [left] and 8,12,14 [right]; 0.1–0.5 sec time window). Notably, these analyses were based on bipolar derivatives of the original electrodes to avoid any confounds due to volume conduction (see Methods).

For later forgotten TBF items, we indeed found that DLPFC-LTC alpha/beta-frequency (14–16 Hz) coherence patterns during the initial presentation of specific items were more strongly correlated with the patterns during the later presentation of those same items as compared with other items (i.e., indicating item-specific ERS of connectivity patterns; Fig. 5A; cluster statistics = 7.94, $p = 0.046$; max $t(9)$ value = 2.98). This effect was not observed for TBF words that were later remembered (no positive cluster found). A direct comparison between successfully forgotten and incidentally remembered TBF words revealed more pronounced coherence-based ERS for forgotten words (14–16 Hz, cluster statistics = −9.06, $p = 0.034$; min $t(9)$ value = −3.67). These results demonstrate functionally relevant item-specific ERS of alpha/beta-frequency connectivity patterns between DLPFC and LTC during the presentation of successfully forgotten words.

A similar effect did not occur for TBR items, where we did not find any item-specific ERS of DLPFC-LTC patterns for either later remembered or forgotten TBR items (only one non-significant positive low-frequency cluster for the forgotten TBR items, $p = 0.334$). Directly comparing remembered and forgotten words showed significant effects in a small frequency interval for both the TBR and TBF condition (Fig. 5A; TBR: 145–150 Hz; cluster statistics = −6.14, $p = 0.009$; min $t(9)$ value = −3.71; TBF: 155−160 Hx; cluster statistics = −5.28, $p = 0.023$; min $t(9)$ value = −2.82).

Control analyses showed that the differential ERS effects of DLPFC-LTC connectivity patterns of forgotten vs. remembered TBF items could not be explained by overall differences in the strength of DLPFC-LTC coherence (Fig. 5B; no significant difference) or in the power at DLPFC channels (Fig. 5C; low-frequency power between 2 and 19 Hz increased for forgotten TBF items only in a later time window between 0.59 and 1.5 sec).

Previous studies suggested that intentional forgetting is related to a suppression of hippocampal memory functions (e.-g.,[15,22,39,40]). We thus investigated ERS effects in the hippocampus ($n = 13$ patients; Supplementary Fig. 4) and in hippocampal-LTC connectivities ($n = 13$ patients; Supplementary Fig. 8). However, although hippocampal ERS was higher for TBF than TBR items, it did not depend on memory, and hippocampal-LTC connectivities did not show any condition differences.

**Item-specific ERS of top–down control**. Next, we investigated whether ERS effects could not only be found in patterns of DLPFC-LTC coherence but also in patterns of directed (i.e., causal) influences between DLPFC and LTC. In order to assess directional ERS, we extracted trial-by-trial transfer entropy (TE) values. We first focused on TBF items and calculated TE on the band-passed filtered absolute Hilbert transform ±2 Hz around the effect found in the coherence analysis (Fig. 5A), i.e., filtered from 12 to 18 Hz (filtering from 14 to 16 Hz resulted in similar effects). For each channel pair separately (same channel pairs as for the coherence-based ERS), we considered variable interregional delays in a time range reported before (50–150 ms[22]). Overall TE was estimated as the average TE across these delays[41]. Indeed, for later forgotten TBF items, patterns of DLPFC→LTC connectivity (i.e., of top–down influences) during encoding were more similar to patterns during later presentation of the same items than of different items, indicating item-specific ERS of top–down control patterns ($t(9) = 4.094$, $p = 0.004$; Fig. 6A). No item-specific TE ERS effects were found for remembered TBF items ($t(9) = −0.838$, $p = 0.423$). A direct comparison revealed that item-specific ERS was significantly more pronounced for forgotten than remembered TBF items ($t(9) = 3.810$, $p = 0.005$). For the TBR items, no significant effects were found (Supplementary Fig. 9). Control analyses showed that average TE values did not differ between remembered and forgotten TBF words (Fig. 6C).

**Region- and hemisphere-specific analyses**. To further scrutinize the spatial location of ERS effects, we divided the LTC into five separate ROIs (superior, middle, and inferior temporal gyrus, fusiform gyrus, and temporal pole) and re-calculated ERS with the subset of electrodes in each of these areas in the frequency ranges that were part of the significant clusters in the respective analyses. For the TBR effects, none of the regions showed significant ERS after Bonferroni correction for multiple comparisons ($p > 0.05$). By contrast, TBF ERS effects were consistently found in middle temporal gyrus (ERS based on power and coherence: both $p$(corrected) < 0.05; Supplementary Fig. 10).

We then repeated the same analyses separately for the left and right hemisphere. For the TBF effects, there was no lateralization found ($p > 0.05$). For the TBR effects, we found a trend for lateralization such that for remembered TBR items, the left

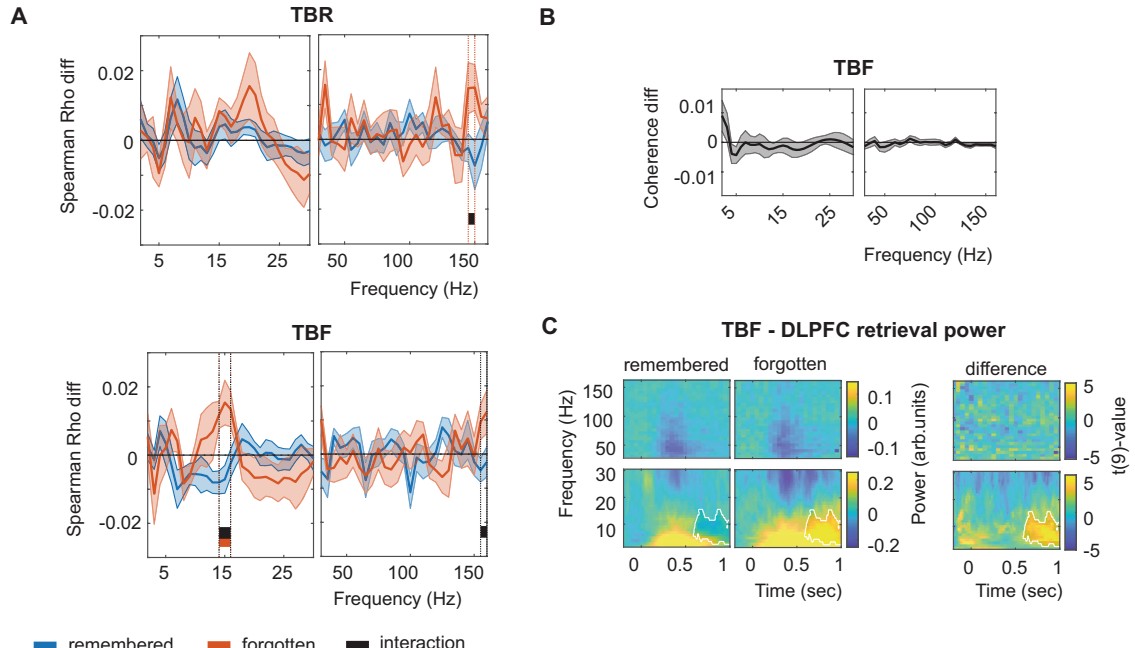

**Fig. 5 Item-specific encoding-retrieval similarity of connectivity patterns between DLPFC and LTC. A** Encoding-retrieval similarity for all to-be-remembered (TBR; top) and to-be-forgotten (TBF; bottom) items, separately for words that were later remembered (blue) or forgotten (orange) based on dorsolateral prefrontal cortex (DLPFC)—lateral temporal cortex (LTC) coherence (n = 10). Shaded error bars indicate the standard error of the mean. Blue, orange, and black lines indicate significant item-specific encoding-retrieval similarity of remembered words, forgotten words, and their difference, respectively. **B** Overall DLPFC-LTC coherence difference between remembered and forgotten words. **C** Time-frequency responses during retrieval for DLPFC averaged across all frontal bipolar channels for TBF words. Power was significantly different in a late window as indicated by the white outline. Source data are provided as a Source Data file.

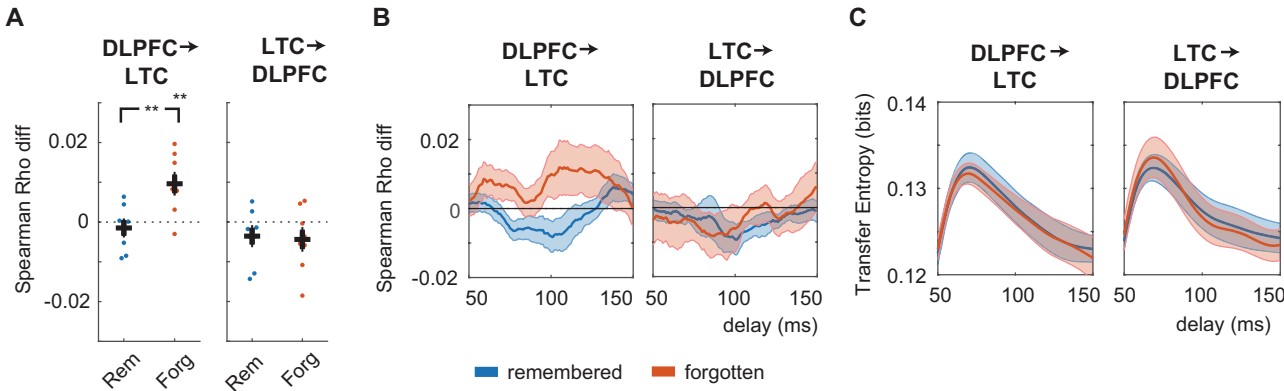

**Fig. 6 Encoding-retrieval similarity of item-specific patterns of top–down control. A** Item-specific encoding-retrieval similarity of remembered (blue) and forgotten (red) to-be-forgotten (TBF) items based on transfer entropy. Left: top–down control (dorsolateral prefrontal cortex (DLPFC)→ lateral temporal cortex (LTC)), right: bottom–up connectivity (LTC→DLPFC). Double asterisks indicate significance at the 0.01 level (n = 10; two-sided paired t test [p = 0.005] and two-sided one-sample t test [p = 0.004]). **B** Item-specific encoding-retrieval similarity of top–down control patterns for various interregional delays. Shaded error bars indicate the standard error of the mean. **C** Overall transfer entropy values during retrieval. Error bars indicate the standard error of the mean. Source data are provided as a Source Data file.

hemisphere showed a trend for stronger ERS than the right hemisphere (Supplementary Fig. 11; independent *t* test: *t*(23) = 1.98, *p* = 0.059). Restricting the analyses to only the participants with both left and right electrodes did indicate a significant hemisphere effect (paired *t* test: *t*(8) = 4.79, *p* = 0.001).

**Cue-related effects in DLPFC and LTC.** Finally, we evaluated time-frequency patterns in DLPFC and LTC during the time period of the cue presentation in order to investigate their possible relationship to ERS effects. In the LTC, we found a main

effect of instruction (6–30 Hz; 0.35–0.85 sec; *p* = 0.013; max *t*(16) value = 4.636; Fig. 7A) with a cluster showing relatively higher alpha/beta power after a TBF cue compared with a TBR cue (less pronounced reductions as compared with baseline, i.e., reduced desynchronization) as well as an interaction effect (2–11 Hz; 0.75–1.5 sec; *p* = 0.009; max *t*(16) value = 3.89). This interaction was a consequence of stronger theta/alpha activity in the remembered compared with forgotten condition for TBF items (2–13 Hz; 0.8–1.5 sec; *p* = 0.004; max *t*(16) value = 5.35; Fig. 7B), but not for TBR items (*p* > 0.05; Fig. 7C). In the DLPFC, we observed larger DLPFC gamma power increases for forgotten as

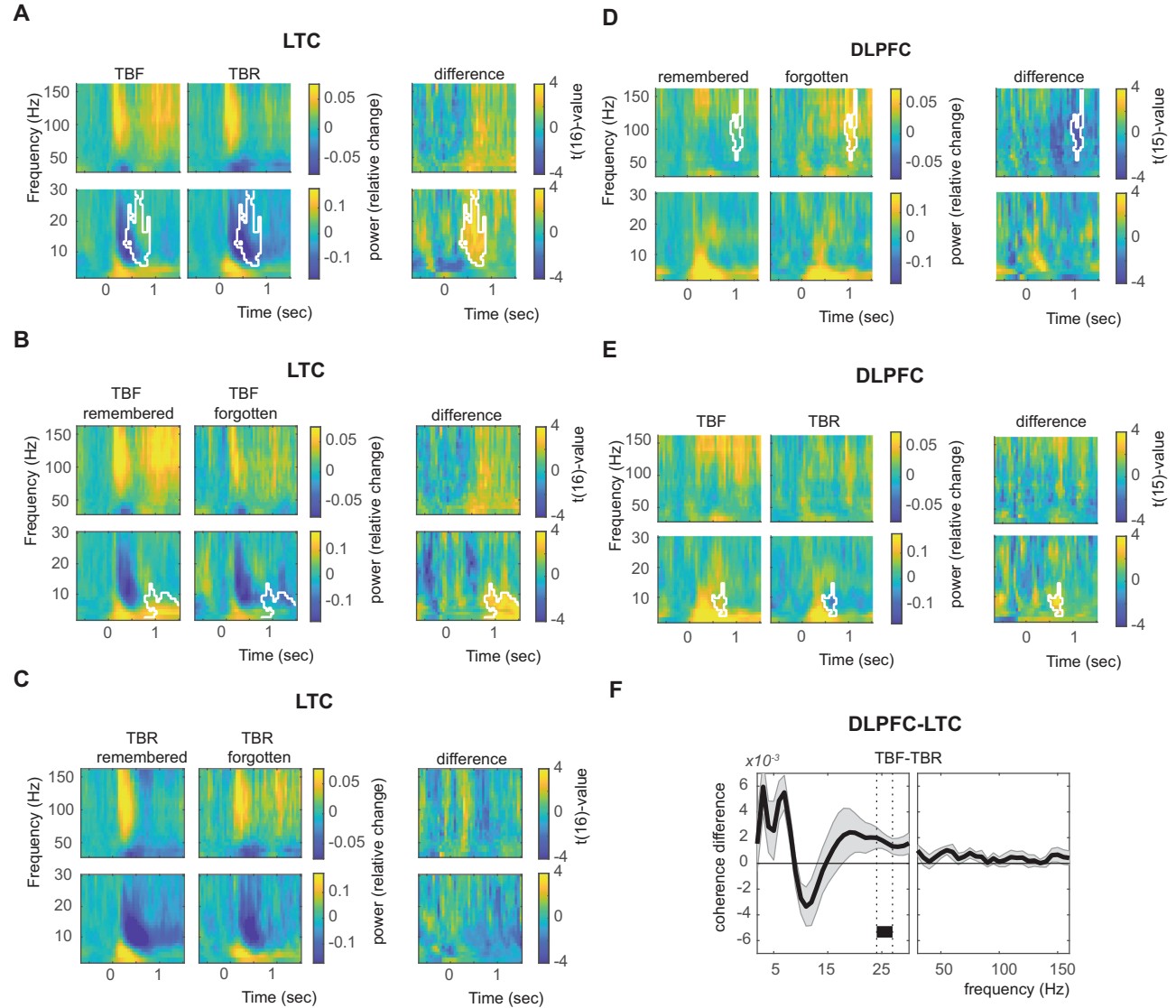

**Fig. 7 Cue effects. A** Instruction effect on power in lateral temporal cortex (LTC; $n = 17$). **B** Memory effect on power in LTC for to-be-forgotten (TBF) items. **C** Memory effect on power in LTC for to-be-remembered (TBR) items. **D** Memory effect on power in dorsolateral prefrontal cortex (DLPFC; $n = 16$). **E** Instruction effect on power in LTC. **F** Instruction effect on DLPFC-LTC coherence ($n = 10$). Shaded error bars indicate the standard error of the mean. White outlines and black bar indicate significance at the 0.05 level (cluster corrected). Source data are provided as a Source Data file.

compared with remembered words (55–160 Hz; 0.95–1.15 sec; $p = 0.022$; min $t(15)$ value $= -4.35$; Fig. 7D). In addition, we found more pronounced increases of theta/alpha activity for TBF compared to TBR cues (4–14 Hz; 0.5–0.75 sec; $p = 0.046$; max $t(15)$ value $= 4.67$; Fig. 7E; also see[22]). We did not find any interaction.

We also analyzed cue effects on coherence. We repeated the coherence calculations as performed for the ERS analysis, but using a longer encoding time window (coherence was calculated per trial over the 0–1 s window for each frequency point separately), and then averaged over all trials and channel pairs. Here, we found the main effect of instruction (24–27 Hz; $p = 0.04$; min $t(9)$ value $= -4.13$; Fig. 7F) showing stronger DLPFC-LTC coherence for the TBF compared with the TBR condition, but did not observe a memory or interaction effect.

We assessed possible inter-individual correlations between the ERS and the cue power effects. For each cue effect reported in Fig. 7, we calculated per subject the strength of the respective contrast, averaged over the frequency/time data in the cluster (separately for the LTC instruction effect, the LTC TBF memory

effect, the DLPFC instruction effect, and the coherence instruction effect). For example, for the LTC instruction effect, we subtracted the average TBF power from the average TBR power in the interval between 6 and 30 Hz; 0.35–0.85 sec. We then calculated per frequency bin used in the original ERS analyses the corresponding ERS effect (again for the LTC instruction effect subtracting the TBF ERS values from the TBR ERS values). Then, we correlated the ERS effects with the cue power/coherence effects, separately for each ERS frequency bin. Again, we corrected for multiple comparisons using cluster-based statistics. For the LTC instruction and LTC TBF memory effect, we found no correlations. Interestingly, however, for the instruction effect on DLPFC power, we found a significant positive correlation between the ERS effect and the cue effect ($p = 0.036$; 16–19 Hz; max $t(10)$ value $= 4.01$; Fig. 8). A similar correlation was found for the DLPFC-LTC coherence ($p = 0.022$, 15–17 Hz; max $t(9)$ value $= 4.93$). Investigating intra-individual correlations between cue power and ERS size across trials did not result in any significant effect (Supplementary Fig. 12). This indicates that participants showing more pronounced DLPFC alpha/beta power

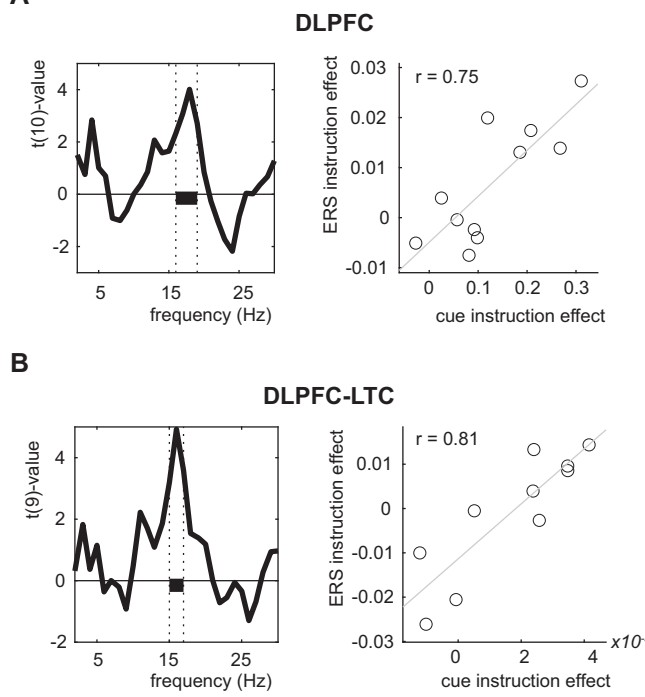

**Fig. 8 Inter-individual correlations across participants between the cue instruction effects and the encoding-retrieval similarity instruction effect. A** Correlation between dorsolateral prefrontal cortex (DLPFC) power (*n* = 11) and **B** DLPFC—lateral temporal cortex (LTC) coherence (*n* = 10). Correlations are estimated at the time/frequency ranges of the cue effects (see Fig. 7). The black bar indicates significance at the *p* = 0.05 levels. Scatterplots indicate the effect at the peak of the significant power/coherence effect. Circles indicate individual participants. Inset indicates the Spearman correlation value. Source data are provided as a Source Data file.

increases, and DLPFC-LTC alpha/beta coherence, also showed a more prominent stimulus-specific alpha/beta ERS effect of LTC patterns for TBF vs. TBR items.

## Discussion

Intentionally forgetting unwanted information requires the active involvement of DLPFC[22,33]. Up to now, it is unknown how DLPFC activity influences memory traces of subsequently forgotten information. Here, we aimed to unravel whether we could find selectively modified memory traces of intentionally forgotten information. We replicated previous findings that gamma activity patterns during retrieval of TBR items resembled the patterns during encoding when items are remembered compared to (incidentally) forgotten[8,11,42]. Interestingly, we found evidence for stimulus-specific ERS of TBF items, which depended on activity in the alpha/beta-frequency range. This effect was functionally relevant, as it was stronger for words that were actually forgotten. Our analyses using coherence and TE showed that ERS did not only occur in distributed patterns of alpha/beta power in the LTC, but also in item-specific signatures of DLPFC-LTC connectivity and of DLPFC→LTC top–down control. These results reveal that while incidental (i.e., accidental) forgetting results in a reduction of item-specific information, memory traces of intentionally forgotten information have a different fate: intentional forgetting forms item-specific representations that specifically rely on patterns classically associated with inhibitory control[16].

Although retrieval of TBR items depended on activity in the gamma frequency range, TBF items showed an effect at alpha/beta frequency. This suggests that forgetting instructions

(signaled by the TBF cues) do not only lead to an omission (Fig. 1D) or reversal (Fig. 1B) of encoding-related plasticity, but to an active modification of memory traces[21]. As participants are unaware at the moment of encoding whether an item will be instructed to be remembered or forgotten, this modification occurs after the initial memory trace has been built. Importantly, this modification is not a transformation that removes item-specific features (Fig. 1B), but rather a selective shaping of a unique set of inhibitory connections (Fig. 1C). We further showed that this selective effect on item-specific activity patterns is functionally relevant, as it is more pronounced for items that are actually forgotten.

Our results allow us to draw some conclusions on the specific neural signatures that support ERS of TBR and TBF items. TBR items showed increased gamma-band ERS when words were actually remembered as compared with when they were forgotten (Fig. 4A). When we directly compared ERS of intentionally vs. incidentally forgotten items (i.e., of forgotten TBF vs. forgotten TBR items), we found evidence for more pronounced ERS of item-specific patterns in the alpha/beta-frequency range for forgotten TBF items. This indeed shows that ERS in this frequency range is highly specific for intentionally forgotten items. The direct contrast between remembered TBR and remembered TBF items was not significant, suggesting that ERS of gamma-band activity patterns may also support memory for TBF items to a certain extent (even though it did not reach significance). Gamma activity patterns are usually taken as a proxy of neural activation and have been shown to correlate with multi-unit activity (e.g.,[43]). In addition, and specific for TBF items, more pronounced ERS in the alpha/beta-frequency range supported successful forgetting. Alpha/beta-frequency oscillations have been proposed as a measure of active inhibition[19,20], restricting representations to narrow temporal "duty cycles"[44] and generally lowering the amount of information that can be represented in a given brain region[35]. This inhibitory process has been shown to occur at different spatial scales, from inhibition of a full sensory region[19,45] or hemisphere[46] to more local, e.g., retinotopic effects[47,48]. Temporal cortex is no exception, as alpha/beta effects have also been found in temporal cortex as a response to increased processing demands during word processing[49,50]. Based on this previous research, we suggest that the frequency characteristics of TBF ERS effects reflect functional inhibition. However, it should be emphasized that this interpretation relies on the logic of reverse inference, which has been criticized since its validity relies, among other factors, on the specificity of the observed neural measures[51,52]. We would thus like to emphasize that the validity of our interpretation needs to be confirmed by future studies and in particular corroborated by formal analyses on the specificity of alpha/beta activity as a measure of functional inhibition.

As described in the introduction, an alternative explanation for directed forgetting is that forgetting instructions terminate the encoding process. However, this cannot easily explain our finding of higher alpha/beta ERS for TBF vs TBR words, but would rather suggest absence (or reduction) of ERS (Fig. 1D).

Importantly, we did not find evidence for an overall difference in either alpha/beta or gamma power in the time window of the ERS effects during retrieval. Such an overall power change might have been an indication that the functional excitation–inhibition balance has changed[53]. This commonly occurs during active cognitive tasks such as memory and attention[27,54]. More pronounced overall excitation–inhibition imbalances have also been associated with neuropathological mechanisms in various diseases[55,56]. Instead, we found item-specific changes in the alpha/beta and gamma patterns of activity. Therefore, our results provide evidence of instruction- and memory-dependent rewiring of functional connectivity patterns. This is exactly what would be

expected from an adaptive memory mechanism whose function is to selectively incorporate forgetting or remembering attempts into item-specific memory traces, whereas maintaining the homeostatic balance between excitation and inhibition.

In addition to ERS of distributed power values, we found a parallel effect based on DLPFC-LTC coherence patterns. Although most previous studies on ERS have used either band-pass-filtered time-domain data or power estimates, there is no reason not to use other features as long as they can be estimated in single trials[57,58]. Indeed, our analysis of stimulus-specific coherence patterns showed that inhibitory prefrontal control is not exerted in an unspecific manner, but specifically targets item-specific representations of unwanted stimuli. Our results further demonstrate that inhibitory DLPFC-LTC interactions and top–down control are not transiently exerted processes during the forgetting cue; instead, these interactions are permanently incorporated into the memory traces of the respective items, at least for the time course assessed in our study.

There are various measures of directed connectivity, such as Granger causality analysis and the phase slope index[59]. We here chose TE, a model-free information-theoretic measure (i.e., not based on linearity assumptions) that has been successfully applied in various neuroscience contexts[41,60,61]. Using this metric, we observed significant item-specific ERS of successfully forgotten TBF items in directional alpha/beta-frequency patterns from DLPFC to LTC, indicating top–down connectivity. Alpha/beta-frequency oscillations have not only been described as a measure of active inhibition as described above, but also as a signature of feedback interactions[62,63] (but see ref. [64]). These frameworks are not mutually exclusive though, because feedback connectivity may predominantly serve to inhibit bottom–up information flow. Importantly, we found that the magnitude of TE did not generally differ during retrieval of TBF vs. TBR trials. Thus, patterns of TE-based ERS indeed reflect item-specific top–down control. Together, these results indicate that active forgetting is associated with a selective modification of item-specific connections from DLPFC to LTC.

Most previous studies investigating directed forgetting have focused on the cue-period window[14,15,22]. Our analysis of activity during the cue-period confirms that during the cue, DLPFC exhibits increased activity for the forgetting instruction (Fig. 7E). Moreover, our results show the effects of forgetting instructions in the LTC (Fig. 7A–C) and also indicate that DLPFC-LTC coherence is stronger for forgetting than remembering cues. Finally, we could show that these cue-period effects, previously associated with inhibition[15], are correlated to the strength of the ERS effects across subjects. This further strengthens our interpretation that directed forgetting depends on a selective modification of inhibitory patterns present during encoding. There is no doubt that the cue to instruct needs to have a consequence on the memory trace, perhaps via hippocampal connections that are frequently recruited in directed forgetting tasks (e.g.,[22,39,40]). Interestingly, however, when we regressed out cue-related activity, our main ERS effects remained (see Supplementary Fig. 7B and C). Although activity during the cue-period should modify the memory trace, this does not necessarily imply that activity during the cue period reflects those aspects of item-specific activity that occur at encoding and retrieval in response to a word. During the cue, activation of memory traces requires active retrieval of sensory representations. During encoding and retrieval, this is not needed as the word itself is presented. The ERS thus reflects the response to a sensory stimulus, combining both perceptual processes and memory-related item-specific information associated with the sensory representation. More

specifically, the recognition probe needs to first be processed along the sensory-semantic (word) processing pathways, elicit the stored index in the hippocampus, to then reactivate the missing contextual information. By contrast, during the cue period, the word is not presented, which may result in different activation patterns.

As participants during encoding are unaware of the instruction to forget a particular item, it is striking that ERS effects of TBF items rely on neural signatures that are classically associated with inhibitory feedback. This means that item-specific patterns of functional inhibition and of inhibitory connectivity are already present when the items are initially presented, i.e., prior to the instruction to forget them. It could be that this effect reflects an automatic feedback signal created by DLPFC to increase the specificity of sensory activation in the LTC[65,66]. Active forgetting mechanisms could latch onto this signal to actively inhibit a memory trace in a stimulus-dependent manner. Alternatively—or in addition—it could be that this effect is specific (or particularly pronounced) in the current task in which participants are aware that half of the items need to be forgotten. Thereby, a strategy would be to incorporate a functional inhibitory trace at encoding and after the cue decide which trace to keep[67,68]. In the future, employing a list-method directed forgetting paradigm[69] may allow disambiguating these options. If ERSs of items from the TBF list again rely specifically on alpha/beta frequencies, this would speak in favor of a more general and automatic feedback signal.

Directed forgetting paradigms have the intrinsic limitation that they do not contain a control condition, which may be considered a problem for most studies on voluntary memory suppression during encoding[12]. With respect to our results, it may be that the ERS effects in the alpha/beta-frequency range that we observed for TBF items could still be weakly present also in a control condition without explicit instruction.

In both the item-method and, in particular, the list-method directed forgetting paradigm, inhibition has been conceptualized as one specific context that differs from the remembering context. Our results speak against overall context reactivation as all our analyses control for overall within-condition correlations (e.g., comparing ERS of same vs. different forgotten TBF items during encoding and retrieval). Furthermore, we did not find any evidence for instruction-specific cue-retrieval similarity (Supplementary Fig. 7).

The time windows containing item-specific information during encoding and retrieval started relatively early, i.e., within the first 100 ms after stimulus onset during both encoding and retrieval. It is notable that the timing of encoding-retrieval effects seems to be variable across studies. Some studies have found reactivation of broad-band activity from relatively early encoding and retrieval time periods, similarly to what we found here[42]. This seems to contrast with results from human single-unit recordings in the medial temporal lobe showing reactivation of item-specific information of substantially later activities[9,70]. Some of this discrepancy may be owing to differences in tasks between studies: ERS has been analyzed in both cued recall and recognition memory paradigms (used in this study). During cued recall tests, participants see a cue and need to perform a pattern completion process in order to successfully retrieve a correct memory representation corresponding with the cue. During recognition memory, the recognition probe is first processed along the sensory-semantic (in our study, word) processing pathways after which missing contextual information is reactivated via memory processes in the hippocampus. The early ERS effects found in our study during retrieval likely reflect initial sensory and semantic

processing steps rather than an active memory process that reactivates information about the encoding context.

Our results are highly relevant for the broader field of extinction learning, where the sudden omission of an expected reinforcement leads to the unlearning of previously acquired behavior[25]. Specifically, there is an ongoing debate on whether extinction is a result of new learning to inhibit a specific behavior[71] or whether it is owing to weakening and eventual erasure of the original behavior[72]. Drosophila research has shown that either of these processes may occur, depending on the specificities of the unlearning process[73], but in humans, the results are still conflicting[25,74]. Our results point to the relevance of inhibitory learning for voluntarily forgotten information. Of course, this does not exclude that other—e.g., more automatic—unlearning conditions may in fact lead to an erasure of memory traces in humans as well[75,76]. In the future, it will be important to study whether memory traces can be actually deleted, or whether memory suppression invariably relies on a modification of traces that may subsequently be recovered.

Our study shows different frequency-specific ERS patterns for intentionally remembered and forgotten information: although ERS of successfully remembered information relied on high gamma activity patterns, it depended on alpha/beta activity when participants intentionally and successfully forget items. We further show that intentional forgetting involves the selective modification of stimulus-specific patterns of top–down control. These results have far-ranging implications for how the brain stores information as well as for memory control and extinction learning, showing that intentional forgetting is an active process in which inhibitory feedback connections of the original memory trace are actively maintained.

## Methods

**Patients**. In total, we recorded 42 patients with pharmacoresistant epilepsy who had been implanted with iEEG electrodes for diagnostic purposes. We had to exclude 12 patients because of missing data (either in the EEG or in the post-MR implantation scheme). Six more patients were excluded owing to poor behavioral performance (more false alarms than hits in the TBR condition and/or fewer than 20% remembered TBR items). This resulted in a total of 24 patients. Patients had a wide variety of electrode positioning (see Fig. 2C and Supplementary Fig. 1; LTC: $N = 16$; frontal cortex: $N = 16$; DLPFC: $N = 15$; parietal cortex: $N = 10$; hippocampus: $N = 13$). Our analysis was restricted to all electrodes in areas that were free of morphological alterations identified via MRI and were outside of the seizure onset zone. The activity during the cue period of this dataset has been previously described[22,77]. The study was approved by the ethics committee at the University of Bonn and was in accordance with the Declaration of Helsinki. All patients provided written informed consent.

**Recordings**. Cortical recordings were gathered from stainless steel subdural strips or grid electrodes. Medial temporal electrodes were multi-contact depth electrodes implanted stereotactically along the longitudinal axis of the hippocampus. The sampling rate was 1,000 Hz with a linked mastoid reference, and data were band-pass filtered online using the digital EPAS system and its implemented Harmonie EEG software v7.0a [0.01 Hz (6 dB/octave) to 300 Hz (12db/octave); Schwarzer, Munich, Germany; Stellate, Montreal, Canada]. All recordings were performed in the Department of Epileptology of the University of Bonn, Germany.

**Electrode localization**. MRI data were segmented using Freesurfer's automatic reconstruction software[78]. Pre- and post-surgery MRIs were aligned, and then electrodes were positioned manually using FieldTrip[79] running under Matlab2016a. This was repeated twice to check the reliability of the manual positioning. Any electrode that was not positioned within 7.5 mm Euclidian distance of the initial positioning was thoroughly checked and underwent the procedure again. For the other electrodes, the average location was used. We used the hull method introduced by Dykstra[80] to back-project the ECoG electrodes onto the cortical surface. Anatomical labels were given for all electrodes using FreeSurfer's cortical parcellation. Then MRIs were normalized to the MNI brain via surface normalization and the transformation matrix was applied to the electrodes. The anatomical labels were further combined into five regions of interest (ROI): parietal, occipital, lateral temporal, frontal, and medial temporal areas. We do not report on the occipital ROI as only two patients had occipital channels. For all other ROIs we had the following channel numbers: 27.3 ± 11.1 LTC channels (range, 7–45; total, 437 channels); 14.2 ± 4.7 DLPFC channels (range, 8–23; total, 213 channels); 10.7 ± 5.6 parietal channels (range, 5–24, total, 107 channels). 11.5 ± 4.7 hippocampal channels (range, 6–23; total, 213 channels).

**Paradigm**. The paradigm consisted of three different phases. In the first (study) phase, patients were presented with written words on a screen (300 ms). After a variable delay (1500–2000 ms) with a blank screen, a fixation cross appeared. The color of the fixation cross served as a cue to indicate whether the previously presented word was TBR (green color) or TBF (red). This cue stayed on the screen for 2000 ms. After a 500 ms delay, the next word was presented. A total of 50 unique words were presented in random order. The study phase was followed by a free recall phase (not analyzed here). After the free recall phase, a recognition memory test was conducted. Each word was presented for 300 ms and the patient was asked to indicate via button press whether it was presented before (old word) or not (new word). Participants did not receive any instructions with regard to the speed of their response and did the experiment at their own pace. Per block, all words from the study phase were presented (25 TBR and 25 TBF), randomly intermixed with 50 new words. Data of multiple blocks with unique words were collected per patient (2–4 blocks, mean: 3.69 ± 0.60 in the 16 patients with LTC channels).

## Analysis

*Preprocessing and time-frequency analysis*. A band-pass filter (0.5–200 Hz) and a discrete Fourier transform filter were run over all data. Data were demeaned and epoched around word onset during the encoding and retrieval phase (−2 s to +3 s with respect to word onset). Trials with a variance of >3.5 standard deviations above the mean were removed from further analysis. Data were re-referenced to the average of all channels (only including healthy, non-noisy channels). All trials were then subjected to time-frequency analysis. For the low frequencies ranging from 2 to 30 Hz, wavelet analysis was conducted (1 Hz step size), using variable widths from two to six cycles (linearly increasing) and a time range between −0.65 s pre-stimulus to 1.5 s post-stimulus (0.05 s step size). Activity at high frequencies (30–150 Hz, step size of 5 Hz) were subjected to a multi-taper analysis (dpss taper, 10 cycles data at 0.5 cycles smoothing). All power values were single-trial log-transformed and baseline corrected (using a baseline window from −0.3 s to −0.1 s pre-stimulus). Again, extreme values were removed (data with a mean power above 3.5 standard deviations above the mean).

*Power-based ERS*. For each ROI, we computed Spearman correlations between every encoding and every retrieval trial. The features for this correlation consisted of the power values across all channels in the respective region of interest vectorizing channel x time data in a time window between 0.1–0.5 s (the time period of the sensory-evoked response). Since we were particularly interested in distinguishing high-frequency (putatively excitatory) and low-frequency (putatively inhibitory) effects, the correlations were conducted at each frequency separately. This analysis resulted in a correlation matrix for each frequency between all matching encoding-retrieval word pairs as well as all non-matching encoding-retrieval word pairs (see Fig. 3A). The difference between matching and non-matching word pairs corresponds to item-specific ERS. In detail, we compared ERS values in three contrasts:

1. *Matched versus non-matched ERS in TBR trials*: this contrast compared the mean correlations between matching and non-matching encoding-retrieval TBR words, irrespective of subsequent memory. In other words, we compared the correlations when the same TBR word was shown during encoding and retrieval (on-diagonal dark orange values in the matrix in Fig. 3A) with correlations when different words were shown at encoding and retrieval (off-diagonal light orange values). Note that by only choosing off-diagonal TBR words (instead of all off-diagonal words including non-matched categories such as TBR encoding—TBF retrieval) we control for any overall differences between the TBF versus the TBR condition that are not item-specific.

2. *Matched versus non-matched ERS in TBF trials*: this contrast compared the mean correlation between same and different TBF words during encoding and retrieval (analogous approach to contrast #1).

3. *Item-specific TBR versus TBF ERS*: this contrast compared contrasts #1 and #2 (analogous to an interaction analysis).

We focused on the LTC ROI (see Supplementary Fig. 4 for other ROIs). Statistical comparisons of averaged on-diagonal vs off-diagonal values per condition were performed via paired $t$ tests over subjects for each frequency separately (after Fisher-z-transforming the correlation values). Correction for multiple comparisons was performed via cluster statistics. Clusters were defined as frequency-sequential points that had each $p$ values lower than 0.05 based on permutations. The sum of $t$ values entering this cluster was defined as a dependent variable for second-level non-parametric testing. A null-distribution was created by shuffling patient condition labels (1000 permutations). For each permutation, the largest surrogate cluster would enter the null-distribution. $P$ values were then defined as the proportion of clusters in the null distributions with summed t-values higher than values of the empirically observed clusters. All subsequent analyses following cluster corrections were performed accordingly.

*LTC subregion analysis*. To scrutinize the location of the ERS effects in greater detail, we divided the LTC into five subregions defined by the AAL atlas: fusiform gyrus (69 electrodes/14 patients), superior temporal gyrus (40

electrodes/9 patients), middle temporal gyrus (120 electrodes/16 patients), inferior temporal gyrus (122 electrodes/15 patients), and temporal pole (47 electrodes/14 patients). We repeated the analysis of the power-based ERS in the respective significant frequency ranges of interest and performed a paired $t$ test on the average of this frequency range. Effects were Bonferroni corrected for multiple comparisons (i.e., the five subregions).

*Time-shifting ERS analyses.* We also investigated possible temporal generalization or time shift of the correlations between encoding and retrieval. We used the frequency ranges of the significant non-time-shifted effects, and repeated the ERS analysis for varying time-shifted encoding and retrieval time windows (centered at $-0.2$ to $+0.8$ s using again windows of 400 ms like in the original analyses). Statistical comparisons were performed similar to before (correcting for multiple comparisons used cluster-based permutation testing), but now clusters were defined based on temporal proximity.

*Memory-specific ERS.* To investigate whether ERS differed depending on the participants' memory, we repeated the contrasts for TBR and TBF as described above, but separately depending on whether the participant had actually remembered or forgotten the specific word.

*Coherence-based ERS.* For the coherence analysis, some of the preprocessing steps were conducted separately. Specifically, preprocessing steps were identified as those described for the analysis of power-based ERS until the re-referencing step: for the coherence analysis, we calculated bipolar derivatives of all channel pairs in order to ensure that no spurious coherence was induced through volume conduction. Again, trials with a variance above 3.5 standard deviations above the mean were removed. Then, we extracted the Fourier spectra. For low frequencies, we used the same parameters as described above. Then, we calculated the coherence between each ipsilateral pair of DLPFC channels and LTC channels. The DLPFC was chosen as it is crucial for controlling directed forgetting[22,33–35] and was defined via the AAL atlas (regions 7,11,13 [left] and 8,12,14 [right]). We then applied the same contrasts as described above, but used the coherence values as features for the correlation matrix. We assessed the contributions of the five LTC subregions as well.

*TE-based ERS.* To estimate the directionality of item-specific ERS of DLPFC-LTC connectivity patterns, we calculated trial-by-trial TE values using the Gaussian Copula method from the GCMI toolbox[60]. TE is the mutual information between X at lag t0 and Y at lag t1 conditioned on Y at lag t0[81,82]. For preprocessing, we first band-pass filtered the bipolar-referenced data between ±2 Hz around the significant frequency range of the coherence ERS effect using a one-pass zero-phase firws filter of order 200 (controlling for directionality in the filter). We took the absolute of the Hilbert transform and baseline corrected the data (time window of $-0.2$ s before stimulus onset to 0). Data were then re-epoched from $-0.05$ to 0.5 sec. This window resembled the original analyzed window (0.1–0.5 sec) but was slightly extended to account for the delay that was needed to calculate the TE values (here up to 0.15 sec). Variance outlier trials above 3.5 standard deviations were removed. TE was calculated trial-by-trial for every ipsilateral channel pair separately correcting for the bias (calculating the copula per trial and channel). This was performed for the LTC→DLPFC and DLPFC→LTC direction at lag values ranging between 0.05 and 0.15 s, thus always maintaining a 400 ms analysis window (corresponding to previous results reported in ref. [22]). The memory-specific ERS analysis was repeated as described above but using the TE values as features for the correlation matrix, separately for LTC→DLPFC and DLPFC→LTC TE. We first took the average of the delays as the TE of interest[37] using paired t tests, but also looked at the different delays and corrected for multiple comparisons using clusters statistics (with clusters formed across adjacent delays). Again, we also investigated the contributions of the five LTC subregions.

*Cue-based analysis.* Last, we evaluated the time-frequency patterns in DLPFC and LTC during the cue period in order to investigate their possible contributions to the ERS effects. The time-frequency analyses were repeated as before but now aligned to cue onset. Then, we estimated the effects of instruction, memory, and their interaction for a window ranging from $-0.2$ sec to 1.5 sec around cue onset (using the same cluster-based statistics as before). In addition, we calculated the coherence between LTC and DLPFC during the cue (for a period of 0–1 s) and investigated the role of instruction and memory on this coherence.

**Reporting summary**. Further information on research design is available in the Nature Research Reporting Summary linked to this article.

## Data availability

The raw iEEG and MRI data generated in this study have been deposited in a local database. The iEEG and MRI data are available under restricted access as it contains personally identifiable information and patients have not consented to data distribution. Access can be obtained by sending a request to sanne.tenoever@mpi.nl or

nikolai.axmacher@ruhr-uni-bochum.de. The raw iEEG and MRI data are protected and are not available owing to data privacy laws. Source data are provided with this paper.

## Code availability

Code to generate the figures belonging to the main findings can be found in the Supplementary Information/Source Data file. For all other code access can be obtained by sending a request to sanne.tenoever@mpi.nl.

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

## Acknowledgements

N.A. received funding by the Deutsche Forschungsgemeinschaft (DFG, German Research Foundation)—Projektnummer 316803389—SFB 1280 as well as via Projektnummer 122679504—SFB 874, A.S. received funding from the Netherlands Organization for Scientific Research (NWO; VICI grant 453-15-008). We thank Marie-Christin Fellner for useful suggestions for the manuscript.

## Author contributions

Conceptualization, S.O. and N.A.; methodology, S.O. and N.A.; formal analysis, S.O.; resources, C.R.O. and N.A.; data curation, C.R.O.; writing—original draft, S.O.; writing—review & editing: S.O., A.S., C.R.O., and N.A.; visualization: SO; supervision: N.A.; funding acquisition: A.S. and N.A.

## Funding

## Competing interests

The authors declare no conflict of interest.
