## [Peer Review File · Nature Communications]

An engram of intentionally forgotten informationReviewers' comments:

Reviewer #1 (Remarks to the Author):

In this manuscript, the authors present intriguing findings regarding the mechanisms of motivated forgetting. Using ECoG, they show that the lateral temporal cortex (LTC) shows similar frequency responses to word onsets in a directed forgetting task: response pattern similarity for to-be-forgotten (TBF) words peak at the alpha-beta bands; and for to-be-remembered (TBR) words at the gamma band. Similar results are shown for forgotten vs remembered words and when considering DLPFC-LTC coherence. Using transfer entropy they show that the direction of forget-related inhibition is from DLPFC to LTC.

The presented results are intriguing and quite surprising, given the fact that the encoding epoch used for the encoding-retrieval similarity (ERS) analysis occurred before words were designated as TBF or TBR. After addressing some possible confounding factors and conducting some additional analysis, I believe this manuscript would be suitable for publication in this journal.

Major issues:

1. A major challenge the authors face is explaining how neural traces at word onset serve to predict its subsequent association with an experimental condition that has yet to be assigned. Unfortunately, I'm not sure they do a very convincing job in this respect. If I understand correctly, their model suggests that the excitation-inhibition balance - both within LTC and between DLPFC and LTC - always includes the infrastructure for suppression, conveyed by the inhibitory circuits. When an item is intentionally suppressed, according to Fig 1c, the inhibitory synapses overcome the excitatory ones, breaking the balance; whereas successful retrieval involves the excitatory circuits prevailing. There are a number of issues with this hypothesis that need to be addressed. First, it is unclear why retrieval shouldn't simply involve a full recapitulation of the original memory trace, including both excitatory and inhibitory (balanced) nodes. Second, the literature on excitation-inhibition balance in neural circuits suggests it is maintained over time, and breaking the balance is thought to relate to pathology (e.g., Markicevic et al., 2020; Dehghani et al., 2016). If the authors indeed suggest there is some function to breaking the balance, they should state so and link their hypothesis to the existing literature. Third, it makes little intuitive sense that cross-cortical inhibitory circuits are pre-embedded in every memory engram. Previous studies, cited in the manuscript, show that DLPFC is active in memory suppression, but to the best of my knowledge all refer to the inhibition-period itself, not the pre-inhibition encoding period (e.g., Anderson et al., 2004). I wish to stress that I do not discourage the authors from presenting new ideas or challenging the literature, nor am I claiming that the authors are wrong. On the contrary - I believe their results are challenging, and therefore require a more thorough literature review and an in-depth comparison with current ideas and previous findings. The current version of the manuscript avoids the complexity involved with the notion that memories are pre-wired to be forgotten, which is, to my understanding, a major point of the paper.

2. The manuscript is missing crucial information regarding the response period, which may have substantial effects on the main results. It is unclear whether the participants had to state if the word is old/new as fast as possible and how their response was conveyed (vocally?). This is important, because it may mean that the "retrieval" period was actually contaminated by responses, and different response patterns between TBR/TBF and remembered/forgotten words may have been at the core of ERS results. For example, the difference in response times for different conditions (e.g., slower for new vs old) may have affected the correlation patterns. The authors deal with some of these effects by presenting and comparing time-frequency patterns (e.g., Fig3C), but there may be more nuanced differences that do not result in statistical significance. This is especially true given the small effect sizes that are reported ($|\rho| \leq 0.02$). The authors should address this possibility with more detailed methods, RT data, and additional analyses.

3. The authors emphasize that their data provide evidence that intentional forgetting does not erode the memory trace, but rather suppresses it, leaving it intact (e.g., line 24). This is based on

the finding that there is some ERS for forgotten memories. However, this does not show that the memory trace is intact. It still might be substantially - but not entirely - eroded. I suspect the authors' claim is somewhat of a strawman argument since very few people (if any) would suggest the neural trace is reduced to zero. Taken together, with the small effect size of the ERS analysis, I suggest using more nuanced language. If the authors choose to leave this claim in the manuscript, I think it would be helpful if they explicitly state how their data would have looked like have the trace been erased (i.e., present their H0 directly). This may make this point more data-based and easier to comprehend.

4. A related point concerns the lack of a real control condition. Comparing TBF and TBR one to the other leaves many questions open. For example, do TBF instructions reduce ERS values at high frequencies (Fig 3b) when compared to no instructions at all? Do TBR instructions increase them or leave them unchanged? I'm not suggesting running more experiments, but rather addressing this limitation and changing the language accordingly. This relates to the previous point as well, since it could be that TBF ERS would have always been lower than a non-included control (e.g., that control words would lie consistently at $\rho=0.02$ in Fig 3b), indicating that TBF low-frequency ERS is lower than control (and TBR representation even lower). TBF>TBR does not imply TBF>control (and same for TBR, of course).

5. Many of the issues mentioned up until this point concern the authors' choice to analyze the word-onset periods during encoding and retrieval. An analysis of the post-cue epoch may add some information regarding the neural processes. For example, does alpha-beta power following the TBF cue add information regarding the memory fate of the presented word? Does it mediate the ERS effect? How do DLPFC-LTC interactions look like during this time period and how do they correspond to the coherence-based ERS?

6. The RSA analyses seem to diverge from the manuscript's narrative and I am not sure they substantially add to it. As a reader, I felt that this section somewhat threw me off balance and disrupted the natural flow of the manuscript, and I would consider removing it to improve the paper. If the authors choose to leave it in, however, I would suggest avoiding the use of reverse inference in the discussion (i.e., "area X is involved in Y, so process Z that also happens in area X must also concern Y"). There is no real evidence showing that any of the word-related metrics measured have anything to do with memory suppression.

Minor issues:

7. Line 338 (and perhaps elsewhere) - the claim that higher ERS reflects "strengthening of inhibitory circuits" seems to be somewhat exaggerated. It may suggest that the same inhibitory circuits are active during encoding and retrieval, but that doesn't necessarily mean that these circuits were strengthened.

8. Line 125 - "TBF words were remembered better than chance". In fact, this is not a comparison with chance level (which is 50%), but with the false alarm level. Please rephrase.

9. The term "interaction" is used throughout the manuscript to describe "difference". Interaction suggests that there are two independent factors that jointly affect some measure. No interaction analysis is conducted in this paper, so the term may be misleading. Even in cases where there are two independent variables (e.g., Fig 2, TBR/F x Hz), the used method is not one that may reveal interactions and the term interaction, when used, actually means "difference".

10. Please explain why different time-frequency methods were used for <30 Hz and >30 Hz frequencies. Would using the same methods change the results?

11. Notes regarding the figures:

a. Fig1 - I suggest using "unsuccessful encoding" instead of the confusing term "unsuccessful memory" (which could be interpreted as a memory for a bad event).

b. Fig2c (and SuppF1) - each dot represented an electrode, not a patient. The legend is somewhat misleading.

- c. Fig2a bottom - please indicate when the participant response was allowed.
- d. Fig 5 - please indicate what the timeframe used for this analysis was.
- e. Fig 5b - the lines are overlapping and hard to see. Perhaps reduce the line width.
- f. Fig 5c - please mention that this panel reflects time-frequency responses in DLPFC.
- g. SuppF5 - The word "brackets" should be replaced with "parentheses". Also, it would be helpful if the tested frequencies were mentioned in each subplot.

Reviewer #2 (Remarks to the Author):

This paper describes electrophysiological activity in the lateral temporal and prefrontal cortex using a directed forgetting paradigm. Overall, the authors do an admirable job of walking through the implications of a series of complex analyses. It is interesting and novel to look at this paradigm in the setting of intracranial recordings.

Power extraction follows reasonable methods. Entropy for directional connections is not commonly done, but seems reasonable overall and the authors provide sufficient motivation for its use.

In terms of the chosen contrasts, I would have assumed that one would compare forgotten TBF items to forgotten TBR items (to identify a signal related to intentional rather than unintentional forgetting) and recalled TBF to recalled TBR items (to understand how reinstatement differs when someone intends to forget it vs not). In Fig 4, it looks like these contrasts would show significant differences in line with author interpretation, but I don't see this as a direct contrast in the manuscript. Along these lines, a signal that is truly characteristic of intentional forgetting should 1) be different than unintentional forgetting (see above) and 2) different than the signal for novel items. I understand that this would require looking at similarity between novel items and TBF items as a class (rather than item by item), but it would be reassuring that the TBF reinstatement does not reflect some non specific (less interesting) response to novel items.

More generally, I am having trouble understanding how to interpret reinstatement differences in the paradigm. It appears to me as though the participants are cued regarding the TBR/TBF classification for the word after the item is shown. This would mean that at the time of item presentation, the individual does not know if they were supposed to be remembered or not. That would lead me to assume that TBF items should exhibit lower reinstatement, because the representation of the item at retrieval is now different than it was at initial encoding, as the subject would still have been trying to remember the item at the time the encoding-related brain activity was first recorded. The authors examine the issue of post-presentation modification of a memory trace, such that reinstatement in the TBF condition is evidence of strengthened inhibitory connections. Reinstatement of this connection implies that it is present at encoding (prior to the forget signal), but then is normally pruned/eliminated during purposeful memory such that it is not reinstated? To phrase another way, the concept of an inhibitory top down connection facilitated by the DLPFC and characterized by alpha/beta activity seems reasonable, but greater reinstatement in this connection would imply that the connection exists during encoding (before the TBF/TBR cue) and then is preserved at retrieval uniquely in the TBF condition. This seems far less intuitive than the addition of some inhibitory signal that occurs in response to the TBF/TBR cue. Would the authors predict that the TBR/TBF cue would elicit the alpha/beta signal they observed being reinstated? Could this period of time be analyzed to reinforce the conclusions regarding the inhibitory model postulated by authors?

An alternative interpretation is that the frontal cortex is providing context information rather than inhibitory control. This would be the case if TBF "forgotten" items were actually remembered, but the individual also retrieved the TBF cue association for the item and then reported it as novel/not remembered based on this associative information. In this interpretation, greater reinstatement in alpha/beta for TBF is not the failed pruning of an inhibitory connection, but the preservation of associative information. The authors may mention this possibility in the Discussion, or specifically why they reject this idea (maybe something in the behavioral data that I missed)?

My prior assumption is that a retrieval/memory control signal (post retrieval monitoring) is more likely to come from the RIGHT DLPFC, but I don't know if the authors specify the hemisphere for

recordings or break down by hemisphere in the analysis. Similarly - is the non-dominant lateral temporal cortex associated with semantic representations? Are the electrodes implanted in a mix of dominant/non-dominant hemispheres? My opinion is that the analysis on page 15 raises more questions than it answers. If there is an inhibitory signal in the alpha beta range, but this is also related to word frequency, it is hard to fit everything together. For the issue of controlled forgetting, I agree it is reassuring that gamma reinstatement is unique for recalled items (not just activity related to reading words), but I'm not sure a link to semantic association is especially helpful for the main analysis.

Methodologically, the reinstatement analysis seems to assume that encoding/retrieval similarity occurs at the same time points, ie no shifting of encoding relative to retrieval is performed, as in the Yaffe paper they cite. This paper shows that retrieval-related activity occurs relatively earlier in time. The authors re calculate their similarity values after changing their window, but this is not the same thing as shifting encoding relative to retrieval.

Can the authors report the degrees of freedom in their statistical tests (pre clustering)? Did the authors shuffle trial labels as part of clustering shuffle? This is often appropriate in iEEG data. Reporting the initial test statistic is also useful along with the clustering based p values. Also, how many individual connections were included in the analysis in Fig 5? The authors are aggregating temporal cortex recordings over a large area, so in subjects with frontal electrodes, I can imagine that hundreds of connections are potentially being averaged together?

Finally, the authors mention that they Bonferroni corrected results across 5 temporal cortical regions when trying to pin down anatomical specificity for the effects. Did they also correct across the other brain regions they examined for their LTC results (shown in supplemental material)?

I think this paradigm is a good one to link to iEEG. The Discussion and interpretation go a bit further than the data would seem to justify (or is needed to amplify the novelty of results), especially the paragraph starting on line 377 which is overly speculative. The paragraph starting line 404 even more so. I am not sure the connections identified by the authors can clearly be called inhibitory, other than to say that previous studies report that connections in this freq range exhibit inhibitory properties in other paradigms and other behaviors.

Reviewer #3 (Remarks to the Author):

This manuscript reports a study investigating the electrophysiological signatures of forgetting, using human intracranial recordings from the lateral temporal and frontal cortices of epilepsy patients. Neural patterns are measured while patients encode single words, and are later asked to distinguish between previously seen and unseen words in a recognition memory test. Critically, the encoding of each word is followed by an instruction to either remember this word (to-be-remembered, TBR), or to intentionally forget the word (to-be-forgotten, TBF). Neural patterns from encoding and recognition are then compared by means of encoding-retrieval similarity (ERS), a measure typically used to indicate memory reinstatement. The central finding is that both TBR and TBF words show significant ERS, suggesting reinstatement of the memory traces originally formed during encoding. Critically, this similarity is found in high gamma frequencies for TBR words, and in lower alpha/beta frequencies for TBF words. This tendency is present for similarity based on spectral power, as well as similarity based on cross-electrode connectivity patterns. In terms of connectivity, the data additionally suggest that TBF words that were actually forgotten show reinstatement of low-frequency connectivity from frontal (DLPFC) to lateral temporal cortex. Apart from some statistical weaknesses, the study is solid in terms of data analysis and reporting, and my concerns are primarily of conceptual nature.

The authors interpret their findings as showing that intentional forgetting depends on the strengthening of "inhibitory engrams", supposedly reflected in alpha/beta encoding patterns that are then reinstated for TBF (but not TBR) items. Although the manuscript is well-written, and the story is intriguing, the data do not seem to provide any direct evidence for inhibitory engrams and their strengthening. There is thus a large gap between the actual data and their interpretation, and

conclusions like the “Our results demonstrate that successful intentional forgetting is due to a selective strengthening of item-specific inhibitory top-down connections, rather than an erasure of the inhibited memory traces” seem overstated.

Conceptual concerns regarding the interpretation of the results:

(1) The interpretation of alpha/beta patterns as inhibitory engrams strongly relies on the assumption that alpha/beta power is a direct and specific index of inhibitory connectivity within a neural assembly that forms the engram. The authors cite the alpha inhibition hypothesis (Klimesch et al, 2007) and the inhibitory gating hypothesis (Jensen & Mazaheri, 2010) as evidence for such a mapping. As far as I am aware, neither of these ideas assumes that alpha/beta power in any way reflect inhibitory connections within a neural circuit or assembly. If anything, slice preparations and modelling work show that gamma oscillations are strongly related to inhibitory interneurons. Having said that, I doubt that any robust evidence exists to map an oscillation in a limited frequency band to excitatory connections within a neural circuit, and an oscillation in a different frequency band to inhibitory connections. The authors need to make a very strong case to justify this central assumption.

(2) Second, there is a mismatch between this interpretation and the measurements used to support it. All encoding-retrieval similarity results in this study come from relatively early time windows of stimulus processing, 100-500msec after the onset of the words during encoding and recognition. What makes the authors think that reinstatement in LTC happens as early as 100sec after the onset of a recognition cue? Neocortical reinstatement is typically observed from approx. 500 msec post-cue onwards (e.g., Yaffe et al., 2014, PNAS; Staresina et al., 2019, NatComm). There are very few studies showing memory reinstatement in earlier time windows, and early effects are much more likely to reflect the initial processing of the word cue, or maybe early repetition priming or familiarity effects, which is in fact also in line with the word frequency findings reported in the present study (see e.g., Stenberg et al., 2009, JOCN). Indeed, the data shown in Fig. 4 and Fig. 5 (panels C in both figures) suggest that remembered and forgotten items do not differ in power and coherence during the initial word processing window, but do differ in a later time window more likely related to successful reinstatement.

(3) Another point related to the time window of interest. The authors report an analysis testing whether ERS is also found in the DLPFC -> LTC coherence patterns, on top of the power patterns. They do find “reinstatement” of such connectivity patterns for TBF items only, and interpret this as demonstrating that inhibitory top-down control from DLPFC to LTC is incorporated into the memory trace at encoding, and reinstated during retrieval (l. 218-220), and that such reinstatement of top-down control is particularly important for successful forgetting. I find it difficult to see why inhibitory control should be exerted during this early 100-500msec encoding period, when patients have not even been instructed to forget the item. Inhibitory control processes are typically assumed to happen during the cuing period, when patients are instructed to remember or forget the preceding word, and thus need to terminate or suppress the encoding process. In fact, previous studies from this group (Fellner et al., in press, Oehr et al., 2018, CurrBiol), one of them using the same dataset, demonstrate that forgetting-relevant electrophysiological processes happen in this critical cuing period.

(4) There is another reason why it can be doubted that the measured effects reflect memory reinstatement. A wealth of decoding and similarity-based EEG and fMRI studies have shown that memory success (vs failure) is associated with more (vs less) neural reinstatement. It is thus difficult to understand why TBF items that are forgotten should, even in theory, exhibit the same amount of memory reinstatement as TBR remembered items. At minimum, the authors need to make a convincing argument as to why intentional forgetting is different from incidental forgetting.

(5) Most intentional forgetting studies (e.g. Anderson and Green, 2001, Nature; Benoit and Anderson, 2012, Neuron; Gagnepain et al., 2014, PNAS; Oehr et al, 2018, CurrBiol) assume that DLPFC exerts a top-down influence directly onto the hippocampus, and content-specific processing in neocortical areas is only modulated as an indirect result. Evidence for a direct impact of DLPFC on LTC, and its relevance for forgetting, should be presented.

Results:

(6) I find it surprising that no power and connectivity differences exist between to-be-remembered and to-be-forgotten items during the recognition test. If an instruction to forget induces inhibition, and the major neural correlate of such inhibition is an alpha/beta synchronization, one would expect a TBF > TBR power and coherence difference at retrieval, reflecting the presumed reinstatement of the inhibitory portions of the engram. The authors should explain this apparent discrepancy. More generally, it is not easy to understand what might cause the difference in encoding-retrieval similarity between conditions, if there are no differences at encoding (which is expected because all words should be processed the same before the instruction comes on), and no differences during retrieval. Differences between conditions should be present at least on some channels in order to explain the differential correlations. Channel-wise comparisons between TBR and TBF (e.g. cluster- or FDR-corrected) should thus be reported instead of the channel-averaged comparisons shown in the present figures.

(7) One of the authors' central argument is that recognizing TBR and TBF items depends on reinstatement in different frequency bands, with high gamma frequencies presumably reflecting excitatory connections of an engram that are relevant for remembering TBR items, and alpha/beta frequencies presumably reflecting inhibitory connections of the engram that are relevant for forgetting TBF items. However, no analysis is reported that directly compares ERS between high and low frequencies, or reports the interaction between high/low frequencies and TBR/TBF conditions. Since the authors use normalized Spearman correlation values as dependent measure, these should be directly comparable. As it stands, the evidence is statistically rather weak, based on the absence of a significant low frequency cluster for TBR, and the absence of a high frequency cluster for TBF. A direct comparison is required to support the major claims.

Methods:

(8) The trial number statement is somewhat ambiguous. It should be clarified whether 50 words (25 to-be-remembered, 25 TBF) was the total for a single block, or the total overall per patient. It should also be made explicit how many trials were on average available per condition per patient for the similarity analyses, after artefact rejection.

(9) I could not find information in the methods section regarding what hospital the data was recorded in.

(10) In order for the reader to understand the nature (e.g. average length) of the feature vectors used for the similarity analyses, the method section should state (a) how many electrodes per patient were available in LTC, DLPFC and the other regions of interest; and (b) into how many overlapping or non-overlapping time windows the 100-500ms time windows were binned before vectorization.

(11) Why did the authors choose a frequency band of 10-19Hz for the transfer entropy analysis, if the significant cluster for coherence (which was used for frequency selection according to the results section) was 12-17Hz?

(12) The methods or results section should clarify what time window was used for the transfer entropy analysis (i.e., was it the same 0.1-0.5 sec as for the other analyses).

(13) The results section should report any potential differences in response times between TBR and TBF items. If patients have longer response latencies for forgotten items, reinstatement would likely take longer too, creating a confound for the neural analyses.

Minor:

- The authors should consider reporting d' as a behavioural measure that is corrected for false alarm rates.

- It would be reassuring to see a supplementary figure plotting the raw correlation values that the differential ERS (i.e., within vs between items) is based on. A strong positive within-vs-between difference can arise because the item-specific (within) similarities are very high, or because the non-specific similarities are very low.

- Do the authors have any empirical evidence for the statement that "word frequency maps to an automatic inhibitory signal initiated during encoding" (l.391-2)? If this was the case, high and low frequency words during encoding (and retrieval) should elicit very distinct power signatures in the low frequency bands, and differential connectivity patterns, according to the authors' own logic. Could be tested empirically. Also, did high and low frequency words show different amounts of remembering/forgetting?

Please find below the response letter in which we address all concerns of the reviewers point-by-point. All changes that we made in the manuscript are highlighted in **yellow** and our answers in the response letter are in **red font**.

We would like to point all reviewers to several numerical changes regarding the exact statistical values in the manuscript. During the re-analyses in the revision process, we realized that some numerical results in the ERS analyses changed slightly. This was likely a consequence of the change of the first author's work location and a resulting change in the version and operation systems used for the analyses (windows vs linux and change of matlab and fieldtrip version compatible with the linux cluster infrastructure), as well as a change in the outlier detection that we had implemented halfway through the analyses in order to optimize the robustness of our results (a differential epoching length slightly changed the outliers that were identified). In response to this observation, we decided to run all analyses again and report the new values.

We would like to stress that these numerical changes did not affect any of our conclusions. There were two results where statistical effects crossed the significance threshold (in both directions): First, the main effect of TBR items in the gamma frequency range (comparing correlations between same vs. different items during encoding and retrieval) only reached trend-level significance instead of significance (see Fig. 3). Importantly, however, the memory effect of stronger gamma-band ERS for remembered compared to forgotten TBR items did not change qualitatively.

Second, the post-hoc analyses of ERS in the hippocampus now shows a significant main effect of instruction (stronger ERS for TBF versus TBR items), an effect that was not significant before. As in the original version, no effect of memory was observed in the hippocampus (consistent with previous intracranial EEG studies analyzing effects of item memory on ERS in the hippocampus: Staresina et al., 2016; Pacheco et al., 2019).

For full transparency, these changes are highlighted throughout the revised version of the manuscript.

Reviewer #1 (Remarks to the Author):

In this manuscript, the authors present intriguing findings regarding the mechanisms of motivated forgetting. Using ECoG, they show that the lateral temporal cortex (LTC) shows similar frequency responses to word onsets in a directed forgetting task: response pattern similarity for to-be-forgotten (TBF) words peak at the alpha-beta bands; and for to-be-remembered (TBR) words at the gamma band. Similar results are shown for forgotten vs remembered words and when considering DLPFC-LTC coherence. Using transfer entropy they show that the direction of forget-related inhibition is from DLPFC to LTC.

The presented results are intriguing and quite surprising, given the fact that the encoding epoch used for the encoding-retrieval similarity (ERS) analysis occurred before words were designated as TBF or TBR. After addressing some possible confounding factors and conducting some additional analysis, I believe this manuscript would be suitable for publication in this journal.

Major issues:

1.

A major challenge the authors face is explaining how neural traces at word onset serve to predict its subsequent association with an experimental condition that has yet to be assigned. Unfortunately, I'm not sure they do a very convincing job in this respect. If I understand correctly, their model suggests that the excitation-inhibition balance - both within LTC and between DLPFC and LTC - always includes the infrastructure for suppression, conveyed by the inhibitory circuits. When an item is intentionally suppressed, according to Fig 1c, the inhibitory synapses overcome the excitatory ones, breaking the balance; whereas successful retrieval involves the excitatory circuits prevailing. There are a number of issues with this hypothesis that need to be addressed. First, it is unclear why retrieval shouldn't simply involve a full recapitulation of the original memory trace, including both excitatory and inhibitory (balanced) nodes. Second, the literature on excitation-inhibition balance in neural circuits suggests it is maintained over time, and breaking the balance is thought to relate to pathology (e.g., Markicevic et al., 2020; Dehghani et al., 2016). If the authors indeed suggest there is some function to breaking the balance, they should state so and link their hypothesis to the existing literature. Third, it makes little intuitive sense that cross-cortical inhibitory circuits are pre-embedded in every memory engram. Previous studies, cited in the manuscript, show that DLPFC is active in memory suppression, but to the best of my knowledge all refer to the inhibition-period itself, not the pre-inhibition encoding period (e.g., Anderson et al., 2004). I wish to stress that I do not discourage the authors from presenting new ideas or challenging the literature, nor am I claiming that the authors are wrong. On the contrary - I believe their results are challenging, and therefore require a more thorough literature review and an in-depth comparison with current ideas and previous findings. The current version of the manuscript avoids the complexity involved with the notion that memories are pre-wired to be forgotten, which is, to my understanding, a major point of the paper.

RESPONSE: We very much appreciate the Reviewer's positive overall assessment of our results and their insightful comments. We fully agree that our finding of "pre-wired" inhibition in stimulus-specific memory traces at encoding is surprising and counter-intuitive at first sight; we also agree that our interpretation of these results may not have been sufficiently thorough and nuanced. We would like to split up our responses to these

points into two sections addressing separately the question of excitation-inhibition imbalance and the possible functional relevance of pre-wiring of cross-cortical inhibition.

1) Does selective reinstatement of inhibitory circuits break the excitation-inhibition balance?

We believe that some of the confusion may have arisen because we did not sufficiently emphasize the level of brain organization that we could measure, so that our interpretations seemed to equate synaptic inhibition with functional (network-level) inhibition. Obviously, we did not measure the strength of synaptic connections but the amplitude of mesoscopic iEEG oscillations. The vast majority of such oscillations depend on a complex interplay of excitation – that may be either provided by exogenous inputs to a certain brain region or generated by intrinsic pacemaker cells in this region – and inhibition, which is required for the refinement of this excitation to specific windows of opportunities, or “duty cycles” (e.g., (Jensen, Bonnefond, & VanRullen, 2012)). Therefore, our results have no direct implications on the inhibition-excitation balance of *synaptic connections* referred to by the papers that are mentioned by the Reviewer (also see comment 1 of Reviewer 3).

Indeed, we fully agree with the Reviewer that dysregulation of the homeostatic balance between excitation and inhibition at the cellular level constitutes a major neuropathological mechanism in various diseases including neurodevelopmental disorders (Markicevic et al., 2020), epilepsy (Dehghani et al., 2016), and Alzheimer’s disease (Palop & Mucke, 2016). It is very unlikely that voluntary forgetting exerts similarly pronounced effects on the balance between excitation and inhibition, in particular because it is considered a beneficial cognitive process that may actually promote resilience to psychological trauma (Engen & Anderson, 2018; Mary et al., 2020). Instead, we refer to *functional* inhibition, which has been repeatedly associated with alpha oscillations (Jensen et al., 2012; Klimesch, 1999). For example, distractor suppression during selective attention to one visual hemifield is associated with increased posterior alpha oscillations in areas that process the distractors (e.g., (Sauseng et al., 2009)). This effect seems to be controlled by the prefrontal cortex (e.g., (Sauseng, Freunberger, Feldheim, & Hummel, 2011)). By contrast, gamma band activity is typically interpreted as reflecting functional activation of a certain area (e.g., (Bauer, Oostenveld, Peeters, & Fries, 2006; Lachaux, Axmacher, Mormann, Halgren, & Crone, 2012; Ray, Niebur, Hsiao, Sinai, & Crone, 2008)). Importantly, both alpha oscillations and gamma band activity arise from a complex interplay between excitatory and inhibitory neurons (e.g., (Buzsaki, 2006; Gips, van der Eerden, & Jensen, 2016; Nimmrich, Draguhn, & Axmacher, 2015)).

During voluntary memory suppression, several previous studies reported alpha oscillations in both sensory areas (Fellner, Waldhauser, & Axmacher, 2020) and in the hippocampus (Oehrn et al., 2018); again, this effect is controlled by prefrontal cortex, which shows pronounced increases of gamma band activity (Oehrn et al., 2018). It is exactly this sense of *functional* inhibition or suppression related to alpha oscillations, and of *functional* activation related to gamma band activity, that we refer to in our interpretation of our novel findings.

Various previous studies, also indicated by the Reviewer, used the ratio between low (including alpha) and high (gamma) frequencies as a measure of the functional excitation-inhibition (E:I) balance. Since this ratio is reflected by the slope of a power spectrum in

log-log space, larger slopes indicate relatively more functional inhibition (Dehghani et al., 2016; Gao, Peterson, & Voytek, 2017; Markicevic et al., 2020). Importantly, we did not find any evidence that directed forgetting influences the E:I ratio, as overall power did not differ between the TBF and TBR conditions in the time window of the ERS effects. Instead, we found *item-specific* changes in the alpha/beta and gamma patterns of activity.

Therefore, our results provide evidence for a selective strengthening of functional inhibition in item-specific activity patterns, but this effect does not change the overall E:I balance. We changed the manuscript to clarify these important issues. First, to avoid any confusion regarding functional versus synaptic inhibition, we have changed Figure 1. Here is the new version:

Response figure 1: Reinstatement of memory traces via encoding-retrieval similarity (Identical to the new figure 1 in the manuscript).

Second, we clearly emphasize in the text that we did not measure the strength or sign of synaptic connections. (line 75-77): *“Note that these inhibitory processes concern functional inhibition, which likely comprises inhibitory as well as excitatory synaptic connections [17, 18]”*.

Third, we emphasize this point explicitly in the discussion. This now reads (lines 430-439): *“Importantly, we did not find evidence for an overall difference in either alpha/beta or gamma power in the time window of the reinstatement. Such an overall power change might have been an indication that the functional excitation-inhibition balance has changed [45], an effect that has been associated with major neuropathological mechanisms in various diseases [46,47]. Instead, we found item-specific changes in the alpha/beta and gamma patterns of activity. Therefore, our results provide evidence of an instruction- and memory-dependent rewiring of functional connectivity patterns. This is exactly what would be expected from an adaptive memory mechanism whose function it is to selectively incorporate forgetting or remembering attempts into item-specific memory traces, while maintaining the homeostatic balance between excitation and inhibition.”*

2) Why would cross-cortical inhibition be pre-wired at encoding?

It is indeed a complex and important question why functional inhibition should already be present prior to the instruction to forget, an effect which our data clearly show. While our interpretation of the functional relevance of this inhibitory “pre-wiring” has to remain somewhat speculative at the current stage, we could think of two reasons.

Firstly, inhibition during encoding may reflect an automatic feedback signal created by the DLPFC to control sensory activations in the LTC in order to avoid exacerbated unspecific activation after every sensory input – i.e., to increase the tuning of network responses (Miller & Wang, 2006; Yi, Huang, Simon, & Doyle, 2000). Active forgetting mechanisms could latch onto this inhibitory signal to actively inhibit a memory trace in a stimulus-dependent manner. As described in the first part of the response, this would not imply an overall change in the task-unrelated excitation-to-inhibition balance, but a selective and item-specific strengthening of functional inhibition.

Alternatively – or in addition – it could be that this effect is specific (or particularly pronounced) in the current task in which participants are aware that half of the items need to be forgotten. Thereby, a strategy would be to incorporate a functional inhibitory trace at encoding and after the cue decide which trace to keep (HajiHosseini & Holroyd, 2015; Spitzer & Haegens, 2017).

The current study cannot disambiguate these two options. In the future, a list-method directed forgetting paradigm may provide further information. In this paradigm, participants first learn a list of items (e.g., words) and only afterwards receive the (unexpected) instruction to forget these items (Geiselman, Bjork, & Fishman, 1983). This is followed by a second list of items and then a retrieval test of all items. In this paradigm, one could again test whether encoding-retrieval similarities of items from the to-be-forgotten list relies specifically on alpha/beta frequencies, which would speak in favor of a more general and automatic feedback signal.

In the revised manuscript we now discuss these options (line 475-490): *“As participants during encoding are unaware of the instruction to forget a particular item, it is striking that TBF items are associated with signatures of item-specific reinstatement that are classically associated with inhibitory feedback. This means that item-specific patterns of functional inhibition and of inhibitory connectivity are already present when the items are initially presented, i.e. prior to the instruction to forget them. It could be that this effect reflects an automatic feedback signal created by DLPFC to increase the specificity of sensory activation in the LTC [56,57]. Active forgetting mechanisms could latch onto this signal to actively inhibit a memory trace in a stimulus-dependent manner. Alternatively – or in addition – it could be that this effect is specific (or particularly pronounced) in the current task in which participants are aware that half of the items need to be forgotten. Thereby, a strategy would be to incorporate a functional inhibitory trace at encoding and after the cue decide which trace to keep [58,59]. In the future, employing a list-method directed forgetting paradigm [60] may allow disambiguating these options. If encoding-retrieval similarities of items from the TBF list again rely specifically on alpha/beta frequencies, this would speak in favor of a more general and automatic feedback signal. “*

The manuscript is missing crucial information regarding the response period, which may have substantial effects on the main results. It is unclear whether the participants had to state if the word is old/new as fast as possible and how their response was conveyed (vocally?). This is important, because it may mean that the "retrieval" period was actually contaminated by responses, and different response patterns between TBR/TBF and remembered/forgotten words may have been at the core of ERS results. For example, the difference in response times for different conditions (e.g., slower for new vs old) may have affected the correlation patterns. The authors deal with some of these effects by presenting and comparing time-frequency patterns (e.g., Fig3C), but there may be more nuanced differences that do not result in statistical significance. This is especially true given the small effect sizes that are reported ($|\rho| \leq 0.02$). The authors should address this possibility with more detailed methods, RT data, and additional analyses.

RESPONSE: We agree with the Reviewer to elaborate more on the response period, and regret not having been more specific in the method section regarding the instructions. In the experiment, the participants had to indicate their response via button presses. They did not receive any instruction to respond as fast as possible, but responded at their own pace. We added this information to the method section (see page 6 and page 28).

To test whether differences in reaction times may have influenced the results, we conducted various additional analyses. First, we calculated for the TBR and TBF conditions the median reaction times during the retrieval phase separately for remembered and forgotten items, and compared them in a 2*2 RM ANOVA with factors "condition" (TBR/TBF) and "memory" (remembered or forgotten). Results showed overall faster responses for remembered compared to forgotten words ($F(1,15) = 8.362, p = 0.0118$; Response Figure 2) and faster responses for the TBR compared to the TBF condition ($F(1,15) = 5.212, p = 0.037$). The interaction between memory and condition did not reach significance, but showed a trend ($F(1,15) = 3.48, p = 0.082$). The simple effects analysis comparing TBR and TBF separately for remembered and forgotten words showed that there was only a difference for the remembered ($t(15) = 2.66, p = 0.036$), but not the forgotten words ($t(15) = -0.115, p = 0.98$).

Response Figure 2. Reaction time effects (identical to Supplementary Figure 2a).

This means that overall incorrect responses (i.e., responses in forgotten trials) are given more slowly than correct responses. This effect is not surprising and has been repeatedly observed before and may reflect some hesitation prior to the incorrect response or memory scanning mechanisms (Sternberg, 1969; Theios, 1973).

Furthermore, this pattern of results suggests that reaction times do not influence the significant alpha/beta ERS, because (1) the trend significant interaction suggests that RT differences between remembered and forgotten TBR words are larger than between remembered and forgotten TBF words, while alpha/beta ERS only differs between remembered and forgotten TBF words; and (2) RTs differ only between remembered TBR vs. TBF words and not between forgotten TBR vs. TBF words, while alpha/beta ERS only differs between forgotten TBR vs. TBF words and not between remembered TBR vs. TBF words. This new result is described in lines 221-223: *“Additionally, the ERS for the TBF forgotten words was higher than for the TBR forgotten words in the beta range (17-20 Hz; cluster statistics = -9.58; p = 0.038; max t(15)-value = -2.66).”*

Nonetheless, to further investigate a possible influence of reaction times on ERS results, we correlated trial-by-trial reaction times to the strength of the correlation between the encoding and retrieval pair. This was done using Spearman correlations for each frequency point that was used in the original analysis (separately for each condition). In other words, we investigated whether the ERS effect itself was related to reaction times. The correlation showed no clusters even at an uncorrected level (Response Figure 3). This again indicates that reaction times do not influence ERS effects.

Response Figure 3. Trial-by-trial correlations between reaction times and ERS values.

Finally, we calculated if the response preparation (locked to the button press) had an effect on the ERS at the significant frequency ranges of our main effects (for low frequencies: 17-22 Hz; for high frequencies: 105-110 Hz). We thus re-calculated ERS, but instead of using the stimulus-locked retrieval data, we used response-locked data. As the exact time of a possible effect is unknown, we repeated the analysis at various encoding and response-locked time periods (always using windows of 400ms, as in our original analysis). We did not find any significant clusters for this response-locked ERS analysis (see Response Figure 4). Also, no clusters were found when the same analysis was conducted for memory specific effects (i.e., remembered vs. forgotten TBR items; remembered vs. forgotten TBF items).

In the new version of the manuscript, we included the reaction time analyses and added the respective figure to the supplementary figures (see page 7 and new supplementary figure 2).

Response figure 4. Response locked ERS analysis. ERS analyses using various encoding times and response times. A) Analyses performed at 17-22 Hz. B) Analyses performed at 105-110 Hz.

3.

The authors emphasize that their data provide evidence that intentional forgetting does not erode the memory trace, but rather suppresses it, leaving it intact (e.g., line 24). This is based on the finding that there is some ERS for forgotten memories. However, this does not show that the memory trace is intact. It still might be substantially - but not entirely - eroded. I suspect the authors' claim is somewhat of a strawman argument since very few people (if any) would suggest the neural trace is reduced to zero. Taken together, with the small effect size of the ERS analysis, I suggest using more nuanced language. If the authors choose to leave this claim in the manuscript, I think it would be helpful if they explicitly state how their data would have looked like have the trace been erased (i.e., present their H0 directly). This may make this point more data-based and easier to comprehend.

RESPONSE: Thanks for bringing this up. We never intended to state that our data show the memory trace is entirely intact for either TBR or TBF items. Indeed, as pointed out by the Reviewer, only some features of the original item-specific representation built during encoding are reinstated during retrieval, as indicated by the numerically low (though still stimulus-specific) correlations between encoding and retrieval. Moreover, these features differ between TBR and TBF items, as indicated by the differences in frequency content of the encoding-retrieval similarities in the two conditions.

We carefully went through the manuscript to use more nuanced language regarding this point. Here are some phrases that we changed:

New text in lines 24-26: *“Contrariwise, using intracranial EEG recordings from lateral temporal cortex (LTC), we find that memory traces for actively forgotten information are partially preserved and exhibit unique neural signatures.”* (before: *“Contrariwise, we find intact memory traces of actively forgotten information in intracranial EEG recordings from lateral temporal cortex (LTC) that support successful forgetting.”*)

New text in lines 398-400: *“Intentional forgetting does not completely erase memory traces, but forms item-specific representations that specifically rely on patterns classically associated with inhibitory control.”* (before: *“Intentional forgetting does not erase memory traces, but forms item-specific representations that specifically rely on patterns of inhibitory control [16].”*)

New text in lines 102-104: *“Our results demonstrate that successful intentional forgetting is due to a selective strengthening of item-specific top-down connections, rather than a mere degradation of the memory traces.”* (before: *“Our results demonstrate that successful intentional forgetting is due to a selective strengthening of item-specific inhibitory top-down connections, rather than an erasure of the inhibited memory traces.”*).

We would like to stress that all of our conclusions refer to the fact that different (frequency) features of the memory traces are reactivated at different strengths for TBR vs. TBF items, and as a function of memory. This more nuanced description of the overall strength of the memory trace is thus in line with our conclusions.

4.

A related point concerns the lack of a real control condition. Comparing TBF and TBR one to the other leaves many questions open. For example, do TBF instructions reduce ERS values at high frequencies (Fig 3b) when compared to no instructions at all? Do TBR instructions increase them or leave them unchanged? I'm not suggesting running more experiments, but rather addressing this limitation and changing the language accordingly. This relates to the previous point as well, since it could be that TBF ERS would have always been lower than a non-included control (e.g., that control words would lie consistently at $\rho=0.02$ in Fig 3b), indicating that TBF low-frequency ERS is lower than control (and TBR representation even lower). TBF>TBR does not imply TBF>control (and same for TBR, of course).

RESPONSE: We agree that the lack of a control condition is a clear limitation of our study. However, it is in no way specific to this particular study, but an inherent limitation of the item-method directed forgetting paradigm (and other forgetting paradigms such as the list-method directed forgetting paradigm in which an entire set of items is instructed to be either remembered or forgotten after its encoding). Therefore, all our conclusions relate to items that are instructed to be remembered vs. those that are instructed to be forgotten. This is the main contrast of most forgetting paradigms and potentially a limitation of the whole field. The think/no-think paradigm (Anderson & Green, 2001) differs in this respect, but does not investigate forgetting during encoding but during

retrieval. We would like to point out that using a condition without any instruction would have been problematic as well, since the participant's strategy in this case lacks experimental control. We also agree with the possible interpretation of the ERS results suggested by the Reviewer and now explicitly mention this as an option.

The respective section now reads (line 491-496): *“Directed forgetting paradigms have the intrinsic limitation that it is not possible to compare reinstatement to a control condition, which may be considered a problem for most studies on voluntary memory suppression during encoding (Bjork, 1970). With respect to our results, it may be that the ERS effects in the alpha/beta frequency range that we observed for TBF items could still be weakly present also in a control condition without an explicit instruction.”*

5.

Many of the issues mentioned up until this point concern the authors' choice to analyze the word-onset periods during encoding and retrieval. An analysis of the post-cue epoch may add some information regarding the neural processes. For example, does alpha-beta power following the TBF cue add information regarding the memory fate of the presented word? Does it mediate the ERS effect? How do DLPFC-LTC interactions look like during this time period and how do they correspond to the coherence-based ERS?

RESPONSE: Thank you for this interesting suggestion. We analyzed activity during the cue period as proposed. First, we analyzed the main effect of instruction (TBR vs TBF), the main effect of memory (new vs old items) and their interaction on power values in LTC and DLPFC, and on the coherence between these two areas.

In the LTC, we found a main effect of instruction (6-30 Hz; 0.35-0.85 sec; $p = 0.013$; Response Figure 5; identical to main Figure 7) with a cluster showing relatively higher alpha/beta power after a TBF cue compared to a TBR cue (less pronounced reductions as compared to baseline, i.e. reduced desynchronization) as well as an interaction effect (2-11 Hz; 0.75-1.5 sec; $p = 0.009$). This interaction was a consequence of stronger theta/alpha activity in the remembered compared to forgotten condition for the TBF items (2-13 Hz; 0.8-1.5 sec; $p=0.004$), but not the TBR items ($p > 0.05$).

In the DLPFC, we found more pronounced increases of theta/alpha activity for TBF compared to TBR cues (4-14 Hz; 0.5-0.75 sec; $p = 0.046$; also see Oehrn et al., 2018). Furthermore, we observed larger DLPFC gamma power increases for forgotten as compared to remembered words (55-160 Hz; 0.95-1.15 sec; $p = 0.022$). We did not find any interaction.

For coherence, we repeated the calculations as performed for the ERS analysis, but using a longer encoding time window (coherence was calculated per trial over the 0-1sec window for each frequency point separately), and then averaged over all trials and channel pairs. Here, we found again a main effect of instruction (24-27 Hz; $p = 0.04$) showing stronger DLPFC-LTC coherence for the TBF compared to the TBR condition, but did not observe a memory or interaction effect.

Response Figure 5. Cue effect (identical to figure 7 in main manuscript). A) Instruction effect on power in LTC. B) Power for TBF items in LTC. C) Power for TBR items in LTC. D) Memory effect on power in DLPFC. E) Instruction effect on power in LTC. F) Instruction effect on DLPFC-LTC coherence. Shaded error bars indicate the standard error of the mean. White outlines and black bar indicate significance at the 0.05 level (cluster corrected).

We followed up investigating whether these cue effects had a direct relationship to the ERS effects. We performed the following analyses:

- 1) We analyzed intra-individual correlations between single-trial ERS effects and single-trial cue power effects. The previously calculated single-trial ERS values were correlated across trials with the trial-based power/coherence values at the average power/coherence of the significant time/frequency cluster of the cue effects (separately for the LTC instruction effect, the LTC TBF memory effect, the DLPFC instruction effect, and the coherence instruction effect). A cluster-based permutation test was performed to investigate if there were frequency clusters that showed significant correlations. We found no significant effects (all $p > 0.05$; Response Figure 6, identical to supplementary figure 12). When we conducted the same analysis only for TBF trials or even only for successfully forgotten TBF trials, we again did not find any significant relationship either ($p > 0.05$).

Response Figure 6, identical to Supplementary Figure 12. Intra-individual correlations between single trial cue power/coherence and single trial ERS values. Correlations are estimated at the time/frequency clusters of the respective cue effects (see figure 7) and using either all trials, only TBF trials, or only successfully forgotten TBF trials.

- 2) We assessed possible inter-individual correlations between the ERS effects and cue power effects. For each cue effect reported in figure 7 we calculated per subject the effect size for that respective effect averaged over the frequency/time data in the cluster (again separately for the LTC instruction effect, the LTC TBF memory effect, the DLPFC instruction effect, and the coherence instruction effect). For example, for the LTC instruction effect we subtracted the average TBF power from the average TBR power in the interval between 6-30 Hz; 0.35-0.85 sec. We then calculated per frequency bin used in the original analyses the corresponding ERS effect (again for the LTC instruction effect subtracting the TBF ERS values from the TBR ERS values). Then we correlated the ERS effects with the cue power/coherence effects for each ERS frequency bin separately. Again, we corrected for multiple comparisons using cluster-based statistics.

For the LTC instruction and LTC TBF memory effect we found no correlations. Interestingly, however, for the DLPFC instruction effect we found a significant correlation between the ERS effect and the cue power effect ($p = 0.036$; 16-19 Hz; max $t(10)$ -value = 4.01; Response Figure 7). A similar correlation was found for the DLPFC-LTC coherence ($p = 0.022$, 15-17 Hz; max $t(9)$ -value = 4.93). This indicates that participants showing more pronounced DLPFC alpha/beta power increases, and DLPFC-LTC alpha/beta coherence, also showed a more prominent reinstatement of stimulus-specific alpha/beta LTC encoding patterns for TBF vs. TBR items.

Response Figure 7, identical to Figure 8 Inter-individual correlations across participants between the cue instruction effects and the ERS instruction effect for (A) DLPFC power and (B) DLPFC-LTC coherence. Correlations are estimated at the time/frequency ranges of the cue effects (see figure 7). Black bar indicates significance at the $p = 0.05$ levels. Scatterplots indicate the effect at the peak of the significant power/coherence effect. Circles indicate individual participants. Inset indicates the Spearman correlation value.

Thus, we could show that there are instruction effects during the cue period on both DLPFC and LTC power and DLPFC-LTC coherence, as previously reported (Benoit and Anderson, 2012; Depue et al., 2007; Oehrns et al., 2018). Cue effects in DLPFC and on DLPFC-LTC coherence were related across subjects to ERS effects in LTC and DLPFC-LTC, respectively. We now report on these new interesting findings in the results and the discussion.

The manuscript now reads in the results (lines 329-378):

“Finally, we evaluated time-frequency patterns in DLPFC and LTC during the time period of the cue presentation in order to investigate their possible relationship to ERS effects. In the LTC, we found a main effect of instruction (6-30 Hz; 0.35-0.85 sec; $p = 0.013$; max $t(16)$ -value = 4.636; Figure 7A) with a cluster showing relatively higher alpha/beta power after a TBF cue compared to a TBR cue (less pronounced reductions as compared to baseline, i.e. reduced desynchronization) as well as an interaction effect (2-11 Hz; 0.75-1.5 sec; $p = 0.009$; max $t(16)$ -value = 3.89). This interaction was a consequence of stronger theta/alpha activity in the remembered compared to forgotten condition for TBF items (2-13 Hz; 0.8-1.5 sec; $p = 0.004$; max $t(16)$ -value = 5.35; Figure 7B), but not for TBR items ($p > 0.05$; Figure 7C). In the DLPFC, we observed larger DLPFC gamma power increases for forgotten as compared to remembered words (55-160 Hz; 0.95-1.15 sec; $p = 0.022$; min $t(15)$ -value = -4.35; Figure 7D). Additionally, we found more pronounced increases of theta/alpha activity for TBF compared to TBR cues (4-14 Hz; 0.5-0.75 sec; $p = 0.046$; max $t(15)$ -value = 4.67; Figure 7E; also see [22]). We did not find any interaction.

We also analyzed cue effects on coherence. We repeated the coherence calculations as performed for the ERS analysis, but using a longer encoding time window (coherence was calculated per trial over the 0-1 second window for each frequency point separately), and then averaged over all trials and channel pairs. Here, we found a main effect of instruction (24-27 Hz; $p = 0.04$; min $t(9)$ -value = -4.13; Figure 7F) showing stronger DLPFC-LTC coherence for the TBF compared to the TBR condition, but did not observe a memory or interaction effect.

We assessed possible inter-individual correlations between the ERS and the cue power effects. For each cue effect reported in Figure 7, we calculated per subject the strength of the respective contrast, averaged over the frequency/time data in the cluster (separately for the LTC instruction effect, the LTC TBF memory effect, the DLPFC instruction effect, and the coherence instruction effect). For example, for the LTC instruction effect, we subtracted the average TBF power from the average TBR power in the interval between 6-30 Hz; 0.35-0.85 sec. We then calculated per frequency bin used in the original ERS analyses the corresponding ERS effect (again for the LTC instruction effect subtracting the TBF ERS values from the TBR ERS values). Then we correlated the ERS effects with the cue power/coherence effects, separately for each ERS frequency bin. Again, we corrected for multiple comparisons using cluster-based statistics. For the LTC instruction and LTC TBF memory effect, we found no correlations. Interestingly, however, for the instruction effect on DLPFC power, we found a significant positive correlation between the ERS effect and the cue effect ($p = 0.036$; 16-19 Hz; max $t(10)$ -value = 4.01; Figure 8). A similar correlation was found for the DLPFC-LTC coherence ($p = 0.022$, 15-17 Hz; max $t(9)$ -value = 4.93). Investigating intra-individual correlations between cue power and ERS size across trials did not result in any significant effect (Supplementary Figure 12). This indicates that participants showing more pronounced DLPFC alpha/beta power increases, and DLPFC-LTC alpha/beta coherence, also showed a more prominent reinstatement of stimulus-specific alpha/beta LTC encoding patterns for TBF vs. TBR items."

We also elaborate on this point in the discussion (line 464-472): "Most previous studies investigating directed forgetting have focused on the cue-period window [14,15,22]. Our analysis of activity during the cue-period confirms that during the cue, DLPFC exhibits increased activity for the forgetting instruction (Figure 7E). Moreover, our results show the effects of forgetting instructions in the LTC (Figure 7A-C) and also indicate that DLPFC-LTC coherence is stronger for forgetting than remembering cues. Finally, we could show that these cue-period effects, previously associated with inhibition [15], are correlated to the strength of the ERS effects across subjects. This further strengthens our interpretation that directed forgetting depends on a selective strengthening of inhibitory patterns present during encoding."

6.

The RSA analyses seem to diverge from the manuscript's narrative and I am not sure they substantially add to it. As a reader, I felt that this section somewhat threw me off balance and disrupted the natural flow of the manuscript, and I would consider removing it to improve the paper. If the authors choose to leave it in, however, I would suggest avoiding the use of reverse inference in the discussion (i.e., "area X is involved in Y, so process Z that also happens in area X must also concern Y"). There is no real

evidence showing that any of the word-related metrics measured have anything to do with memory suppression.

RESPONSE: We agree that this analysis diverges from the main story. We have therefore removed it from the manuscript.

Minor issues:

7.

Line 338 (and perhaps elsewhere) - the claim that higher ERS reflects "strengthening of inhibitory circuits" seems to be somewhat exaggerated. It may suggest that the same inhibitory circuits are active during encoding and retrieval, but that doesn't necessarily mean that these circuits were strengthened.

RESPONSE: We agree and have nuanced our language regarding this point. In the sentence mentioned by the Reviewer, we now write: "*[...], but rather a selective reinstatement of a unique set of inhibitory connections*"

8.

Line 125 - "TBF words were remembered better than chance". In fact, this is not a comparison with chance level (which is 50%), but with the false alarm level. Please rephrase.

RESPONSE: We have rephrased the text and now write "*...the false alarm rates were lower than the rates of correctly memorized TBF items*".

9.

The term "interaction" is used throughout the manuscript to describe "difference". Interaction suggests that there are two independent factors that jointly affect some measure. No interaction analysis is conducted in this paper, so the term may be misleading. Even in cases where there are two independent variables (e.g., Fig 2, TBR/F x Hz), the used method is not one that may reveal interactions and the term interaction, when used, actually means "difference".

RESPONSE: All our analyses of condition effects on item-specific reinstatement concern a difference between differences (e.g. TBF same item - TBF different item vs TBR same item - TBR different item). Therefore, we had initially proposed the term "interaction" (see supplementary figure 5 for the raw traces, showing that all comparison refer to differences of differences, i.e. two factors: same or different item and condition). Nonetheless, we understand that we never look at the simple effects of the interaction and throughout the manuscript treat the interaction as a single measure. We therefore

went through the manuscript and refrain from using the word interaction when comparing only two ERS conditions. Also all figures now refer to the term „difference“ instead of „interaction“.

10.

Please explain why different time-frequency methods were used for <30 Hz and >30 Hz frequencies. Would using the same methods change the results?

RESPONSE: Multi-tapering provides good control over frequency smoothing as is often desirable when one wants to improve the signal to noise ratio (at a cost of spectral resolution). Gamma responses are often very broadband and do not require high spectral resolution. In addition, they often have lower signal-to-noise ratios and therefore benefit from multitapering. For low frequencies, signal-to-noise is better and losing frequency resolution is more problematic. This is described for example in some articles on multi-tapering (Mitra & Pesaran, 1999; Percival & Walden, 1993). For this reason, we chose multi-tapering for the high frequencies and wavelets for the low frequencies. A similar procedure was used in several other publications (e.g., (Mikulan et al., 2018; Shah et al., 2019). In general, it is thus most likely that using the same time-frequency transformation method for both low and high frequencies produces similar results, but with a reduced SNR and less optimal resolution.

11.

Notes regarding the figures:

- a. Fig1 - I suggest using "unsuccessful encoding" instead of the confusing term "unsuccessful memory" (which could be interpreted as a memory for a bad event).
- b. Fig2c (and SuppF1) - each dot represented an electrode, not a patient. The legend is somewhat misleading.
- c. Fig2a bottom - please indicate when the participant response was allowed.
- d. Fig 5 - please indicate what the timeframe used for this analysis was.
- e. Fig 5b - the lines are overlapping and hard to see. Perhaps reduce the line width.
- f. Fig 5c - please mention that this panel reflects time-frequency responses in DLPFC.
- g. SuppF5 - The word "brackets" should be replaced with "parentheses". Also, it would be helpful if the tested frequencies were mentioned in each subplot.

RESPONSE: We thank the Reviewer for the points regarding the figures. Figure 1 was fully adapted and the text has also been changed. All other figures have been adapted according to the suggestions of the Reviewer.

Regarding figure 5b: We decided to remove this part of the figure as the coherences were so close that only the difference wave provided useful visualizations.

Reviewer #2 (Remarks to the Author):

This paper describes electrophysiological activity in the lateral temporal and prefrontal cortex using a directed forgetting paradigm. Overall, the authors do an admirable job of walking through the implications of a series of complex analyses. It is interesting and novel to look at this paradigm in the setting of intracranial recordings.

Power extraction follows reasonable methods. Entropy for directional connections is not commonly done, but seems reasonable overall and the authors provide sufficient motivation for its use.

1.

In terms of the chosen contrasts, I would have assumed that one would compare forgotten TBF items to forgotten TBR items (to identify a signal related to intentional rather than unintentional forgetting) and recalled TBF to recalled TBR items (to understand how reinstatement differs when someone intends to forget it vs not). In Fig 4, it looks like these contrasts would show significant differences in line with author interpretation, but I don't see this as a direct contrast in the manuscript.

RESPONSE: We appreciate the positive assessment of our procedures, and agree that this contrast is relevant. We now analyzed the contrast between intentionally forgotten TBF items and incidentally forgotten TBR items. Indeed, we observed a significant cluster showing stronger ERS for the TBF forgotten compared to the TBR forgotten items ($p = 0.038$, frequency range = 17-20 Hz). This comparison was added to the manuscript (page 11) and has been added for comparison in the revised Figure 4 (please see Response Figure 8 below).

Response Figure 8 (identical to Figure 4Aiii). Encoding-retrieval similarity for the direct contrast between intentionally vs. incidentally forgotten items (TBF forgotten vs. TBR forgotten).

For the remembered items, we did not observe a significant difference between remembered TBR and remembered TBF items (lowest p-value for any cluster = 0.254).

This result indicates that the forgetting effect in the alpha/beta frequency range is indeed highly specific to TBF items, since it does not occur similarly for TBR items. The remembering effect in the gamma frequency range is less specific, though; it only reflects

a simple dissociation (significant effect for TBR items but not for TBF items) rather than a clear double dissociation.

We have added a discussion of this point and now write “*When we directly compared ERS of intentionally vs. incidentally forgotten items (i.e., of forgotten TBF vs. forgotten TBR items), we found clear evidence for more pronounced reinstatement of item-specific patterns in the alpha/beta frequency range for forgotten TBF items. This indeed shows that ERS in this frequency range is highly specific for intentionally forgotten items. The direct contrast between remembered TBR and remembered TBF items was not significant, suggesting that reinstatement of gamma-band activity patterns may also support memory for TBF items to a certain extent (even though it does not reach significance).*” (line 410-417).

2.

Along these lines, a signal that is truly characteristic of intentional forgetting should 1) be different than unintentional forgetting (see above) and 2) different than the signal for novel items. I understand that this would require looking at similarity between novel items and TBF items as a class (rather than item by item), but it would be reassuring that the TBF reinstatement does not reflect some non specific (less interesting) response to novel items.

RESPONSE: We thank the Reviewer for their suggestion. We addressed this point by comparing the similarities between TBF/TBR items during encoding with either other TBF/TBR items during retrieval or with novel items (excluding same-item encoding-retrieval pairs), i.e. same-class ERS versus novel ERS. This was also done separately for the remembered and forgotten TBF items.

We found that for both TBR and TBF items, correlations were significantly *lower* within their respective class (e.g., ERS between TBR items during encoding and other TBR items during retrieval) than they were with novel items (e.g., ERS between TBR items during encoding and novel items during retrieval; see Response Figure 9 below; TBR: 120-140 Hz; $p=0.029$; min $t(15)$ -value = -2.21; TBF cluster 1: 90-110 Hz; $p = 0.020$; min $t(15)$ -value = -1.87; TBF cluster 2: 40-50 Hz; $p=0.047$; min $t(15)$ -value = -2.17). There was no interaction between instruction and encoding class (same class or novel items) – i.e., the increases in ERS to novel items were similarly pronounced when compared to previously presented TBR and TBF items from the same class.

This indicates that activity patterns during encoding of items are overall more similar to activity patterns during presentation of novel items than to those of previously presented items. Since novel items have not been seen before, a neural signal to a novel item is similar to an ‘encoding’ signal. This is in contrast with a memory retrieval signal, which contains the information that an item has been presented before.

Next, we split up the analysis for remembered and forgotten TBF items. For remembered TBF items, we again found reduced between-item ERS values as compared to novel items in a high frequency window (120-135 Hz; $p = 0.033$; min $t(15)$ -value = -2.45). Interestingly, when we compared between-item ERS for forgotten TBF items to

novel items, we found a reduction in the beta frequency range, i.e. in a similar frequency window where the stimulus-specific reinstatement was found (19-22 Hz; $p = 0.033$; $\min t(15)\text{-value} = -3.85$). There was no significant difference between forgotten compared to remembered TBF items ($p = 0.105$). These results show that (item-unspecific) ‘encoding signals’ of later forgotten TBF items are more similar to activity during presentation of novel items as compared to ‘retrieval signals’ of other forgotten TBF items. More importantly, they show that activity during retrieval of forgotten TBF items differs from an overall ‘novelty’ signal.

Response Figure 9 (identical to Supplementary figure 6). Comparing off-diagonal ERS with novel items. A) Activity during encoding of an item was correlated with activity either during retrieval of a different item from the same class (TBR or TBF), or during presentation of a novel item. These correlations were statistically compared, separately for TBR and TBF items. B) Identical to A) but separately for remembered and forgotten TBF items. Blue, orange, and black bars indicate significant item-specific reinstatement of remembered items, forgotten items, and their difference, respectively. Shaded error bars indicate the standard error of the mean.

We would like to note that this new analysis differs fundamentally from our main analyses of stimulus-specific reinstatement that are related to the difference between correlations of same items vs. different items (off-diagonal). For completeness, we here present traces of the correlations for the original effects with both the same-item and between-item correlations. It is evident that all effects are driven by a difference of the same-item correlations, not of the off-diagonal/class-specific correlations (Response Figure 10; identical to Supplementary Figure 5). We can therefore exclude that the item-specific ERS

effects are driven by reductions in the class-specific correlations which one might infer from the results shown in Response Figure 9. Note that all comparisons are performed within-instruction and memory condition (e.g. same-item forgotten TBF ERS values are only compared with different-item forgotten TBF ERS values). As such, the main effects cannot be explained by any item non-specific (e.g. response to a novel item) effects.

In sum, our collective results of stronger class-specific ERS for novel compared to same-class retrieval items *and* of higher same-item ERS compared to same class-different item ERS indicate that responses to successfully forgotten TBF items are different from novel items and that the item-specific TBF reinstatement reflects more than a non-specific response to novel items. We have added both the same-class versus novel items ERS analysis as well as the raw traces from Response Figure 10 to the supplementary materials (see supplementary figure 5 and 6).

Response Figure 10, identical to Supplementary figure 5: Raw correlation traces of the encoding-retrieval similarity analysis of same vs. different items. Significance lines reflecting the ERS comparisons (on-diagonal vs. off-diagonal) are identical to those shown in the main manuscript. Effects are driven by increases in same item correlations rather than by differences in the different-item correlations. Conventions are the same as in Figure 3B.

3.

More generally, I am having trouble understanding how to interpret reinstatement differences in the paradigm. It appears to me as though the participants are cued regarding the TBR/TBF classification for the word after the item is shown. This would mean that at the time of item presentation, the individual does not know if they were supposed to be remembered or not. That would lead me to assume that TBF items should exhibit lower reinstatement, because the representation of the item at retrieval is now different than it was at initial encoding, as the subject would still have been trying to remember the item at the time the encoding-related brain activity was first recorded. The authors examine the issue of post-presentation modification of a memory trace, such that reinstatement in the TBF condition is evidence of strengthened inhibitory connections. Reinstatement of this connection implies that it is present at encoding (prior to the forget signal), but then is normally pruned/eliminated during purposeful memory such that it is not reinstated? To phrase another way, the concept of an inhibitory top down connection facilitated by the DLPFC and characterized by alpha/beta activity seems reasonable, but greater reinstatement in this connection would imply that the connection exists during encoding (before the TBF/TBR cue) and then is preserved at retrieval uniquely in the TBF condition. This seems far less intuitive than the addition of some inhibitory signal

that occurs in response to the TBF/TBR cue. Would the authors predict that the TBR/TBF cue would elicit the alpha/beta signal they observed being reinstated? Could this period of time be analyzed to reinforce the conclusions regarding the inhibitory model postulated by authors?

RESPONSE: We thank the Reviewer for this very insightful comment. Indeed, we completely agree with the interpretation of the Reviewer that our results indicate that the inhibitory signal which is reinstated for TBF (but not TBR) items is already present during encoding, rather than being added (as a kind of inhibitory “tag”) during the cue period: such an addition would modify the previous encoding signal and reduce encoding-retrieval similarities for TBF items, which is not what we see. We also agree that this result is somewhat surprising at first sight, but consider it one of the most interesting novel findings in our study. In order to further corroborate this point, we added a novel analysis of activity during the cue period.

In detail, we split our response to this comment into two parts. In the first part, we describe our analysis of activity during the cue period, which was also requested by Reviewer 1 (their comment #5). In the second part, we then provide a more detailed interpretation of our finding that the inhibitory LTC activity and DLPFC-LTC connectivity exists already during encoding prior to the instruction to forget (also see comment #1 of Reviewer 1). For both parts we reiterated (part of) the responses to Reviewer 1.

Activity during the cue period

We analyzed activity during the cue period as proposed. First, we analyzed the main effect of instruction (TBR vs TBF), the main effect of memory (new vs old items) and their interaction on power values in LTC and DLPFC, and on the coherence between these two areas.

In the LTC, we found a main effect of instruction (6-30 Hz; 0.35-0.85 sec; $p = 0.013$; Response Figure 11; identical to main Figure 7) with a cluster showing relatively higher alpha/beta power after a TBF cue compared to a TBR cue (less pronounced reductions as compared to baseline, i.e. reduced desynchronization) as well as an interaction effect (2-11 Hz; 0.75-1.5 sec; $p = 0.009$). This interaction was a consequence of stronger theta/alpha activity in the remembered compared to forgotten condition for the TBF items (2-13 Hz; 0.8-1.5 sec; $p=0.004$), but not the TBR items ($p > 0.05$).

In the DLPFC, we found more pronounced increases of theta/alpha activity for TBF compared to TBR cues (4-14 Hz; 0.5-0.75 sec; $p = 0.046$; also see Oehrns et al., 2018). Furthermore, we observed larger DLPFC gamma power increases for forgotten as compared to remembered words (55-160 Hz; 0.95-1.15 sec; $p = 0.022$). We did not find any interaction.

For coherence, we repeated the calculations as performed for the ERS analysis, but using a longer encoding time window (coherence was calculated per trial over the 0-1sec window for each frequency point separately), and then averaged over all trials and channel pairs. Here, we found again a main effect of instruction (24-27 Hz; $p = 0.04$) showing stronger DLPFC-LTC coherence for the TBF compared to the TBR condition, but did not observe a memory or interaction effect.

Response Figure 11. Cue effect (identical to figure 7 in main manuscript). A) Instruction effect on power in LTC. B) Power for TBF items in LTC. C) Power for TBR items in LTC. D) Memory effect on power in DLPFC. E) Instruction effect on power in LTC. F) Instruction effect on DLPFC-LTC coherence. Shaded error bars indicate the standard error of the mean. White outlines and black bar indicate significance at the 0.05 level (cluster corrected).

We followed up investigating whether these cue effects had a direct relationship to the ERS effects. We performed the following analyses:

- 3) We analyzed intra-individual correlations between single-trial ERS effects and single-trial cue power effects. The previously calculated single-trial ERS values were correlated across trials with the trial-based power/coherence values at the average power/coherence of the significant time/frequency cluster of the cue effects (separately for the LTC instruction effect, the LTC TBF memory effect, the DLPFC instruction effect, and the coherence instruction effect). A cluster-based permutation test was performed to investigate if there were frequency clusters that showed significant correlations. We found no significant effects (all $p > 0.05$; Response Figure 11, identical to supplementary figure 12). When we conducted the same analysis only for TBF trials or even only for successfully forgotten TBF trials, we again did not find any significant relationship either ($p > 0.05$).

Response Figure 12, identical to Supplementary Figure 12. Intra-individual correlations between single trial cue power/coherence and single trial ERS values. Correlations are estimated at the time/frequency clusters of the respective cue effects (see figure 7) and using either all trials, only TBF trials, or only successfully forgotten TBF trials.

- 4) We assessed possible inter-individual correlations between the ERS effects and cue power effects. For each cue effect reported in figure 7 we calculated per subject the effect size for that respective effect averaged over the frequency/time data in the cluster (again separately for the LTC instruction effect, the LTC TBF memory effect, the DLPFC instruction effect, and the coherence instruction effect). For example, for the LTC instruction effect we subtracted the average TBF power from the average TBR power in the interval between 6-30 Hz; 0.35-0.85 sec. We then calculated per frequency bin used in the original analyses the corresponding ERS effect (again for the LTC instruction effect subtracting the TBF ERS values from the TBR ERS values). Then we correlated the ERS effects with the cue power/coherence effects for each ERS frequency bin separately. Again, we corrected for multiple comparisons using cluster-based statistics.

For the LTC instruction and LTC TBF memory effect we found no correlations. Interestingly, however, for the DLPFC instruction effect we found a significant correlation between the ERS effect and the cue power effect ($p = 0.036$; 16-19 Hz; max $t(10)$ -value = 4.01; Response Figure 12). A similar correlation was found for the DLPFC-LTC coherence ($p = 0.022$, 15-17 Hz; max $t(9)$ -value = 4.93). This indicates that participants showing more pronounced DLPFC alpha/beta power increases, and DLPFC-LTC alpha/beta coherence, also showed a more prominent reinstatement of stimulus-specific alpha/beta LTC encoding patterns for TBF vs. TBR items.

Response Figure 13, identical to Figure 8 Inter-individual correlations across participants between the cue instruction effects and the ERS instruction effect for (A) DLPFC power and (B) DLPFC-LTC coherence. Correlations are estimated at the time/frequency ranges of the cue effects (see figure 7). Black bar indicates significance at the $p = 0.05$ levels. Scatterplots indicate the effect at the peak of the significant power/coherence effect. Circles indicate individual participants. Inset indicates the Spearman correlation value.

Thus, we could show that there are instruction effects during the cue period on both DLPFC and LTC power and DLPFC-LTC coherence, as previously reported (Benoit and Anderson, 2012; Depue et al., 2007; Oehrns et al., 2018). Cue effects in DLPFC and on DLPFC-LTC coherence were related across subjects to ERS effects in LTC and DLPFC-LTC, respectively. We now report on these new interesting findings in the results and the discussion.

The manuscript now reads in the results (lines 329-378):

“Finally, we evaluated time-frequency patterns in DLPFC and LTC during the time period of the cue presentation in order to investigate their possible relationship to ERS effects. In the LTC, we found a main effect of instruction (6-30 Hz; 0.35-0.85 sec; $p = 0.013$; max $t(16)$ -value = 4.636; Figure 7A) with a cluster showing relatively higher alpha/beta power after a TBF cue compared to a TBR cue (less pronounced reductions as compared to baseline, i.e. reduced desynchronization) as well as an interaction effect (2-11 Hz; 0.75-1.5 sec; $p = 0.009$; max $t(16)$ -value = 3.89). This interaction was a consequence of stronger theta/alpha activity in the remembered compared to forgotten condition for TBF items (2-13 Hz; 0.8-1.5 sec; $p = 0.004$; max $t(16)$ -value = 5.35; Figure 7B), but not for TBR items ($p > 0.05$; Figure 7C). In the DLPFC, we observed larger DLPFC gamma power increases for forgotten as compared to remembered words (55-160 Hz; 0.95-1.15 sec; $p = 0.022$; min $t(15)$ -value = -4.35; Figure 7D). Additionally, we found more pronounced increases of theta/alpha activity for TBF compared to TBR cues (4-14 Hz; 0.5-0.75 sec; $p = 0.046$; max $t(15)$ -value = 4.67; Figure 7E; also see [22]). We did not find any interaction.

We also analyzed cue effects on coherence. We repeated the coherence calculations as performed for the ERS analysis, but using a longer encoding time window (coherence was calculated per trial over the 0-1 second window for each frequency point separately), and then averaged over all trials and channel pairs. Here, we found a main effect of instruction (24-27 Hz; $p = 0.04$; min $t(9)$ -value = -4.13; Figure 7F) showing stronger DLPFC-LTC coherence for the TBF compared to the TBR condition, but did not observe a memory or interaction effect.

We assessed possible inter-individual correlations between the ERS and the cue power effects. For each cue effect reported in Figure 7, we calculated per subject the strength of the respective contrast, averaged over the frequency/time data in the cluster (separately for the LTC instruction effect, the LTC TBF memory effect, the DLPFC instruction effect, and the coherence instruction effect). For example, for the LTC instruction effect, we subtracted the average TBF power from the average TBR power in the interval between 6-30 Hz; 0.35-0.85 sec. We then calculated per frequency bin used in the original ERS analyses the corresponding ERS effect (again for the LTC instruction effect subtracting the TBF ERS values from the TBR ERS values). Then we correlated the ERS effects with the cue power/coherence effects, separately for each ERS frequency bin. Again, we corrected for multiple comparisons using cluster-based statistics. For the LTC instruction and LTC TBF memory effect, we found no correlations. Interestingly, however, for the instruction effect on DLPFC power, we found a significant positive correlation between the ERS effect and the cue effect ($p = 0.036$; 16-19 Hz; max $t(10)$ -value = 4.01; Figure 8). A similar correlation was found for the DLPFC-LTC coherence ($p = 0.022$, 15-17 Hz; max $t(9)$ -value = 4.93). Investigating intra-individual correlations between cue power and ERS size across trials did not result in any significant effect (Supplementary Figure 12). This indicates that participants showing more pronounced DLPFC alpha/beta power increases, and DLPFC-LTC alpha/beta coherence, also showed a more prominent reinstatement of stimulus-specific alpha/beta LTC encoding patterns for TBF vs. TBR items."

We also elaborate on this point in the discussion (line 464-472): "Most previous studies investigating directed forgetting have focused on the cue-period window [14,15,22]. Our analysis of activity during the cue-period confirms that during the cue, DLPFC exhibits increased activity for the forgetting instruction (Figure 7E). Moreover, our results show the effects of forgetting instructions in the LTC (Figure 7A-C) and also indicate that DLPFC-LTC coherence is stronger for forgetting than remembering cues. Finally, we could show that these cue-period effects, previously associated with inhibition [15], are correlated to the strength of the ERS effects across subjects. This further strengthens our interpretation that directed forgetting depends on a selective strengthening of inhibitory patterns present during encoding."

Existing connections during encoding

We fully agree that our finding of item-specific inhibitory features already during encoding is surprising and counter-intuitive at first sight; we also believe that our interpretation of these results may not have been sufficiently thorough and nuanced. The existence of LTC-DLPFC connections could have two different causes. Firstly, inhibition during encoding may reflect an automatic feedback signal created by the DLPFC to control sensory activations in the LTC in order to avoid exacerbated unspecific activation after every sensory input – i.e., to increase the tuning of network responses. Active forgetting mechanisms could latch onto this inhibitory signal to actively inhibit a memory trace in a stimulus-dependent manner. As described in the first part of the response, this would not imply an overall change in the excitation-to-inhibition balance, but a selective and item-specific strengthening of functional inhibition.

Alternatively – or in addition – it could be that this effect is specific (or particularly pronounced) in the current task in which participants are aware that half of the items need to be forgotten. Thereby, a strategy would be to incorporate a functional inhibitory trace at encoding and after the cue decide which trace to keep.

The current study cannot disambiguate these two options. In the future, a list-method directed forgetting paradigm may provide further information. In this paradigm, participants first learn a list of items (e.g., words) and only afterwards receive the (unexpected) instruction to forget these items (Geiselman et al., 1983). This is followed by a second list of items and then a retrieval test of all items. In this paradigm, one could again test whether encoding-retrieval similarities of items from the to-be-forgotten list relies specifically on alpha/beta frequencies, which would speak in favor of a more general and automatic feedback signal.

In the revised manuscript we now discuss these options (line 475-490): “As participants during encoding are unaware of the instruction to forget a particular item, it is striking that TBF items are associated with signatures of item-specific reinstatement that are classically associated with inhibitory feedback. This means that item-specific patterns of functional inhibition and of inhibitory connectivity are already present when the items are initially presented, i.e. prior to the instruction to forget them. It could be that this effect reflects an automatic feedback signal created by DLPFC to increase the specificity of sensory activation in the LTC [56,57]. Active forgetting mechanisms could latch onto this signal to actively inhibit a memory trace in a stimulus-dependent manner. Alternatively – or in addition – it could be that this effect is specific (or particularly pronounced) in the current task in which participants are aware that half of the items need to be forgotten. Thereby, a strategy would be to incorporate a functional inhibitory trace at encoding and after the cue decide which trace to keep [58,59]. In the future, employing a list-method directed forgetting paradigm [60] may allow disambiguating these options. If encoding-retrieval similarities of items from the TBF list again rely specifically on alpha/beta frequencies, this would speak in favor of a more general and automatic feedback signal. “

4.

An alternative interpretation is that the frontal cortex is providing context information rather than inhibitory control. This would be the case if TBF “forgotten” items were actually remembered, but the individual also retrieved the TBF cue association for the item and then reported it as novel/not

remembered based on this associative information. In this interpretation, greater reinstatement in alpha/beta for TBF is not the failed pruning of an inhibitory connection, but the preservation of associative information. The authors may mention this possibility in the Discussion, or specifically why they reject this idea (maybe something in the behavioral data that I missed)?

RESPONSE: This is an interesting suggestion. However, if the specific signatures during reinstatement of TBF items were due to their context (i.e., the fact that they were instructed to be forgotten), then this context should be the same for all TBF items (or for all successfully forgotten TBF items, respectively). This cannot explain our results on item-specific reinstatement in the individual conditions, though: For these analyses, we contrasted trials from the same condition, e.g., comparing activity during presentation of forgotten TBF items during encoding with either activity during presentation of the same forgotten TBF items during retrieval or different forgotten TBF items during retrieval. Since all items from this condition share the same context, item-specific differences in their reinstatement patterns cannot be explained by this context.

In other analyses, we did compare reinstatement of TBF items with reinstatement of TBR items, though, and in these cases, the shared “forgetting” context of TBF items may in principle drive a particularly prominent reinstatement. However, this is unlikely for two reasons. First, our new analyses (shown in response to comment #2) show that encoding-retrieval similarities are actually *lower* than correlations between encoding of an item and presentation of a novel item during retrieval. This is the opposite of what one would expect if encoding-retrieval similarities were due to a shared forgetting context of all TBF items. Second, we found that the condition differences in reinstatement of TBF and TBR items were actually not present when only the between-item correlations were considered that all shared the same context (see dotted lines in Response Figure 10), but were specific to the same-item correlations (solid lines in Response Figure 10).

One may still argue that it is not shared activity during encoding of TBF items but during the cue period which is reinstated as a context of all TBF items. We thus conducted a further control analysis in which we correlated activity during the forgetting instruction of one item with activity during the presentation of a different TBF item during retrieval. The same was done for activity during remember instructions and presentation of different TBR items during retrieval, and correlations were then compared between conditions. This analysis of instruction-specific cue-retrieval similarity is relevant for the question of context reinstatement, because the forgetting context is shared among all TBF items, but does not occur for the TBR items. We found no difference between same- or different-instruction cue-retrieval similarity (Response Figure 14).

Response figure 14. Identical to Supplementary figure 7. Instruction-specific cue-retrieval similarity. Instruction-specific cue-retrieval similarity. We correlated activity during the forgetting instruction of one item with activity during the presentation of a different TBF item during retrieval. The same was done for activity during remembering instructions and presentation of different TBR items during retrieval, and correlations were then compared between conditions. This analysis of instruction-specific cue-retrieval similarity is relevant for the question of context reinstatement, because the forgetting context is shared among all TBF items, but does not occur for the TBR items. We found no difference between same- or different-instruction cue-retrieval similarity. The figure displays the difference between same-instruction cue-retrieval similarity and different-instruction cue-retrieval similarity (legend indicates the instruction at the cue). Shaded error bars indicate the standard error of the mean.

5.

My prior assumption is that a retrieval/memory control signal (post retrieval monitoring) is more likely to come from the RIGHT DLPFC, but I don't know if the authors specify the hemisphere for recordings or break down by hemisphere in the analysis. Similarly - is the non-dominant lateral temporal cortex associated with semantic representations? Are the electrodes implanted in a mix of dominant/non-dominant hemispheres?

RESPONSE: In our original analyses we collapsed over hemispheres. This was done since we only have a limited number of patients and splitting would significantly reduce our statistical power. Nonetheless, we now analyzed possible hemispheric differences by contrasting results in the two hemispheres using independent samples t-tests (note that this is the most conservative option as there are some patients with electrodes in both hemispheres). For the LTC analyses there were 4 patients with electrodes in the left hemisphere only, 3 with electrodes in the right hemisphere only and 9 with electrodes in both hemispheres. For the DLPFC-LTC coherence and transfer entropy analyses this was 4, 4, and 2, for left, right, and bilateral, respectively (note that in this analysis only hemispheres were included with both DLPFC and LTC electrodes).

For the low-frequency effects (at the frequencies of either the TBF vs TBR contrast or of the TBF remembered vs. TBF forgotten contrast), we found no evidence for lateralization (all $p > 0.05$). For the high-frequency effect (estimated at the frequency of the TBR remembered vs. TBR forgotten contrast), we found a trend for stronger reinstatement in the left compared to the right LTC for items that were successfully remembered (t-test: $t(23) = 1.98$, $p = 0.059$). This trend became significant in a paired t-test when considering only participants with channels in both hemispheres ($n=9$ participants; $t(8) = 4.79$, $p = 0.001$). When we compared this trend significant effect at frequency ranges of the high frequency cluster in the main analysis (figure 3 comparing TBF vs TBR), we did not find any difference ($p > 0.05$).

This suggests that the left, dominant LTC is associated with intentional remembering of words (all patients were right handed). However, hemispheric differences between left and right LTC do not seem to play a specific role for intentional forgetting of these items.

Notably, for DLPFC-LTC coherence or transfer entropy, we found no statistically significant lateralization effects ($p > 0.05$). Of course, this can still be due to the relatively small numbers of participants that were included for this comparison.

The hemisphere-specific analysis was added to the manuscript (page 16 and Supplementary Figure 11).

Response Figure 15. Identical to Supplementary figure 11. Hemisphere -specific effects in the time window from 0.1-0.5s. All comparisons are based on the same frequency ranges as reported in Supplementary Figure 9. T indicates trend significant effect ($p < 0.1$)

6.

My opinion is that the analysis on page 15 raises more questions than it answers. If there is an inhibitory signal in the alpha beta range, but this is also related to word frequency, it is hard to fit everything together. For the issue of controlled forgetting, I agree it is reassuring that gamma reinstatement is unique for recalled items (not just activity related to reading words), but I'm not sure a link to semantic association is especially helpful for the main analysis.

RESPONSE: We realized this analysis is indeed somewhat unrelated to the main analyses (also see comment 6 of Reviewer 1). We have removed this section from the manuscript.

7.

Methodologically, the reinstatement analysis seems to assume that encoding/retrieval similarity occurs at the same time points, ie no shifting of encoding relative to retrieval is performed, as in the Yaffe paper they cite. This paper shows that retrieval-related activity occurs relatively earlier in time.

The authors re calculate their similarity values after changing their window, but this is not the same thing as shifting encoding relative to retrieval.

RESPONSE: We agree this is a relevant additional analysis. We re-ran the ERS analysis for the frequency window of interest (at the significant cluster for the relevant condition contrast). Then we shifted the retrieval and the encoding time windows to create a similar profile across various windows as in the Yaffe paper. Please note that time point 0 encompasses activity from -200ms until +200ms (as all our analyses integrated 400 millisecond of data).

Response Figure 16. Identical to Supplementary figure 3B. Time-shifting ERS analyses using various encoding and retrieval time windows (in the frequency range of the significant low-frequency difference between TBR and TBF items).

Response Figure 17. Identical to Figure 4C. Time-shifting ERS analyses showed similar patterns as the main analyses.

The results of these analyses show that for most of the individual conditions, the effects of interest were most pronounced on the diagonal (TBF same vs. different items, Response Figure 16, middle; TBR same vs. different remembered items, Response Figure 17, left; TBF same vs. different forgotten items, Response Figure 17, middle).

Some contrasts appeared to be more pronounced during earlier retrieval than encoding time windows (TBR same vs. different items, Response Figure 16, left; TBR vs. TBF, Response Figure 16, right; TBF remembered vs. forgotten items, Response Figure 17, right). These results are similar to the findings in the Yaffe paper, which makes sense because they used a paradigm in which participants actively tried to remember items, as in our “TBR” condition.

Interestingly, the contrast of remembered vs. forgotten TBR items showed two retrieval time clusters of the same encoding time cluster, very similar to previous results in (Zhang, Fell, & Axmacher, 2018) and in (Pacheco-Estefan et al., 2019).

We would like to point out some details in the Yaffe task which may further explain why this study showed very consistently effects in earlier retrieval as compared to encoding time windows. Specifically, during encoding two words were presented, while during retrieval only one word was shown. In our experiment, the presentation during encoding and retrieval was identical, making non-time-shifted effects more likely.

We added the time-shifting ERS analysis to the manuscript for a better evaluation of the temporal evolution of reinstatement effects.

8.

Can the authors report the degrees of freedom in their statistical tests (pre clustering)? Did the authors shuffle trial labels as part of clustering shuffle? This is often appropriate in iEEG data. Reporting the initial test statistic is also useful along with the clustering based p values. Also, how many individual connections were included in the analysis in Fig 5? The authors are aggregating temporal cortex recordings over a large area, so in subjects with frontal electrodes, I can imagine that hundreds of connections are potentially being averaged together?

RESPONSE: We apologize for the lack of information here. The updated manuscript contains the degrees of freedom pre-clustering as well as the values of the original test statistics. For all LTC analyses, the degrees of freedom is 15 (n=16 patients); for the DLPFC-LTC interaction analyses, this is 9 (n=10 patients). In our current analyses, we shuffled only the average condition labels, not the trial labels themselves. While we agree that it is often appropriate to also shuffle at the trial level, we have not done this in the current study because various analyses involve interaction contrasts (e.g., same vs. different TBR items compared to same vs. different TBF items), in which trial-label shuffling is conceptually complicated: On one hand, one could argue that the basic contrasts (same vs. different items) have to be computed first and the interaction contrasts should only be run on the resulting differences, in order to make the interactions clearly interpretable. In this case, the shuffling should only been done e.g. between TBR and TBF items, after computing the first basic contrast. On the other hand, this would imply that the different effects are treated differently, which is conceptually undesirable as well (because it may result in two different interaction results). In that case, one would argue that shuffling should be done already at the level of the basic contrast (e.g., same vs. different). In previous studies, we compared shuffling at the trial label and shuffling of condition averages and yielded very similar results (e.g., (Zhang et al., 2015)).

Indeed, the number of DLPFC-LTC connections is large, ranging from 66-777 (mean of 332 connections). This information is provided in Supplementary Figure 6B. Since we did not have any strong a priori hypotheses on specific subregions in DLPFC and LTC driving the item-specific representations, averaging all connections was the most parsimonious and data-driven approach.

9.

Finally, the authors mention that they Bonferroni corrected results across 5 temporal cortical regions when trying to pin down anatomical specificity for the effects. Did they also correct across the other brain regions they examined for their LTC results (shown in supplemental material)?

RESPONSE: Our a priori hypotheses regarded the LTC, based on previous findings about item-specific representations in iEEG data (e.g., (Jang, Wittig Jr, Inati, & Zaghoul, 2017; Pacheco-Estefan et al., 2019)). The other brain regions were only performed post-hoc and for completeness. We therefore did not correct across these regions. We clarified this in the manuscript (page 6 and page 9).

10.

I think this paradigm is a good one to link to iEEG. The Discussion and interpretation go a bit further than the data would seem to justify (or is needed to amplify the novelty of results), especially the paragraph starting on line 377 which is overly speculative. The paragraph starting line 404 even more so. I am not sure the connections identified by the authors can clearly be called inhibitory, other than to say that previous studies report that connections in this freq range exhibit inhibitory properties in other paradigms and other behaviors.

RESPONSE: We thank the Reviewer for this comment. We removed all sections related to the representational similarity analyses and the concept of different representational formats. This includes part of the paragraph that started previously in line 377. We removed the paragraph starting at line 404 altogether. Moreover, we are now more careful regarding the interpretation of our data related to inhibition, and refer in this regard to the previous literature that found activity at alpha/beta frequency and DLPFC top-down influences to be relevant for *functional* inhibition.

Reviewer #3 (Remarks to the Author):

This manuscript reports a study investigating the electrophysiological signatures of forgetting, using human intracranial recordings from the lateral temporal and frontal cortices of epilepsy patients. Neural patterns are measured while patients encode single words, and are later asked to distinguish between previously seen and unseen words in a recognition memory test. Critically, the encoding of each word is followed by an instruction to either remember this word (to-be-remembered, TBR), or to intentionally forget the word (to-be-forgotten, TBF). Neural patterns from encoding and recognition are then compared by means of encoding-retrieval similarity (ERS), a measure typically used to indicate memory reinstatement. The central finding is that both TBR and TBF words show significant ERS, suggesting reinstatement of the memory traces originally formed during encoding. Critically, this similarity is found in high gamma frequencies for TBR words, and in lower alpha/beta frequencies for TBF words. This tendency is present for similarity based on spectral power, as well as similarity based on cross-electrode connectivity patterns. In terms of connectivity, the data additionally suggest that TBF words that were actually forgotten show reinstatement of low-frequency connectivity from frontal (DLPFC) to lateral temporal cortex. Apart from some statistical weaknesses, the study is solid in terms of data analysis and reporting, and my concerns are primarily of conceptual nature.

The authors interpret their findings as showing that intentional forgetting depends on the strengthening of “inhibitory engrams”, supposedly reflected in alpha/beta encoding patterns that are then reinstated for TBF (but not TBR) items. Although the manuscript is well-written, and the story is intriguing, the data do not seem to provide any direct evidence for inhibitory engrams and their strengthening. There is thus a large gap between the actual data and their interpretation, and conclusions like the “Our results demonstrate that successful intentional forgetting is due to a selective strengthening of item-specific inhibitory top-down connections, rather than an erasure of the inhibited memory traces” seem overstated.

Conceptual concerns regarding the interpretation of the results:

1.

The interpretation of alpha/beta patterns as inhibitory engrams strongly relies on the assumption that alpha/beta power is a direct and specific index of inhibitory connectivity within a neural assembly that forms the engram. The authors cite the alpha inhibition hypothesis (Klimesch et al, 2007) and the inhibitory gating hypothesis (Jensen & Mazaheri, 2010) as evidence for such a mapping. As far as I am aware, neither of these ideas assumes that alpha/beta power in any way reflect inhibitory connections within a neural circuit or assembly. If anything, slice preparations and modelling work show that gamma oscillations are strongly related to inhibitory interneurons. Having said that, I doubt that any robust evidence exists to map an oscillation in a limited frequency band to excitatory connections within a neural circuit, and an oscillation in a different frequency band to inhibitory connections. The authors need to make a very strong case to justify this central assumption.

RESPONSE: We apologize for the confusion here. Indeed, our description and in particular our previous version of Figure 1 was misleading because it seemed to refer to inhibitory

synaptic connections and inhibitory interneurons and link them exclusively to alpha oscillations. Obviously, intracranial EEG recordings cannot measure activity at this microscopic level of brain organization. We also fully agree with the Reviewer’s statement that the vast majority of oscillations, and certainly those in the alpha and gamma frequency range, critically depend on inhibitory interneurons. Our notions of “excitation” and “inhibition” were supposed to refer to *functional* inhibition – in a similar manner as (Klimesch, Sauseng, & Hanslmayr, 2007) and (Jensen & Mazaheri, 2010) refer to the role of alpha oscillations, that is, as functionally inhibiting neurocognitive processing in specific brain areas. In this framework, coherence in the alpha frequency range is supposed to reflect inhibitory control – typically exerted by prefrontal cortex (e.g., (Sauseng et al., 2011)) – over sensory processing areas such as, in our case, the lateral temporal cortex.

We regret this misunderstanding came up, and blame it mostly to our visualization in Figure 1 and our introduction of inhibition. Therefore, we have changed the introduction and Figure 1 to make this point clearer (see e.g. page 4 that now reads: “*Note that these inhibitory processes concern functional inhibition, which likely comprises inhibitory as well as excitatory synaptic connections [17,18]. Indeed, intentional forgetting is accompanied by increases in alpha/beta (8-25Hz) oscillations – a signature of active functional inhibition [19,20] – in both DLPFC [21] and hippocampus [22].*”

We updated figure 1 as follows:

Response figure 18: Reinstatement of memory traces via encoding-retrieval similarity (Identical to figure 1).

2.

Second, there is a mismatch between this interpretation and the measurements used to support it. All encoding-retrieval similarity results in this study come from relatively early time windows of stimulus processing, 100-500msec after the onset of the words during encoding and recognition. What makes the authors think that reinstatement in LTC happens as early as 100sec after the onset of a recognition cue? Neocortical reinstatement is typically observed from approx. 500 msec post-cue onwards (e.g., Yaffe et al., 2014, PNAS; Staresina et al., 2019, NatComm). There are very few studies showing memory

reinstatement in earlier time windows, and early effects are much more likely to reflect the initial processing of the word cue, or maybe early repetition priming or familiarity effects, which is in fact also in line with the word frequency findings reported in the present study (see e.g., Stenberg et al., 2009, JOCN). Indeed, the data shown in Fig. 4 and Fig. 5 (panels C in both figures) suggest that remembered and forgotten items do not differ in power and coherence during the initial word processing window, but do differ in a later time window more likely related to successful reinstatement.

RESPONSE: Thanks for raising this important point. First of all, we would like to emphasize that all our analyses involved time windows of 400ms – i.e., any effects in the 100-500ms temporal cluster do not need to start at 100ms in order to yield significance.

With respect to previous studies, we would like to point out that the encoding time periods from which stimulus-specific activity was found to be reinstated in the Yaffe et al. (Yaffe et al., 2014) study were frequency specific and seemed to start very early after cue onset for high gamma activity (see their Figure 4B, right), which would be consistent with our gamma-band findings for intentionally remembered items (the Yaffe study did not report any reinstatement in the alpha and beta frequency range, possibly because it did not involve any intentional forgetting). In our own work, we found reinstatement of encoding activity that occurred in two different clusters, with time windows starting as early as 100ms after cue onset ((Zhang et al., 2018); see Figure 2A). Thus, it seems that the time window during encoding that contains stimulus-specific information is relatively well matched to these studies which used very similar features (i.e., distributed power values across electrodes in time windows of several hundreds of milliseconds).

With regard to reinstatement of stimulus-specific information during *retrieval*, this was analyzed locked to vocalizations in the study by Yaffe et al. which used a cued recall paradigm. In our own study that used a recognition memory test as the current study (Zhang et al., 2018), we observed reinstatement during a time window starting 100ms after the onset of the retrieval cue, again in close correspondence to our current findings.

The study by Staresina et al., (Staresina et al., 2019) differed because (1) it used distributed patterns of single units (rather than iEEG activity), and (2) these recordings were from very circumscribed areas in the medial temporal lobe, i.e. at the end of the ventral visual processing stream. By contrast, the studies by Yaffe et al. (2014), Zhang et al. (2018) and the current study all included activity from earlier sensory processing stages (putatively with shorter latencies).

In the revised manuscript, we discuss these results, and their relationship to previous studies, more explicitly. We now write (lines 504-516): *“The time windows containing item-specific information during encoding and showing reinstatement of that information during retrieval started relatively early, i.e. 100ms after stimulus onset during both encoding and retrieval. Notably, all our analyses involved time windows of 400ms during, i.e. effects in the 100-500ms temporal cluster do not need to start at 100ms in order to yield significance. Furthermore, this timing may reflect processing in all areas along the ventral visual stream in which electrodes were implanted (see Figure 2C). It is in line with findings in previous studies on reinstatement of high gamma-band activity [31] or broad-band activity [42] from relatively early encoding time periods. This seems to contrast with results from human single-unit recordings in the medial temporal lobe showing reinstatement of substantially later activities [9]. Finally, participants in our study were*

only asked to give an old/new response during the recognition memory test, and therefore responses may rely on both, recollection or familiarity. This may again explain the relatively early timing of these effects.”

Notably, we did not aim to differentiate between different cognitive processes that may drive recognition memory such as recollection vs. familiarity – participants just gave an old/new response, which may rely on either process. This is now emphasized more clearly when we write (lines 513-516): “Finally, participants in our study were asked to give an old/new response during the recognition memory test, and therefore responses may rely on both, recollection or familiarity. This may again explain the relatively early timing of reinstatement effects.”

To obtain a more comprehensive understanding of the temporal evolution of our effects, we performed a time-shifting ERS analyses (centered at the frequency of the significant results) in which we shifted the time window of both the encoding and retrieval times. These results are shown below and included in the revised manuscript in supplementary Figure 3B and new Figure 4C.

Response Figure 19. Identical to Supplementary figure 3B. Time-shifting ERS analyses using various encoding and retrieval time windows (in the frequency range of the significant low-frequency difference between TBR and TBF items).

Response Figure 20. Identical to Figure 4C. Time-shifting ERS analyses showed similar patterns as the main analyses.

The results of these analyses show that for most of the individual conditions, the effects of interest were most pronounced on the diagonal (TBF same vs. different items, Response Figure 19, middle; TBR same vs. different remembered items, Response Figure 20, left; TBF same vs. different forgotten items, Response Figure 20, middle). Some contrasts appeared to be more pronounced during earlier retrieval than encoding time windows (TBR same vs. different items, Response Figure 19, left; TBR vs. TBF, Response Figure 19, right; TBF remembered vs. forgotten items, Response Figure 20, right). Interestingly, the contrast of remembered vs. forgotten TBR items showed two retrieval

time clusters of the same encoding time cluster, very similar to previous results in Zhang et al., 2018 and in Pacheco-Estefan et al., 2019.

3.

Another point related to the time window of interest. The authors report an analysis testing whether ERS is also found in the DLPFC -> LTC coherence patterns, on top of the power patterns. They do find “reinstatement” of such connectivity patterns for TBF items only, and interpret this as demonstrating that inhibitory top-down control from DLPFC to LTC is incorporated into the memory trace at encoding, and reinstated during retrieval (l. 218-220), and that such reinstatement of top-down control is particularly important for successful forgetting. I find it difficult to see why inhibitory control should be exerted during this early 100-500msec encoding period, when patients have not even been instructed to forget the item. Inhibitory control processes are typically assumed to happen during the cuing period, when patients are instructed to remember or forget the preceding word, and thus need to terminate or suppress the encoding process. In fact, previous studies from this group (Fellner et al., in press, Oehr et al., 2018, CurrBiol), one of them using the same dataset, demonstrate that forgetting-relevant electrophysiological processes happen in this critical cuing period.

RESPONSE: This is an important point that was also raised by other Reviewers. We agree that active inhibition (or inhibitory control processes) has been shown to occur during the cueing period and is relevant for the subsequent success of intentional forgetting. Indeed, this process of active inhibition should be locked to the forgetting cue (and was, among other studies, investigated by the studies cited by the Reviewer). However, these effects during the cue period do not indicate the consequences of this inhibition process on stimulus-specific memory representations. This is what we aimed at in the current study. In other words, we do not argue that active inhibitory control is exerted during encoding – instead, our results indicate that any stimulus triggers a cascade of processes that are reflected by neural activity in the alpha and gamma frequency range and that can be afterwards selectively strengthened by remembering or forgetting cues.

Since we agree that the interplay between stimulus-specific representations during encoding and inhibitory control processes during the forgetting cues needs to be investigated in greater detail, we conducted an additional analysis of inhibition effects during the cue period. As this comment relates also to comments of previous reviewers we would like to reiterate the answer to point 3 of Reviewer 2.

Cue response analyses

We analyzed activity during the cue period as proposed. First, we analyzed the main effect of instruction (TBR vs TBF), the main effect of memory (new vs old items) and their interaction on power values in LTC and DLPFC, and on the coherence between these two areas.

In the LTC, we found a main effect of instruction (6-30 Hz; 0.35-0.85 sec; $p = 0.013$; Response Figure 21; identical to main Figure 7) with a cluster showing relatively higher alpha/beta power after a TBF cue compared to a TBR cue (less pronounced reductions as

compared to baseline, i.e. reduced desynchronization) as well as an interaction effect (2-11 Hz; 0.75-1.5 sec; $p = 0.009$). This interaction was a consequence of stronger theta/alpha activity in the remembered compared to forgotten condition for the TBF items (2-13 Hz; 0.8-1.5 sec; $p=0.004$), but not the TBR items ($p > 0.05$).

In the DLPFC, we found more pronounced increases of theta/alpha activity for TBF compared to TBR cues (4-14 Hz; 0.5-0.75 sec; $p = 0.046$; also see Oehrns et al., 2018). Furthermore, we observed larger DLPFC gamma power increases for forgotten as compared to remembered words (55-160 Hz; 0.95-1.15 sec; $p = 0.022$). We did not find any interaction.

For coherence, we repeated the calculations as performed for the ERS analysis, but using a longer encoding time window (coherence was calculated per trial over the 0-1sec window for each frequency point separately), and then averaged over all trials and channel pairs. Here, we found again a main effect of instruction (24-27 Hz; $p = 0.04$) showing stronger DLPFC-LTC coherence for the TBF compared to the TBR condition, but did not observe a memory or interaction effect.

Response Figure 21. Cue effect (identical to figure 7 in main manuscript). A) Instruction effect on power in LTC. B) Power for TBF items in LTC. C) Power for TBR items in LTC. D) Memory effect on power in DLPFC. E) Instruction effect on power in LTC. F) Instruction effect on DLPFC-LTC coherence. Shaded error bars indicate the standard error of the

mean. White outlines and black bar indicate significance at the 0.05 level (cluster corrected).

We followed up investigating whether these cue effects had a direct relationship to the ERS effects. We performed the following analyses:

- 5) We analyzed intra-individual correlations between single-trial ERS effects and single-trial cue power effects. The previously calculated single-trial ERS values were correlated across trials with the trial-based power/coherence values at the average power/coherence of the significant time/frequency cluster of the cue effects (separately for the LTC instruction effect, the LTC TBF memory effect, the DLPFC instruction effect, and the coherence instruction effect). A cluster-based permutation test was performed to investigate if there were frequency clusters that showed significant correlations. We found no significant effects (all $p > 0.05$; Response Figure 22, identical to supplementary figure 12). When we conducted the same analysis only for TBF trials or even only for successfully forgotten TBF trials, we again did not find any significant relationship either ($p > 0.05$).

Response Figure 22, identical to Supplementary Figure 12. Intra-individual correlations between single trial cue power/coherence and single trial ERS values. Correlations are estimated at the time/frequency clusters of the respective cue effects (see figure 7) and using either all trials, only TBF trials, or only successfully forgotten TBF trials.

- 6) We assessed possible inter-individual correlations between the ERS effects and cue power effects. For each cue effect reported in figure 7 we calculated per subject the effect size for that respective effect averaged over the frequency/time data in the cluster (again separately for the LTC instruction effect, the LTC TBF memory effect, the DLPFC instruction effect, and the coherence instruction effect). For example, for the LTC instruction effect we subtracted the average TBF power from the average TBR

power in the interval between 6-30 Hz; 0.35-0.85 sec. We then calculated per frequency bin used in the original analyses the corresponding ERS effect (again for the LTC instruction effect subtracting the TBF ERS values from the TBR ERS values). Then we correlated the ERS effects with the cue power/coherence effects for each ERS frequency bin separately. Again, we corrected for multiple comparisons using cluster-based statistics.

For the LTC instruction and LTC TBF memory effect we found no correlations. Interestingly, however, for the DLPFC instruction effect we found a significant correlation between the ERS effect and the cue power effect ($p = 0.036$; 16-19 Hz; max $t(10)$ -value = 4.01; Response Figure 23). A similar correlation was found for the DLPFC-LTC coherence ($p = 0.022$, 15-17 Hz; max $t(9)$ -value = 4.93). This indicates that participants showing more pronounced DLPFC alpha/beta power increases, and DLPFC-LTC alpha/beta coherence, also showed a more prominent reinstatement of stimulus-specific alpha/beta LTC encoding patterns for TBF vs. TBR items.

Response Figure 23, identical to Figure 8 Inter-individual correlations across participants between the cue instruction effects and the ERS instruction effect for (A) DLPFC power and (B) DLPFC-LTC coherence. Correlations are estimated at the time/frequency ranges of the cue effects (see figure 7). Black bar indicates significance at the $p = 0.05$ levels. Scatterplots indicate the effect at the peak of the significant power/coherence effect. Circles indicate individual participants. Inset indicates the Spearman correlation value.

Thus, we could show that there are instruction effects during the cue period on both DLPFC and LTC power and DLPFC-LTC coherence, as previously reported (Benoit and Anderson, 2012; Depue et al., 2007; Oehrn et al., 2018). Cue effects in DLPFC and on DLPFC-LTC coherence were related across subjects to ERS effects in LTC and DLPFC-LTC, respectively. We now report on these new interesting findings in the results and the discussion.

The manuscript now reads in the results (lines 329-378):

“Finally, we evaluated time-frequency patterns in DLPFC and LTC during the time period of the cue presentation in order to investigate their possible relationship to ERS effects. In the LTC, we found a main effect of instruction (6-30 Hz; 0.35-0.85 sec; $p = 0.013$; max $t(16)$ -value = 4.636; Figure 7A) with a cluster showing relatively higher alpha/beta power after a TBF cue compared to a TBR cue (less pronounced reductions as compared to baseline, i.e. reduced desynchronization) as well as an interaction effect (2-11 Hz; 0.75-1.5 sec; $p = 0.009$; max $t(16)$ -value = 3.89). This interaction was a consequence of stronger theta/alpha activity in the remembered compared to forgotten condition for TBF items (2-13 Hz; 0.8-1.5 sec; $p=0.004$; max $t(16)$ -value = 5.35; Figure 7B), but not for TBR items ($p > 0.05$; Figure 7C). In the DLPFC, we observed larger DLPFC gamma power increases for forgotten as compared to remembered words (55-160 Hz; 0.95-1.15 sec; $p = 0.022$; min $t(15)$ -value = -4.35; Figure 7D). Additionally, we found more pronounced increases of theta/alpha activity for TBF compared to TBR cues (4-14 Hz; 0.5-0.75 sec; $p = 0.046$; max $t(15)$ -value = 4.67; Figure 7E; also see [22]). We did not find any interaction.

We also analyzed cue effects on coherence. We repeated the coherence calculations as performed for the ERS analysis, but using a longer encoding time window (coherence was calculated per trial over the 0-1 second window for each frequency point separately), and then averaged over all trials and channel pairs. Here, we found a main effect of instruction (24-27 Hz; $p = 0.04$; min $t(9)$ -value = -4.13; Figure 7F) showing stronger DLPFC-LTC coherence for the TBF compared to the TBR condition, but did not observe a memory or interaction effect.

We assessed possible inter-individual correlations between the ERS and the cue power effects. For each cue effect reported in Figure 7, we calculated per subject the strength of the respective contrast, averaged over the frequency/time data in the cluster (separately for the LTC instruction effect, the LTC TBF memory effect, the DLPFC instruction effect, and the coherence instruction effect). For example, for the LTC instruction effect, we subtracted the average TBF power from the average TBR power in the interval between 6-30 Hz; 0.35-0.85 sec. We then calculated per frequency bin used in the original ERS analyses the corresponding ERS effect (again for the LTC instruction effect subtracting the TBF ERS values from the TBR ERS values). Then we correlated the ERS effects with the cue power/coherence effects, separately for each ERS frequency bin. Again, we corrected for multiple comparisons using cluster-based statistics. For the LTC instruction and LTC TBF memory effect, we found no correlations. Interestingly, however, for the instruction effect on DLPFC power, we found a significant positive correlation between the ERS effect and the cue effect ($p = 0.036$; 16-19 Hz; max $t(10)$ -value = 4.01; Figure 8). A similar correlation was found for the DLPFC-LTC coherence ($p = 0.022$, 15-17 Hz; max $t(9)$ -value = 4.93). Investigating intra-individual correlations between cue power and ERS size across trials did not result in any significant effect (Supplementary Figure 12). This indicates that participants showing more pronounced DLPFC alpha/beta power increases, and DLPFC-LTC alpha/beta coherence, also showed a more prominent reinstatement of stimulus-specific alpha/beta LTC encoding patterns for TBF vs. TBR items. ”

We also elaborate on this point in the discussion (line 464-472): "Most previous studies investigating directed forgetting have focused on the cue-period window [14,15,22]. Our analysis of activity during the cue-period confirms that during the cue, DLPFC exhibits increased activity for the forgetting instruction (Figure 7E). Moreover, our results show the effects of forgetting instructions in the LTC (Figure 7A-C) and also indicate that DLPFC-LTC

coherence is stronger for forgetting than remembering cues. Finally, we could show that these cue-period effects, previously associated with inhibition [15], are correlated to the strength of the ERS effects across subjects. This further strengthens our interpretation that directed forgetting depends on a selective strengthening of inhibitory patterns present during encoding."

4.

There is another reason why it can be doubted that the measured effects reflect memory reinstatement. A wealth of decoding and similarity-based EEG and fMRI studies have shown that memory success (vs failure) is associated with more (vs less) neural reinstatement. It is thus difficult to understand why TBF items that are forgotten should, even in theory, exhibit the same amount of memory reinstatement as TBR remembered items. At minimum, the authors need to make a convincing argument as to why intentional forgetting is different from incidental forgetting.

RESPONSE: Thanks for bringing this up. Indeed we agree that numerous previous studies with various different methods showed more pronounced reinstatement for successfully remembered vs. forgotten items. One may therefore expect that this is also true for intentionally forgotten items – and this is exactly one of the possible hypotheses that we wanted to test. This may not have been sufficiently explicit in our Introduction. We now write (lines 83-86): "... [one may hypothesize that] successfully forgotten TBF items should show reduced reinstatement, i.e., lower encoding-retrieval similarity, as compared to (incidentally) remembered TBF items – similar to what is seen for forgotten vs. remembered TBR items (Figure 1B)."

Interestingly, however, this is not what we found. Instead, our results clearly demonstrate more item-specific reinstatement for later *forgotten* than remembered items in the TBF condition. This resonates with various findings in the directed forgetting literature showing that intentional forgetting differs critically from incidental forgetting. In the current study, the behavioral results indicate that participants are more likely to forget information if they were instructed to do so. Also, various EEG, fMRI and intracranial EEG studies, some of them from our group, have delineated the specific brain responses to intentional forgetting cues, and shown how they affect mnemonic processing e.g. in the hippocampus (e.g., (Andersen, Tiippana, & Sams, 2004; Benoit & Anderson, 2012; Fellner et al., 2020; Oehrns et al., 2018).

In the revised manuscript, we addressed this issue in greater detail. First, we now describe neural activity from the cue period, and how it interacts with encoding-retrieval similarities of the different item types; second, we directly compared ERS of intentionally vs. incidentally forgotten items. In the following, we would like to elaborate on these analyses and results.

Related to the first point, we analyzed activity during the cue period (please see above, answer to comment 3).

Related to the second point, we directly compared ERS of intentionally vs. incidentally forgotten items (as also requested by Reviewer 2 comment #1). We observed a significant

cluster showing stronger ERS for TBF forgotten compared to TBR forgotten items ($p = 0.038$, frequency range = 17-20 Hz). This comparison was added to the manuscript (page 12) and has been added for comparison in the revised Figure 4 (also see below Response Figure 24).

Response Figure 24 (identical to Figure 4Aiii). Encoding-retrieval similarity for the direct contrast between forgotten TBF and forgotten TBR items.

For the remembered items, we did not observe a significant difference between remembered TBR and remembered TBF items (lowest p-value for any cluster = 0.254).

This result indicates that the forgetting effect in the alpha/beta frequency range is indeed highly specific to TBF items, since it does not occur similarly for TBR items. The remembering effect in the gamma frequency range is less specific, though; it only reflects a simple dissociation (significant effect for TBR items but not for TBF items) rather than a clear double dissociation.

We have added a discussion of this point and now write “When we directly compared ERS of intentionally vs. incidentally forgotten items (i.e., of forgotten TBF vs. forgotten TBR items), we found clear evidence for more pronounced reinstatement of item-specific patterns in the alpha/beta frequency range for forgotten TBF items. This indeed shows that ERS in this frequency range is highly specific for intentionally forgotten items. However, the direct contrast between remembered TBR and remembered TBF items was not significant, suggesting that reinstatement of gamma-band activity patterns may also support memory for TBF items to a certain extent (even though it does not reach significance)” (line 415-422).

5.

Most intentional forgetting studies (e.g. Anderson and Green, 2001, Nature; Benoit and Anderson, 2012, Neuron; Gagnepain et al., 2014, PNAS; Oehrns et al, 2018, CurrBiol) assume that DLPFC exerts a top-down influence directly onto the hippocampus, and content-specific processing in neocortical areas is only modulated as an indirect result. Evidence for a direct impact of DLPFC on LTC, and its relevance for forgetting, should be presented.

RESPONSE: This is an interesting suggestion. Unfortunately, we only have three patients with electrodes in hippocampus, LTC, and DLPFC, which makes it impossible to partial out possible effects of the hippocampus on the currently presented results. However, 13 patients were implanted with electrodes in both LTC and hippocampus. We thus calculated coherence-based ERS between hippocampal channels and LTC. However, we did not find any significant condition differences in ERS values for either TBF or TBR items in any frequency band, suggesting that indirect hippocampal-LTC connections do not account for our findings in LTC (Response Figure 25; identical to the new Supplementary Figure 8).

Response Figure 25 (Identical to Supplementary figure 8). Hippocampus-LTC coherence effects.

Interestingly, in our updated analyses the hippocampus did show more pronounced ERS in the gamma frequency range for TBF than TBR items (Response Figure 13, identical to Supplementary Figure 4). However, these effects were not memory specific (i.e., there was no difference between remembered vs. forgotten TBF items) – again showing that the more pronounced reinstatement of later forgotten vs. remembered TBF items cannot be explained by influences from the hippocampus.

Together, these results strongly suggest that the condition differences of DLPFC-LTC connectivity cannot be explained by influences from the hippocampus. In the revised manuscript, we added the new results on ERS in hippocampus-LTC connectivity patterns as a new Supplementary Figure, and describe them in the Results part (lines 283-288): „Previous studies suggested that intentional forgetting is related to a suppression of hippocampal memory functions (e.g., [15,22,39,40]). We thus investigated ERS effects in the hippocampus (n=13 patients; Supplementary Figure 4) and in hippocampal-LTC

connectivities ($n=13$ patients; Supplementary Figure 8). However, while hippocampal ERS was higher for TBF than TBR items, it did not depend on memory, and hippocampal-LTC connectivities did not show any condition differences.”

Response Figure 26 (Identical to Supplementary Figure 4). Item-specific memory reinstatement for different ROIs (A)

We would like to stress that these results do not exclude a role of the hippocampus for the selective strengthening of memory traces during the instruction cues. Indeed, as shown by various studies, it is likely that the hippocampus exerts a pronounced and critical effect during this period. However, the representations of item-specific information during encoding, and their reinstatement during retrieval, do not seem to depend on the hippocampus. This is actually in line with various studies showing that representations of individual items rely on cortical areas rather than the hippocampus, which appears to serve as an „index“ to bind multiple different items, item features, or item-context associations (for recent findings via invasive recordings, see e.g. (Pacheco-Estefan et al., 2019; Staresina et al., 2019). Please also note that we did not differentiate between recognition memory based on recollection (for which the hippocampus seems to be critical) and recognition memory based on familiarity (which may be exerted without the hippocampus; (Brown & Aggleton, 2001; Diana, Yonelinas, & Ranganath, 2007).

Finally, we would like to mention that from our understanding of the literature, it is not clear whether DLPFC effects on LTC (which are investigated less often than DLPFC-

hippocampal interactions) are actually mediated by the hippocampus. Obviously, during intentional forgetting, the DLPFC influences the hippocampus (Oehrns et al., 2018), but it is not clear whether it also exerts effects on the neocortex, or whether this is mediated by the hippocampus. Interestingly, the study by Gagnepain et al. (Gagnepain, Henson, & Anderson, 2014) found similar exceedance probabilities between a model with direct prefrontal-neocortical interactions (in their case between middle frontal and fusiform gyrus) and a model with indirect connections via the hippocampus. The other mentioned papers did not partial out indirect effects of relevant neocortical regions either.

In sum, while we cannot exclude a role of the hippocampus on lateral temporal ERS effects or ERS of DLPFC-LTC connectivity patterns, the hippocampus does not seem to influence the reinstatement of item-specific memory traces directly. Instead, the hippocampus most likely plays a role for the selective strengthening of aspects of the trace during the cue period. We added this point to the discussion, which now reads (lines 472-474): *“There is no doubt that the cue to instruct needs to have a consequence on the memory trace, perhaps via hippocampal connections that are frequently recruited in directed forgetting tasks (e.g., [22,39,40]).”*

Results:

6.

I find it surprising that no power and connectivity differences exist between to-be-remembered and to-be-forgotten items during the recognition test. If an instruction to forget induces inhibition, and the major neural correlate of such inhibition is an alpha/beta synchronization, one would expect a TBF > TBR power and coherence difference at retrieval, reflecting the presumed reinstatement of the inhibitory portions of the engram. The authors should explain this apparent discrepancy. More generally, it is not easy to understand what might cause the difference in encoding-retrieval similarity between conditions, if there are no differences at encoding (which is expected because all words should be processed the same before the instruction comes on), and no differences during retrieval. Differences between conditions should be present at least on some channels in order to explain the differential correlations. Channel-wise comparisons between TBR and TBF (e.g. cluster- or FDR-corrected) should thus be reported instead of the channel-averaged comparisons shown in the present figures.

RESPONSE: We apologize for the lack of clarity in our descriptions. First of all, we did not conduct channel-averaged comparisons, but correlated activity across the distributed patterns (conceptually similar to encoding-retrieval similarity in fMRI data based on correlations across voxel patterns). Thus, we obtained one correlation value per trial, based on the correlation across channels. It is these correlations values which were then averaged across trials and compared between conditions. Second, we would like to emphasize that the correlation in every individual trial is independent of the overall activity level in that trial. I.e., there can be pronounced correlations at relatively low overall (channel-averaged, or channel-wise) activity levels; or low, negative, or absent correlations at high overall activity levels.

More specifically, our finding of relatively higher correlations of alpha/beta power values for successfully forgotten TBF items is well consistent with a lack of overall power enhancements in any single channel when averaging across trials. To illustrate this, we added a small example in the table below. The numbers in this table could reflect any measure of brain activity (e.g., power values in a specific frequency band).

phase Channel	TBF trial 1		TBF trial 2		TBR trial 1		TBR trial 2		TBF avg		TBR avg	
	E	R	E	R	E	R	E	R	E	R	E	R
1	1	1	2	2	1	1	2	2	1.5	1.5	1.5	1.5
2	2	2	1	1	1	2	2	1	1.5	1.5	1.5	1.5
3	1	1	2	2	2	1	1	2	1.5	1.5	1.5	1.5
4	2	2	1	1	2	2	1	1	1.5	1.5	1.5	1.5
E-R correlation across channels	1		1		0		0		1		0	

Example of encoding-retrieval similarity. E = encoding, R = retrieval

In the table above, all channels have a TBR/TBF condition average (across trials) for both encoding (E) and retrieval (R) of 1.5. However, encoding-retrieval correlations are 1 for every single TBF trial and 0 for every TBR trial. This example shows that ERS values reflect correlations over features (here, channels) in each trial which are independent from trial-averaged power values, making them a useful complementary metric.

These considerations are described in greater detail in the revised manuscript where we write (lines 149-153): *“ERS values reflect correlations of frequency-specific power values across LTC electrodes in every encoding and retrieval trial, which are independent from the overall activity levels in these trials. I.e., there can be pronounced correlations at relatively low overall (channel-averaged) activity levels; or low, negative, or absent correlations at high overall activity levels.”*

Nonetheless, we calculated averaged (over 0.1-0.5 sec) power values in every LTC channel to investigate if single channels showed differential patterns for TBR and TBF trials. We focused this analysis on the frequencies that were significant during the ERS analysis, and analyzed power during both encoding and retrieval. None of the channels survived FDR correction for any of the comparisons. At an uncorrected level, we found 5.1% and 4.8 % significant channels (for a total of 437 LTC channels * 2 study phase (encoding and retrieval) * 3 comparisons (remembered, forgotten, and both)), which is expected due to the type I error proportion of 5%.

7.

One of the authors’ central argument is that recognizing TBR and TBF items depends on reinstatement in different frequency bands, with high gamma frequencies presumably reflecting excitatory connections of an engram that are relevant for remembering TBR items, and alpha/beta frequencies presumably reflecting inhibitory connections of the engram that are relevant for forgetting TBF items.

However, no analysis is reported that directly compares ERS between high and low frequencies, or reports the interaction between high/low frequencies and TBR/TBF conditions. Since the authors use normalized Spearman correlation values as dependent measure, these should be directly comparable. As it stands, the evidence is statistically rather weak, based on the absence of a significant low frequency cluster for TBR, and the absence of a high frequency cluster for TBF. A direct comparison is required to support the major claims.

RESPONSE: Thanks for this comment. We directly compared the alpha/beta and gamma band effects statistically as proposed. We performed a 2*2 RM ANOVA with the factors "frequency" (alpha/beta vs. gamma) and "condition" (TBR vs TBF), using the fisher z-transformed correlation values of the main analysis in the significant cluster for the TBF and the trend cluster for the TBR condition.

This analysis revealed a highly significant interaction between "condition" and "frequency" ($F(1,15) = 11.148$; $p = 0.004$). The follow-up simple comparisons of this interaction have already been reported (see Figure 3). The interaction effect between frequency and condition has been added in the revised version of the manuscript (page 8, lines 172-174).

Methods:

8.

The trial number statement is somewhat ambiguous. It should be clarified whether 50 words (25 to-be-remembered, 25 TBF) was the total for a single block, or the total overall per patient. It should also be made explicit how many trials were on average available per condition per patient for the similarity analyses, after artefact rejection.

RESPONSE: We have added this information (page 7 and page 25). These numbers were the values per block, and there were between 2 and 4 blocks in each patient (mean: 3.69 ± 0.60 blocks in the 16 patients with LTC channels).

After artifact correction, there were 89.0 ± 21.1 artifact-free encoding/retrieval pairs for TBR items, and 89.4 ± 20.7 encoding/retrieval pairs for TBF items. This is described in lines 147-149.

9.

I could not find information in the methods section regarding what hospital the data was recorded in.

RESPONSE: We have added this information. All patients were recorded in the Department of Epileptology of the University of Bonn, Germany.

10.

In order for the reader to understand the nature (e.g. average length) of the feature vectors used for the similarity analyses, the method section should state (a) how many electrodes per patient were available in LTC, DLPFC and the other regions of interest; and (b) into how many overlapping or non-overlapping time windows the 100-500ms time windows were binned before vectorization.

RESPONSE: We have added this information (page 28). Related to (a), there were 27.3 ± 11.1 LTC channels (range, 7-45; total, 437 channels); 14.2 ± 4.7 DLPFC channels (range, 8-23; total, 213 channels); 10.7 ± 5.6 parietal channels (range, 5-24, total, 107 channels). 11.5 ± 4.7 hippocampal channels (range, 6-23; total, 213 channels). Related to (b), there were 9 non-overlapping time bins between 100-500ms.

11.

Why did the authors choose a frequency band of 10-19Hz for the transfer entropy analysis, if the significant cluster for coherence (which was used for frequency selection according to the results section) was 12-17Hz?

RESPONSE: We choose to slightly increase the frequency window as any filter has a roll-off around the filter edge. However, using the exact window of the significant effect gives qualitatively the same result (Response Figure 13; main effect of forgetting, $t(9) = 3.979$, $p = 0.004$; difference between remembered and forgotten items: $t(9) = 2.609$, $p = 0.031$).

Response Figure 27. TBF TE with narrow-band filter

12.

The methods or results section should clarify what time window was used for the transfer entropy analysis (i.e., was it the same 0.1-0.5 sec as for the other analyses).

RESPONSE: This was indeed the 0.1-0.5 window and we have clarified this further.

13.

The results section should report any potential differences in response times between TBR and TBF items. If patients have longer response latencies for forgotten items, reinstatement would likely take longer too, creating a confound for the neural analyses.

RESPONSE: We reiterate here the response to reviewer 1 (comment 2) who had the same concern. Specifically, we conducted additional analyses regarding the reaction times. We calculated for the TBR and TBF conditions the median reaction times separately for remembered and forgotten items, and compared them in a 2*2 RM ANOVA with factors "condition" (TBR/TBF) and "memory" (remembered or forgotten). Results showed overall faster responses for remembered compared to forgotten words ($F(1,15) = 8.362$, $p = 0.0118$; Response Figure 28) and faster responses for the TBR compared to the TBF condition ($F(1,15) = 5.212$, $p = 0.037$). The interaction between memory and condition did not reach significance, but showed a trend ($F(1,15) = 3.48$, $p = 0.082$). The simple effects analysis comparing TBR and TBF separately for remembered and forgotten words showed that there was only a difference for the remembered ($t(15) = 2.66$, $p = 0.036$), but not the forgotten words ($t(15) = -0.115$, $p = 0.98$).

Response Figure 28. Reaction time effects (identical to Supplementary Figure 2a).

This means that overall incorrect responses (i.e., responses in forgotten trials) are given more slowly than correct responses. This effect is not surprising and has been

repeatedly observed before and may reflect some hesitation prior to the incorrect response or memory scanning mechanisms (Sternberg, 1969; Theios, 1973).

Furthermore, this pattern of results suggests that reaction times do not influence the significant alpha/beta ERS, because (1) the trend significant interaction suggests that RT differences between remembered and forgotten TBR words are larger than between remembered and forgotten TBF words, while alpha/beta ERS only differs between remembered and forgotten TBF words; and (2) RTs differ only between remembered TBR vs. TBF words and not between forgotten TBR vs. TBF words, while alpha/beta ERS only differs between forgotten TBR vs. TBF words and not between remembered TBR vs. TBF words. This new result is described in lines 221-223: “Additionally, the ERS for the TBF forgotten words was higher than for the TBR forgotten words in the beta range (17-20 Hz; cluster statistics = -9.58; $p = 0.038$; max $t(15)$ -value = -2.66).”

Nonetheless, to further investigate a possible influence of reaction times on ERS results, we correlated trial-by-trial reaction times to the strength of the correlation between the encoding and retrieval pair. This was done using Spearman correlations for each frequency point that was used in the original analysis (separately for each condition). In other words, we investigated whether the ERS effect itself was related to reaction times. The correlation showed no clusters even at an uncorrected level (Response Figure 29). This again indicates that reaction times do not seem to influence ERS effects.

Response Figure 29. Trial-by-trial correlations between reaction times and ERS values.

Finally, we calculated if the response preparation (locked to the button press) had an effect on the ERS at the significant frequency ranges of our main effects (for low frequencies: 17-22 Hz; for high frequencies: 105-110 Hz). We thus re-calculated ERS, but instead of using the stimulus-locked retrieval data, we used response-locked data. As the exact time of a possible effect is unknown, we repeated the analysis at various encoding and response-locked time periods (always using windows of 400ms, as in our original analysis). We did not find any significant clusters for this response-locked ERS analysis (see Response Figure 30). Also, no clusters were found when the same analysis was conducted for memory specific effects (i.e., remembered vs. forgotten TBR items; remembered vs. forgotten TBF items).

In the new version of the manuscript, we included the reaction time analyses and added the respective figure to the supplementary figures (see page ** and new supplementary figure 2).

Response figure 30. Response locked ERS analysis. ERS analyses using various encoding times and response times. A) Analyses performed at 17-22 Hz. B) Analyses performed at 105-110 Hz.

Minor:

- The authors should consider reporting d' as a behavioural measure that is corrected for false alarm rates.

RESPONSE: The results for d' are statistically similar to the comparison of the hit rates, since the same number of false alarms was considered for the TBR and TBF items (since there were no condition-specific false alarms). This can be seen in Response figure 31 (TBR vs. TBF: $t(15) = 3.00$, $p = 0.009$). Since the exact values of d' may be interesting for comparison to previous studies, we added this information as new Supplementary Figure 2B.

Response figure 31. Identical to Supplementary Figure 2B. D-prime analysis

- It would be reassuring to see a supplementary figure plotting the raw correlation values that the differential ERS (i.e., within vs between items) is based on. A strong positive within-vs-between difference can arise because the item-specific (within) similarities are very high, or because the non-specific similarities are very low.

RESPONSE: We added a new supplementary figure showing the raw correlation values (Response Figure 32; identical to Supplementary Figure 5). As can be seen from these figures, all effects are due to the same-item similarities being higher, and not because of a difference in the different-item condition.

5.

Response Figure 32, identical to Supplementary figure 5: Raw correlation traces of the encoding-retrieval similarity analysis of same vs. different items. Significance lines reflecting the ERS comparisons (on-diagonal vs. off-diagonal) are identical to those shown in the main manuscript. Effects are driven by increases in same item correlations rather than by differences in the different-item correlations. Conventions are the same as in Figure 3B.

- Do the authors have any empirical evidence for the statement that “word frequency maps to an automatic inhibitory signal initiated during encoding” (l.391-2)? If this was the case, high and low

frequency words during encoding (and retrieval) should elicit very distinct power signatures in the low frequency bands, and differential connectivity patterns, according to the authors' own logic. Could be tested empirically. Also, did high and low frequency words show different amounts of remembering/forgetting?

RESPONSE: The analysis and discussion regarding the RSA were removed from the main manuscript (based on comment 6 of reviewer 1 and comment 6 of reviewer 2).

References:

- Andersen, T. S., Tiippana, K., & Sams, M. (2004). Factors influencing audiovisual fission and fusion illusions. *Cognitive Brain Research*, *21*(3), 301-308.
- Anderson, M. C., & Green, C. (2001). Suppressing unwanted memories by executive control. *Nature*, *410*(6826), 366-369.
- Bauer, M., Oostenveld, R., Peeters, M., & Fries, P. (2006). Tactile spatial attention enhances gamma-band activity in somatosensory cortex and reduces low-frequency activity in parieto-occipital areas. *Journal of Neuroscience*, *26*(2), 490-501.
- Benoit, R. G., & Anderson, M. C. (2012). Opposing mechanisms support the voluntary forgetting of unwanted memories. *Neuron*, *76*(2), 450-460.
- Bjork, R. A. (1970). Positive forgetting: The noninterference of items intentionally forgotten. *Journal of Verbal Learning and Verbal Behavior*, *9*(3), 255-268.
- Brown, M. W., & Aggleton, J. P. (2001). Recognition memory: what are the roles of the perirhinal cortex and hippocampus? *Nature Reviews Neuroscience*, *2*(1), 51-61.
- Buzsaki, G. (2006). *Rhythms of the Brain*: Oxford University Press.
- Dehghani, N., Peyrache, A., Telenczuk, B., Le Van Quyen, M., Halgren, E., Cash, S. S., . . . Destexhe, A. (2016). Dynamic balance of excitation and inhibition in human and monkey neocortex. *Scientific reports*, *6*(1), 1-12.
- Diana, R. A., Yonelinas, A. P., & Ranganath, C. (2007). Imaging recollection and familiarity in the medial temporal lobe: a three-component model. *Trends in Cognitive Sciences*, *11*(9), 379-386.
- Engen, H. G., & Anderson, M. C. (2018). Memory control: a fundamental mechanism of emotion regulation. *Trends in Cognitive Sciences*, *22*(11), 982-995.
- Fellner, M.-C., Waldhauser, G. T., & Axmacher, N. (2020). Tracking selective rehearsal and active inhibition of memory traces in directed forgetting. *Current Biology*, *30*(13), 2638-2644. e2634.
- Gagnepain, P., Henson, R. N., & Anderson, M. C. (2014). Suppressing unwanted memories reduces their unconscious influence via targeted cortical inhibition. *Proceedings of the National Academy of Sciences*, *111*(13), E1310-E1319.
- Gao, R., Peterson, E. J., & Voytek, B. (2017). Inferring synaptic excitation/inhibition balance from field potentials. *NeuroImage*, *158*, 70-78.
- Geiselman, R. E., Bjork, R. A., & Fishman, D. L. (1983). Disrupted retrieval in directed forgetting: a link with posthypnotic amnesia. *Journal of Experimental Psychology: General*, *112*(1), 58.
- Gips, B., van der Eerden, J. P., & Jensen, O. (2016). A biologically plausible mechanism for neuronal coding organized by the phase of alpha oscillations. *European Journal of Neuroscience*, *44*(4), 2147-2161.
- HajiHosseini, A., & Holroyd, C. B. (2015). Reward feedback stimuli elicit high-beta EEG oscillations in human dorsolateral prefrontal cortex. *Scientific reports*, *5*(1), 1-8.
- Jang, A. I., Wittig Jr, J. H., Inati, S. K., & Zaghoul, K. A. (2017). Human cortical neurons in the anterior temporal lobe reinstate spiking activity during verbal memory retrieval. *Current Biology*, *27*(11), 1700-1705. e1705.
- Jensen, O., Bonnefond, M., & VanRullen, R. (2012). An oscillatory mechanism for prioritizing salient unattended stimuli. *Trends in Cognitive Sciences*, *16*(4), 200-206.
- Jensen, O., & Mazaheri, A. (2010). Shaping functional architecture by oscillatory alpha activity: gating by inhibition. *Frontiers in Human Neuroscience*, *4*.
- Klimesch, W. (1999). EEG alpha and theta oscillations reflect cognitive and memory performance: a review and analysis. *Brain Research Reviews*, *29*(2), 169-195.
- Klimesch, W., Sauseng, P., & Hanslmayr, S. (2007). EEG alpha oscillations: the inhibition-timing hypothesis. *Brain Research Reviews*, *53*(1), 63-88.
- Lachaux, J.-P., Axmacher, N., Mormann, F., Halgren, E., & Crone, N. E. (2012). High-frequency neural activity and human cognition: past, present and possible future of intracranial EEG research. *Progress in neurobiology*, *98*(3), 279-301.

- Markicevic, M., Fulcher, B. D., Lewis, C., Helmchen, F., Rudin, M., Zerbi, V., & Wenderoth, N. (2020). Cortical excitation: inhibition imbalance causes abnormal brain network dynamics as observed in neurodevelopmental disorders. *Cerebral Cortex*, *30*(9), 4922-4937.
- Mary, A., Dayan, J., Leone, G., Postel, C., Fraise, F., Malle, C., . . . De la Sayette, V. (2020). Resilience after trauma: The role of memory suppression. *Science*, *367*(6479).
- Mikulan, E., Hesse, E., Sedeño, L., Bekinschtein, T., Sigman, M., del Carmen García, M., . . . Ibáñez, A. (2018). Intracranial high- γ connectivity distinguishes wakefulness from sleep. *NeuroImage*, *169*, 265-277.
- Miller, P., & Wang, X.-J. (2006). Inhibitory control by an integral feedback signal in prefrontal cortex: a model of discrimination between sequential stimuli. *Proceedings of the National Academy of Sciences*, *103*(1), 201-206.
- Mitra, P. P., & Pesaran, B. (1999). Analysis of dynamic brain imaging data. *Biophysical Journal*, *76*(2), 691-708.
- Nimmrich, V., Draguhn, A., & Axmacher, N. (2015). Neuronal network oscillations in neurodegenerative diseases. *Neuromolecular medicine*, *17*(3), 270-284.
- Oehr, C. R., Fell, J., Baumann, C., Rosburg, T., Ludowig, E., Kessler, H., . . . Axmacher, N. (2018). Direct Electrophysiological Evidence for Prefrontal Control of Hippocampal Processing during Voluntary Forgetting. *Current Biology*.
- Pacheco-Estefan, D., Sánchez-Fibla, M., Duff, A., Principe, A., Rocamora, R., Zhang, H., . . . Verschure, P. F. (2019). Coordinated representational reinstatement in the human hippocampus and lateral temporal cortex during episodic memory retrieval. *Nature communications*, *10*(1), 1-13.
- Palop, J. J., & Mucke, L. (2016). Network abnormalities and interneuron dysfunction in Alzheimer disease. *Nature Reviews Neuroscience*, *17*(12), 777-792.
- Percival, D. B., & Walden, A. T. (1993). *Spectral analysis for physical applications*: cambridge university press.
- Ray, S., Niebur, E., Hsiao, S. S., Sinai, A., & Crone, N. E. (2008). High-frequency gamma activity (80–150 Hz) is increased in human cortex during selective attention. *Clinical Neurophysiology*, *119*(1), 116-133.
- Sauseng, P., Freunberger, R., Feldheim, J. F., & Hummel, F. C. (2011). Right prefrontal TMS disrupts interregional anticipatory EEG alpha activity during shifting of visuospatial attention. *Frontiers in psychology*, *2*, 241.
- Sauseng, P., Klimesch, W., Heise, K. F., Gruber, W. R., Holz, E., Karim, A. A., . . . Hummel, F. C. (2009). Brain oscillatory substrates of visual short-term memory capacity. *Current Biology*, *19*(21), 1846-1852.
- Shah, P., Bernabei, J. M., Kini, L. G., Ashourvan, A., Boccanfuso, J., Archer, R., . . . Lucas, T. H. (2019). High interictal connectivity within the resection zone is associated with favorable post-surgical outcomes in focal epilepsy patients. *NeuroImage: Clinical*, *23*, 101908.
- Spitzer, B., & Haegens, S. (2017). Beyond the status quo: a role for beta oscillations in endogenous content (re) activation. *eneuro*, *4*(4).
- Staresina, B. P., Reber, T. P., Niediek, J., Boström, J., Elger, C. E., & Mormann, F. (2019). Recollection in the human hippocampal-entorhinal cell circuitry. *Nature communications*, *10*(1), 1-11.
- Sternberg, S. (1969). Memory-scanning: Mental processes revealed by reaction-time experiments. *American Scientist*, *57*(4), 421-457.
- Theios, J. (1973). Reaction time measurements in the study of memory processes: Theory and data *Psychology of learning and motivation* (Vol. 7, pp. 43-85): Elsevier.
- Yaffe, R. B., Kerr, M. S., Damera, S., Sarma, S. V., Inati, S. K., & Zaghloul, K. A. (2014). Reinstatement of distributed cortical oscillations occurs with precise spatiotemporal dynamics during successful memory retrieval. *Proceedings of the National Academy of Sciences*, *111*(52), 18727-18732.
- Yi, T.-M., Huang, Y., Simon, M. I., & Doyle, J. (2000). Robust perfect adaptation in bacterial chemotaxis through integral feedback control. *Proceedings of the National Academy of Sciences*, *97*(9), 4649-4653.

- Zhang, H., Fell, J., & Axmacher, N. (2018). Electrophysiological mechanisms of human memory consolidation. *Nature Communications*, *9*(1), 1-11.
- Zhang, H., Fell, J., Staresina, B. P., Weber, B., Elger, C. E., & Axmacher, N. (2015). Gamma power reductions accompany stimulus-specific representations of dynamic events. *Current Biology*, *25*(5), 635-640.

REVIEWER COMMENTS

Reviewer #1 (Remarks to the Author):

The authors did a great job with this revision, and especially with the additional analyses. They have also made substantial improvements in toning down some of their less supported claims. The only comments I have remaining concern the abstract, which may still be somewhat misleading: 1. (related to original point #3) The authors state: "it is commonly assumed that active forgetting [...] weakens memory traces. Contrariwise, [...] we find that memory traces for actively forgotten information are partially preserved". These two claims are not contrary to each other, and presenting them as such may be misleading. I suggest leaving the commonly assumed claim, which is not refuted in the paper, out of the abstract altogether. 2. (related to original point #4) The authors state: "These results demonstrate that intentional forgetting [...] strengthens inhibitory top-down connections". Due to the lack of a control condition, the data could be interpreted as though intentional remembering weakens inhibitory top-down connections, whereas intentional forgetting has no impact on them. I suggest mentioning the comparison in the abstract (e.g., "intentional forgetting involves stronger inhibitory top-down connection relative to intentional remembering"). I strongly suggest revising similar claims made in other parts of the manuscript and expanding on the challenges to the presented interpretation, but if the authors choose not to do so, the paragraph they added in lines 461-465 would suffice.

Reviewer #2 (Remarks to the Author):

The authors have done extensive work to help explicate and support their findings and this is a comprehensive and interesting paper.

I only have a single point of clarification. It seems as though Figure 3 Aiii shows that there is greater reinstatement in TBF items for an alpha-beta signal (in the LTC). Then, this is the same signal which shows greater power (during the instruction) in the DLPFC and LTC for intentionally forgotten items (Figure 7), and greater DLPFC-LTC coherence. This would seem consistent with the suggestion that the greater reinstatement for TBF forgotten items in this freq range reflect reinstatement of this additional cue related information at the time of retrieval (whether one calls this "context" or "control" information is sort of irrelevant – it comes down to the same thing in this paradigm). The authors summarize this idea in the response: "One may still argue that it is not shared activity during encoding of TBF items but during the cue period which is reinstated as a context of all TBF items." As a general statement, I think this interpretation (additional information provided by the cue that affects reinstatement magnitudes) is something readers will be curious about, and it should be addressed directly in the Discussion. More specifically, the plot in Suppl Figure 7 looks as though there is a consistent correlation in power at that same alpha-beta signal that is common across TBF items, although it apparently doesn't reach significance via the clustering requirement. Could this signal be contributing to observed differences in reinstatement? Isn't the most straightforward way to control for additional cue-related information contributing to the observed ERS simply to execute a multivariate analysis that incorporates the cue-related power differences in the quantification of ERS differences (for LTC power and DLPFC-LTC connectivity)? Perhaps I simply am misunderstanding a step in the logic here (or how one of the controls tackles this questions), as clearly the authors understand that idea and have attempted to grapple with this alternate interpretation (to their credit).

Reviewer #3 (Remarks to the Author):

The authors did an extensive revision of this manuscript, with many new analyses and discussion points added. Many concerns raised in the initial reviews are now sufficiently addressed (including my previous points 4-5 and 7-13), and I find the analysis comparing incidental and intentional forgetting particularly useful and convincing. While the results and statistics are solid, unfortunately the major conceptual concerns remain. Most critically, I still see no compelling evidence for the manuscript's narrative that "inhibitory engrams" are strengthened by a forget cue

(points 1 and 3 below). Moreover, an effect as early as 100msec is highly unlikely to reflect memory reinstatement, again casting doubt on the interpretation of the findings (point 2).

(1) Regarding the “inhibitory engram reactivation” narrative, the authors now clarify that when talking about excitation and inhibition, they never meant to refer to the level of synaptic connections, but rather to functional inhibition on the level of global brain networks (i.e., distinct patches of cortex). It is argued that because previous studies have found increases in alpha/beta oscillations related to functional inhibition, an alpha/beta increase can be interpreted as functional inhibition. I have several difficulties with this argument.

a. The reasoning very strongly relies on reverse inference (i.e., alpha/beta increase = inhibition).

b. The inhibitory engram hypothesis assumes, according to the manuscript (l. 86-89), that a forget cue strengthens inhibitory components/connections in an engram that are already laid down during initial learning. If this was the case, we should strongly expect this strengthening of inhibitory components to show up in average power during retrieval, such that in the critical time window of interest, TBF items should elicit stronger alpha/beta power than TBR items, and TBF-f items should elicit stronger alpha/beta power than TBF-r items. The absence of univariate power difference in the time window of interest thus speaks against the idea of strengthening of inhibitory engram components. I understand it is mathematically possible to get increased correlation without overall activity increases (see authors’ response to previous point 6), but conceptually, an increase would nevertheless be expected if inhibitory features of a memory are strengthened in TBF items. On a similar note, the manuscript suggests that such an alpha/beta increase would reflect a pathological imbalance of excitation/inhibition, which seems quite dramatic given the wealth of findings in the attention/WM literature showing such alpha/beta increases in response to cognitive manipulations in healthy subjects.

c. The attention literature on alpha and inhibition shows that entire cortical areas are inhibited if irrelevant for the task (e.g., ipsilateral hemisphere increases during lateralized attention tasks). If functional inhibition happens at this global level, why would inhibition be observed in LTC, an area that is highly relevant for word processing?

(2) The manuscript, throughout, uses the term “reinstatement” to refer to the encoding-retrieval similarity (ERS), even though these effects are observed very early after word presentation (<300msec). In the memory literature, reinstatement typically refers to the recapitulation of encoding-related information that is not directly conveyed by the sensory cue, and instead is the product of a hippocampal pattern completion process. Previous literature clearly shows that this process takes several hundred milliseconds to complete, such that neural patterns in the first 400-500msec reflect sensory processing of the cue/recognition probe, and only start to reflect reinstated patterns after 500msec. An early effect as reported here is thus highly unlikely to reflect memory reinstatement. Different from what the results section suggests (l. 506-508), the new time-shifted analyses confirm that the ERS peaks extremely early, around 100msec after word onset (see maximum value in Fig. 4Cii, middle panel, TBF-forgotten items). Such an early ERS almost certainly reflects the processing of the retrieval cue, which in a recognition test fully overlaps with the encoding stimulus. To be clear, the results convincingly show that word processing is altered by a preceding forget instruction, and this is an interesting finding; the timing just does not support a reinstatement interpretation. As an alternative suggestion, the forget instruction might terminate the encoding process early, leading to reduced repetition priming of the stimulus? Such an interpretation could, for example, be corroborated by comparing similarity of TBF and TBR items to novel items purely within retrieval (rather than between encoding and retrieval), with the expectation that TBF items (and specifically TBF-forgotten ones) behave more like novel items.

(3) All three reviewers were somewhat surprised that the analyses focus on the ERS between the initial word encoding window and retrieval, rather than the initial cue window and retrieval. The cue window is where, in theory, cognitive control should start modifying the memory trace for a given word. The authors now added a set of results from the cue period, confirming that DLPFC exhibits patterns consistent with inhibitory control during the cue period, and a correlation of this activity with the ERS effect for TBF items. These analyses provide valuable additional evidence for

the top-down influence of DLPFC during this period. Regarding the “inhibitory engram” interpretation, however, it seems more relevant to include the results of an ERS analysis that compares the word representations during the cue window at encoding (where the critical modification of an active memory trace presumably happens) with word representations at later retrieval (where the modified trace is then presumably reactivated).

More minor comments:

(4) The abstract and discussion (l. 387-389, l. 401-402) suggest that TBR items showed item-specific reinstatement, though this effect is no longer significant. These statements should be qualified.

(5) The statistics in the time-shifting analyses, shown in Fig. 4Ci and Cii are probably inflated, because the frequencies are pre-selected to show significant effects in the main analysis. The figure is still informative descriptively, in order to see where the ERS peaks in time.

RESPONSE LETTER

We thank the reviewers for their valuable suggestions. We answered the remaining concerns in this response letter. All changes in the main manuscript are highlighted in yellow.

Reviewer #1 (Remarks to the Author):

The authors did a great job with this revision, and especially with the additional analyses. They have also made substantial improvements in toning down some of their less supported claims. The only comments I have remaining concern the abstract, which may still be somewhat misleading:1. (related to original point #3) The authors state: "it is commonly assumed that active forgetting [...] weakens memory traces. Contrariwise, [...] we find that memory traces for actively forgotten information are partially preserved". These two claims are not contrary to each other, and presenting them as such may be misleading. I suggest leaving the commonly assumed claim, which is not refuted in the paper, out of the abstract altogether.2. (related to original point #4) The authors state: "These results demonstrate that intentional forgetting [...] strengthens inhibitory top-down connections". Due to the lack of a control condition, the data could be interpreted as though intentional remembering weakens inhibitory top-down connections, whereas intentional forgetting has no impact on them. I suggest mentioning the comparison in the abstract (e.g., "intentional forgetting involves stronger inhibitory top-down connection relative to intentional remembering"). I strongly suggest revising similar claims made in other parts of the manuscript and expanding on the challenges to the presented interpretation, but if the authors choose not to do so, the paragraph they added in lines 461-465 would suffice.

We thank the reviewer for their positive assessment and have changed the abstract and manuscript accordingly.

We changed the following:

- In the abstract we now write *"While memory retention strengthens memory traces, it is unclear what happens to the memory traces of events that are actively forgotten. Using intracranial EEG recordings [...]"* and *"These results suggest that intentional forgetting relies more on inhibitory top-down connections than intentional remembering"*
- In the discussion we state *"Here, we aimed to unravel whether we could find selectively modified memory traces of intentionally forgotten information"* instead of *"we aimed to unravel whether memory traces of intentionally forgotten information were either erased (Figure 1B) or selectively modified"*.
- We also kept the paragraph in lines 461-465 in addition to these revisions

Reviewer #2 (Remarks to the Author):

The authors have done extensive work to help explicate and support their findings and this is a comprehensive and interesting paper.

I only have a single point of clarification. It seems as though Figure 3 Aiii shows that there is greater reinstatement in TBF items for an alpha-beta signal (in the LTC). Then, this is the same signal which shows greater power (during the instruction) in the DLPFC and LTC for intentionally forgotten items (Figure 7), and greater DLPFC-LTC coherence. This would seem consistent with the suggestion that the greater reinstatement for TBF forgotten items in this freq range reflect reinstatement of this additional cue related information at the time of retrieval (whether one calls this “context” or “control” information is sort of irrelevant – it comes down to the same thing in this paradigm). The authors summarize this idea in the response: “One may still argue that it is not shared activity during encoding of TBF items but during the cue period which is reinstated as a context of all TBF items.” As a general statement, I think this interpretation (additional information provided by the cue that affects reinstatement magnitudes) is something readers will be curious about, and it should be addressed directly in the Discussion. More specifically, the plot in Suppl Figure 7 looks as though there is a consistent correlation in power at that same alpha-beta signal that is common across TBF items, although it apparently doesn’t reach significance via the clustering requirement. Could this signal be contributing to observed differences in reinstatement? Isn’t the most straightforward way to control for additional cue-related information contributing to the observed ERS simply to execute a multivariate analysis that incorporates the cue-related power differences in the quantification of ERS differences (for LTC power and DLPFC-LTC connectivity)? Perhaps I simply am misunderstanding a step in the logic here (or how one of the controls tackles this questions), as clearly the authors understand that idea and have attempted to grapple with this alternate interpretation (to their credit).

We thank the reviewer for their overall positive verdict and the suggested additional analysis. We addressed the question whether cue-related alpha/beta activity mediates item-specific reactivation by performing partial correlation analyses. This allowed us to account for effects of power changes during the cue period on power and coherence ERS. Our findings indicate that LTC alpha/beta power during the cue period does not exert a substantial effect on item-specific encoding-retrieval, as we obtain similar results to the simple correlation analyses (see figures below).

For the power analyses, the effect of item-specificity remained significant (contrast between same and different TBF forgotten items). For both power and coherence, item-specific ERS was higher for forgotten TBF compared to remembered TBF trials. The contrast between coherence ERS for same versus different forgotten TBF items appeared qualitatively similar, but did not reach significance (largest cluster: $p = 0.197$).

We agree that during the cue period, one would expect the memory trace to be modified and differentiate between TBR and TBF trials. However, this may concern aspects of the memory trace that are either not reactivated during the recognition memory test or not item-specific. While the word is presented during both encoding and retrieval, it is not shown during the cue period. We speculate that these different processing modes may explain why activity during the cue period may affect an aspect (or a “format”) of the memory trace that differs from the one that re-occurs during retrieval.

We added these results as a supplementary analysis and in a new Supplementary Figure 7 and also reported in the discussion about our interpretation of what is occurring during the cue period and why it does not necessarily reflect the same activity that contributes to ERS. The discussion now reads: *“Interestingly, however, when we regressed out cue-related activity, our main ERS effects remained (see Supplementary Figure 7B+C). While activity during the cue period should modify the memory trace, it does not necessarily imply that activity during the cue period reflects those aspects of item-specific activity that occur at encoding and retrieval in response to a word. During the cue, activation of memory traces requires active retrieval of sensory representations. During encoding and retrieval, this is not needed as the word itself is presented. The ERS thus reflects the response to a sensory stimulus, combining both perceptual processes and memory-related item-specific information associated with the sensory representation. By contrast, during the cue period the word is not presented, which may result in different activation patterns”*.

Response Figure 1 (identical to Supplementary Fig 7). ERS after accounting for alpha/beta power during the cue period. B) ERS as in Figure 4Aii, but using partial correlations correcting for cue activity (at matched timepoints and frequencies). The differences between TBF remembered and TBF forgotten trials as well as the contrast between same versus different TBF forgotten items remained significant ($p = 0.043$ and $p = 0.041$, respectively). C) ERS as in Figure 5Aii, but using partial correlations correcting for cue activity. The difference between remembered versus forgotten TBF trials remained significant ($p = 0.03$). The contrast between same versus different TBF forgotten items appeared qualitatively similar, but did not reach significance (largest cluster: $p = 0.197$). Shaded error bars indicate the standard error of the mean.

Reviewer #3 (Remarks to the Author):

The authors did an extensive revision of this manuscript, with many new analyses and discussion points added. Many concerns raised in the initial reviews are now sufficiently addressed (including my previous points 4-5 and 7-13), and I find the analysis comparing incidental and intentional forgetting particularly useful and convincing. While the results and statistics are solid, unfortunately the major conceptual concerns remain. Most critically, I still see no compelling evidence for the manuscript's narrative that "inhibitory engrams" are strengthened by a forget cue (points 1 and 3 below). Moreover, an effect as early as 100msec is highly unlikely to reflect memory reinstatement, again casting doubt on the interpretation of the findings (point 2).

(1) Regarding the "inhibitory engram reactivation" narrative, the authors now clarify that when talking about excitation and inhibition, they never meant to refer to the level of synaptic connections, but rather to functional inhibition on the level of global brain networks (i.e., distinct patches of cortex). It is argued that because previous studies have found increases in alpha/beta oscillations related to functional inhibition, an alpha/beta increase can be interpreted as functional inhibition. I have several difficulties with this argument.

a. The reasoning very strongly relies on reverse inference (i.e., alpha/beta increase = inhibition).

We agree with the reviewer that the interpretation of alpha/beta increases as being related to inhibition relies on previous literature (Jensen & Mazaheri, 2010a; Klimesch, 1999), and as such is an interpretation based on reverse inference. While we clearly acknowledge the general limitations of reverse inference and have added a discussion of this point (see below), we would like to argue that reverse inferences are needed for a mechanistic (neurobiological) understanding of directed forgetting effects.

More specifically, the directed forgetting paradigm is a very robust behavioral approach to study how forgetting instructions during encoding lead to later forgetting. It therefore provides a clear behavioral marker of forgetting. However, it is unclear what the mechanism of forgetting is. Is it rather related to a lack of rehearsal or to active inhibition in the TBF condition? This has been the topic of intense discussions since the directed forgetting paradigm has been introduced. While there are some behavioral approaches for addressing this question (which are described in (Anderson & Hanslmayr, 2014)), it is often assumed that cognitive neuroscience approaches also help address this issue. The logic of these approaches then necessarily relies on reverse inference, as the mechanism of interest (active inhibition versus automatic memory decay) is not directly accessible behaviorally (if this were the case, one may argue that the relevance of cognitive neuroscience approaches would be substantially lower). For example, showing that the DLPFC is active during the cue period has been interpreted as evidence for the involvement of cognitive control processes to active forgetting (e.g. (Anderson & Green, 2001)), which is based on literature showing the involvement of DLPFC in cognitive control. Again, this interpretation obviously relies on reverse inference.

We are well aware of the inherent problems of reverse inference and have explicitly addressed its validity conditions for different neuroimaging modalities (fMRI, EEG and intracranial EEG) in a previous review article (Axmacher, Elger, & Fell, 2009). As also elaborated in (Poldrack, 2006), the validity of reverse inference depends, among other factors, on the specificity with which the observed neural signature reflects a given cognitive function. Related to the current study, this concerns the

specificity of alpha/beta power increases as reflecting inhibition (i.e., how likely is it that there is inhibition if alpha/beta power increases are observed). There is an abundance of literature especially on alpha oscillations and inhibition (Jensen & Mazaheri, 2010b; Klimesch, Sauseng, & Hanslmayr, 2007), as well as recent literature regarding beta oscillations (Bastos et al., 2015; Spitzer & Haegens, 2017), supporting our interpretation. We agree however that this question is not fully resolved yet and needs to be addressed further. Specifically, a formal analysis and empirical quantification of the validity of inferring inhibition from increases in alpha and/or beta frequency oscillations is necessary (Poldrack, 2006). We have now addressed this point in the Discussion section. Moreover, while we would still suggest to interpret alpha/beta frequency oscillations as reflecting inhibition in line with various previous studies, we now explicitly mention that this interpretation relies on reverse inference, and that alternative interpretations are possible. We also changed the title of the manuscript and modified the abstract and the introduction sections. All changes are summarized at the end of the responses to concern #1.

b. The inhibitory engram hypothesis assumes, according to the manuscript (l. 86-89), that a forget cue strengthens inhibitory components/connections in an engram that are already laid down during initial learning. If this was the case, we should strongly expect this strengthening of inhibitory components to show up in average power during retrieval, such that in the critical time window of interest, TBF items should elicit stronger alpha/beta power than TBR items, and TBF-f items should elicit stronger alpha/beta power than TBF-r items. The absence of univariate power difference in the time window of interest thus speaks against the idea of strengthening of inhibitory engram components. I understand it is mathematically possible to get increased correlation without overall activity increases (see authors' response to previous point 6), but conceptually, an increase would nevertheless be expected if inhibitory features of a memory are strengthened in TBF items. On a similar note, the manuscript suggests that such an alpha/beta increase would reflect a pathological imbalance of excitation/inhibition, which seems quite dramatic given the wealth of findings in the attention/WM literature showing such alpha/beta increases in response to cognitive manipulations in healthy subjects.

We acknowledge that our use of the word "strengthening" may have been imprecise and even inaccurate: It seems to suggest an exclusive increase in activity of some connections without concomitant decreases, which indeed would necessarily imply an overall power increase. Instead, it is more accurate to refer to a selective modification of item-specific patterns of activations and connections, such that power increases in some electrodes and decreases in others (or for connectivity, that coherence increases in some pairs of LTC-DLPFC electrodes and decreases in others). This combination of increases and decreases also reflects the pattern in our example table in the previous response letter mentioned by the Reviewer. Indeed, most literature shows that memory formation is associated with strengthening of specific connections, while others weaken as an automatic process of not using these connections, as well as an active rearrangement of connection strengths (Bailey & Kandel, 1993; Lamprecht & LeDoux, 2004; Martin, Grimwood, & Morris, 2000). Our focus on strengthening unintentionally and incorrectly implied an overall change in sum connection strength. We changed our terminology throughout the manuscript and do not refer to overall strengthening, but to a rearrangement or modification of connection patterns (please see below for a summary of all changes).

Importantly, various previous studies found increases in ERS values in a given experimental condition without concomitant increases in overall power (Pacheco Estefan et al., 2019; Yaffe,

Shaikhouni, Arai, Inati, & Zaghloul, 2017; Zhang et al., 2015), or even ERS increases associated with overall reductions in power values (Zhang et al., 2015). Furthermore, it has been argued that overall power increases may bias multivariate analyses (Coutanche, 2013; Davis et al., 2014; de Beeck, 2010; Hebart & Baker, 2018). Thus, ERS effects in a given frequency range do not need to be associated with overall power increases in that frequency range.

Regarding the pathological imbalance, we did not intend to state that any change in the power ratio between lower and higher frequencies (putatively reflecting functional inhibition and excitation, respectively) is pathological, especially not if it occurs in response to a stimulus or task. To clarify this, we have changed the paragraph to be more specific and clearly state that the slope of a power spectrum may reflect both cognitive and pathological processes.

c. The attention literature on alpha and inhibition shows that entire cortical areas are inhibited if irrelevant for the task (e.g., ipsilateral hemisphere increases during lateralized attention tasks). If functional inhibition happens at this global level, why would inhibition be observed in LTC, an area that is highly relevant for word processing?

In our version of the directed forgetting paradigm, participants are instructed to forget words. As pointed out by the Reviewer, alpha-related inhibition effects occur in areas that are actively suppressed – e.g., visual sensory areas that are processing an irrelevant or distractor item. We would thus expect to find alpha effects in areas supporting word processing, in particular the LTC. Indeed, alpha increases during word processing have been reported in LTC (Becker, Pefkou, Michel, & Hervais-Adelman, 2013; Wilsch, Henry, Herrmann, Maess, & Obleser, 2015), especially when distracting (noise) information is added (Wilsch et al., 2015). Our data suggest that similar alpha-related inhibition processes do not only occur at the level of entire brain areas, but also in item-specific activity patterns. We would therefore argue that alpha effects can be more specific than typically measured with EEG. These more spatially selective effects are probably easier to detect with intracranial EEG.

Due to the relatively low spatial resolution of scalp EEG recordings, alpha effects may appear to occur across the majority of an entire hemisphere, although they typically peak in parietal-occipital regions (e.g., (Gould, Rushworth, & Nobre, 2011; Rihs, Michel, & Thut, 2007; Sauseng et al., 2005)). In addition, more regionally specific alpha effects have been found when either ventral or dorsal stream areas are selectively recruited (Jokisch & Jensen, 2007), or depending on which sensory modality is active (Haegens, Händel, & Jensen, 2011; Jiang, van Gerven, & Jensen, 2015). Some previous studies have identified even more spatially specific alpha effects: For example, Popov et al. (Popov, Gips, Kastner, & Jensen, 2019) showed that alpha is modulated in a retinotopically specific manner (also see (Popov, Gips, Weisz, & Jensen, 2021)), and alpha activity even differs across cortical layers (Haegens et al., 2015).

We only found effects of alpha/beta oscillations in multivariate and not in univariate analyses, suggesting a more fine-grained effect (de Beeck, 2010). Whether and in which conditions local and global effects can co-occur is still an open question. We have added this discussion to the manuscript.

In sum, to respond to concern #1 we made the following changes:

- We changed the title, abstract and introduction to be more neutral with respect to the interpretation of alpha/beta effects as inhibition. The title now reads: *“An engram of the forgotten: Intentionally forgotten information shows alpha/beta encoding-retrieval similarity”*.
- In the abstract, we state *“these results suggest that intentional forgetting relies more on inhibitory top-down connections than intentional remembering”*.
- In the introduction we state a neutral conclusion, namely: *“Our results demonstrate that successful intentional forgetting is due to a selective modification of item-specific top-down connections, rather than a mere degradation of the memory traces.”*
- We refrain from referring to “strengthening” of connections, but instead refer to “rearrangement” or “modification”.
- In the discussion we acknowledge that our inhibition interpretation of alpha/beta oscillations relies on reverse inference. It now reads: *“Alpha/beta frequency oscillations have been proposed as a measure of active inhibition (Jensen, Bonnefond, & VanRullen, 2012; Klimesch et al., 2007), restricting representations to narrow temporal “duty cycles” (Jensen, Gips, Bergmann, & Bonnefond, 2014) and generally lowering the amount of information that can be represented in a given brain region (Hanslmayr et al., 2012). This inhibitory process has been shown to occur at different spatial scales, from inhibition of a full sensory region (Haegens et al., 2011; Jensen et al., 2012) or hemisphere (Sauseng et al., 2005) to more local, e.g. retinotopic effects (Popov et al., 2019; Popov et al., 2021). Temporal cortex is no exception, as alpha/beta effects have also been found in temporal cortex as a response to increased processing demands during word processing (Becker et al., 2013; Wilsch et al., 2015). Based on this previous research, we suggest that the reactivated activity patterns of TBF items reflect functional inhibition. However, it should be emphasized that this interpretation relies on the logic of reverse inference, which has been criticized since its validity relies, among other factors, on the specificity of the observed neural measures (Axmacher et al., 2009; Poldrack, 2006). We would thus like to emphasize that the validity of our interpretation needs to be confirmed by future studies and in particular corroborated by formal analyses on the specificity of alpha/beta activity as a measure of functional inhibition.”*
- In the discussion we now discuss alternative explanations for our findings (see answer to next point). We also discuss that changes in excitation and inhibition are not necessarily pathological. The discussion now reads: *“Such an overall power change might have been an indication that the functional excitation-inhibition balance has changed (Gao et al., 2017). This commonly occurs during active cognitive tasks such as memory and attention (Jensen & Mazaheri, 2010b; Roux & Uhlhaas, 2014). More pronounced overall excitation-inhibition imbalances have also been associated with neuropathological mechanisms in various diseases (Dehghani et al., 2016; Markicevic et al., 2020)”*.

(2) The manuscript, throughout, uses the term “reinstatement” to refer to the encoding-retrieval similarity (ERS), even though these effects are observed very early after word presentation (<300msec). In the memory literature, reinstatement typically refers to the recapitulation of encoding-related information that is not directly conveyed by the sensory cue, and instead is the product of a hippocampal pattern completion process. Previous literature clearly shows that this process takes several hundred milliseconds to complete, such that neural patterns in the first 400-500msec reflect

sensory processing of the cue/recognition probe, and only start to reflect reinstated patterns after 500msec. An early effect as reported here is thus highly unlikely to reflect memory reinstatement. Different from what the results section suggests (l. 506-508), the new time-shifted analyses confirm that the ERS peaks extremely early, around 100msec after word onset (see maximum value in Fig. 4Cii, middle panel, TBF-forgotten items). Such an early ERS almost certainly reflects the processing of the retrieval cue, which in a recognition test fully overlaps with the encoding stimulus. To be clear, the results convincingly show that word processing is altered by a preceding forget instruction, and this is an interesting finding; the timing just does not support a reinstatement interpretation. As an alternative suggestion, the forget instruction might terminate the encoding process early, leading to reduced repetition priming of the stimulus? Such an interpretation could, for example, be corroborated by comparing similarity of TBF and TBR items to novel items purely within retrieval (rather than between encoding and retrieval), with the expectation that TBF items (and specifically TBF-forgotten ones) behave more like novel items.

Indeed, recognition memory tasks such as those employed in our paradigm (and in directed forgetting paradigms more generally) involve presentation of identical stimuli during encoding and retrieval. Therefore, stimulus representations do not need to be actively reconstructed from partial cues, and memory only consists in the successful recognition that an item has been presented during encoding. In the previous version of our manuscript, we referred to this process as “reinstatement” because this term has been widely used in different contexts and paradigms in the literature. Specifically, it has been used both to refer to cued recall (e.g. Yaffe et al., 2017) and to recognition memory (e.g. (Lohnas et al., 2018; Pacheco Estefan et al., 2019; Staudigl, Vollmar, Noachtar, & Hanslmayr, 2015; Wälti, Woolley, & Wenderoth, 2020)). Notably, recognition memory still requires participants to identify an item as one that has been shown before. Thus, the memory process brings back information from the encoding period that is not part of the percept during retrieval. Moreover, this information does not just reflect the entire encoding stage as a whole, but is item-specific, as shown in our ERS analysis (and various previous studies).

Nevertheless, we do acknowledge the important differences between recognition memory and cued recall and refrain from using the term “reinstatement” in the current version of the manuscript. We now refer to “reactivation” or to our specific operationalization, i.e. ERS. We also more clearly display the exact timing of ERS peaks for TBR and TBF items during encoding and retrieval (see Response Figure 2 below; this figure replaces the previous Figure 4C) and show that it peaks rather around 300-400 ms, not as early as 100 ms. Finally, we discuss that recognition memory and cued recall putatively differ in their time courses. The discussion now reads: *“Note however that encoding-retrieval similarity has been analyzed in both cued recall and recognition memory paradigms. During cued recall tests, participants see a cue and need to perform a pattern completion process in order to successfully retrieve a correct memory representation corresponding with the cue. During recognition memory this is not needed as the item itself is presented and the participant has to recognize whether they have seen it before. It is therefore possible that in recognition memory, reactivation is overall earlier because participants do not need to bring back sensory representations from memory as in cued recall. On the other hand, the timing of reported effects is highly variable and changes depending on the region of interest (e.g. hippocampus (Lohnas et al., 2018) vs temporal cortex (Pacheco Estefan et al., 2019)) as well depending on the exact study design. Notably, also during recognition memory tests, participants need*

to remember an item as one that has been shown before. The memory process thus brings back information from the encoding context that is not part of the percept during retrieval.”

Response Figure 2 (corresponding to Figure 4C). Time-shifted ERS analyses based on LTC alpha/beta power. White outline indicates significance (cluster-corrected for multiple comparisons). Insets at the bottom represent time windows at which effects are significant when ERS is based on the same encoding and retrieval time (peaking around 300-400 ms for the TBF forgotten condition).

We appreciate the alternative interpretation suggested by the Reviewer. We performed the suggested analyses and correlated activation patterns of novel items and of TBR or TBF items during retrieval. These analyses did not show any effects in the alpha/beta frequency range.

Response Figure 3. Correlations between novel items and TBR / TBF items during retrieval.

Response Figure 4. Memory-specific correlations with novel items during retrieval.

We are glad to hear that the reviewer agrees that our findings are interesting. We added a discussion of this alternative interpretation of the data: *“As described in the introduction, an alternative explanation for directed forgetting is that forgetting instructions terminate the encoding process. However, this cannot easily explain our finding of higher alpha/beta encoding-retrieval similarity for TBF vs. TBR words. Instead, one would expect an absence (or reduction) of ERS (Figure 1D).”*

(3) All three reviewers were somewhat surprised that the analyses focus on the ERS between the initial word encoding window and retrieval, rather than the initial cue window and retrieval. The cue window is where, in theory, cognitive control should start modifying the memory trace for a given word. The authors now added a set of results from the cue period, confirming that DLPFC exhibits patterns consistent with inhibitory control during the cue period, and a correlation of this activity with the ERS effect for TBF items. These analyses provide valuable additional evidence for the top-down influence of DLPFC during this period. Regarding the “inhibitory engram” interpretation, however, it seems more relevant to include the results of an ERS analysis that compares the word representations during the cue window at encoding (where the critical modification of an active memory trace presumably happens) with word representations at later retrieval (where the modified trace is then presumably reactivated).

We agree that during the cue period, a modification of the memory trace should happen and differentiate between TBR and TBF items. This does not necessarily imply, however, that the activity during the cue period reflects those aspects of the item-specific representations that are shared between encoding and retrieval. During the cue period, the word is not presented, which probably reduces item-specific similarities with activity during recognition memory. Indeed, when we analyzed item-specific cue-retrieval similarities for TBF items, we did not find any significant effects for either remembered or forgotten items (see below). Also, when we controlled for influences of cue-related alpha/beta power on ERS between word presentation and retrieval by means of partial correlation analyses, we still observed significant ERS effects for TBF items and for the subset of TBF forgotten items (see also response to reviewer 2).

Response Figure 5. ERS between cue period and retrieval period (window of 0.1-0.5 sec).

Response Figure 6. Temporal generalization analysis of ERS effects between cue and retrieval periods at the frequency of significant TBF simple effects.

More minor comments:

(4) The abstract and discussion (l. 387-389, l. 401-402) suggest that TBR items showed item-specific reinstatement, though this effect is no longer significant. These statements should be qualified.

We have removed this statement from the abstract.

(5) The statistics in the time-shifting analyses, shown in Fig. 4Ci and Cii are probably inflated, because the frequencies are pre-selected to show significant effects in the main analysis. The figure is still informative descriptively, in order to see where the ERS peaks in time.

We agree and note this in the text.

References

- Anderson, M. C., & Green, C. (2001). Suppressing unwanted memories by executive control. *Nature*, *410*(6826), 366-369.
- Anderson, M. C., & Hanslmayr, S. (2014). Neural mechanisms of motivated forgetting. *Trends in cognitive sciences*, *18*(6), 279-292.
- Axmacher, N., Elger, C. E., & Fell, J. (2009). The specific contribution of neuroimaging versus neurophysiological data to understanding cognition. *Behavioural brain research*, *200*(1), 1-6.
- Bailey, C. H., & Kandel, E. R. (1993). Structural changes accompanying memory storage. *Annual review of physiology*, *55*(1), 397-426.
- Bastos, A. M., Vezoli, J., Bosman, C. A., Schoffelen, J.-M., Oostenveld, R., Dowdall, J. R., . . . Fries, P. (2015). Visual areas exert feedforward and feedback influences through distinct frequency channels. *Neuron*, *85*(2), 390-401.
- Becker, R., Pefkou, M., Michel, C. M., & Hervais-Adelman, A. G. (2013). Left temporal alpha-band activity reflects single word intelligibility. *Frontiers in systems neuroscience*, *7*, 121.
- Coutanche, M. N. (2013). Distinguishing multi-voxel patterns and mean activation: why, how, and what does it tell us? *Cognitive, Affective, & Behavioral Neuroscience*, *13*(3), 667-673.
- Davis, T., LaRocque, K. F., Mumford, J. A., Norman, K. A., Wagner, A. D., & Poldrack, R. A. (2014). What do differences between multi-voxel and univariate analysis mean? How subject-, voxel-, and trial-level variance impact fMRI analysis. *NeuroImage*, *97*, 271-283.
- de Beeck, H. P. O. (2010). Against hyperacuity in brain reading: spatial smoothing does not hurt multivariate fMRI analyses? *NeuroImage*, *49*(3), 1943-1948.
- Dehghani, N., Peyrache, A., Telenczuk, B., Le Van Quyen, M., Halgren, E., Cash, S. S., . . . Destexhe, A. (2016). Dynamic balance of excitation and inhibition in human and monkey neocortex. *Scientific reports*, *6*(1), 1-12.
- Gao, R., Peterson, E. J., & Voytek, B. (2017). Inferring synaptic excitation/inhibition balance from field potentials. *NeuroImage*, *158*, 70-78.
- Gould, I. C., Rushworth, M. F., & Nobre, A. C. (2011). Indexing the graded allocation of visuospatial attention using anticipatory alpha oscillations. *Journal of neurophysiology*, *105*(3), 1318-1326.
- Haegens, S., Barczak, A., Musacchia, G., Lipton, M. L., Mehta, A. D., Lakatos, P., & Schroeder, C. E. (2015). Laminar profile and physiology of the α rhythm in primary visual, auditory, and somatosensory regions of neocortex. *Journal of Neuroscience*, *35*(42), 14341-14352.
- Haegens, S., Händel, B. F., & Jensen, O. (2011). Top-down controlled alpha band activity in somatosensory areas determines behavioral performance in a discrimination task. *Journal of Neuroscience*, *31*(14), 5197-5204.
- Hanslmayr, S., Volberg, G., Wimber, M., Oehler, N., Staudigl, T., Hartmann, T., . . . Bäuml, K.-H. T. (2012). Prefrontally driven downregulation of neural synchrony mediates goal-directed forgetting. *Journal of Neuroscience*, *32*(42), 14742-14751.
- Hebart, M. N., & Baker, C. I. (2018). Deconstructing multivariate decoding for the study of brain function. *NeuroImage*, *180*, 4-18.
- Jensen, O., Bonnefond, M., & VanRullen, R. (2012). An oscillatory mechanism for prioritizing salient unattended stimuli. *Trends in cognitive sciences*, *16*(4), 200-206.
- Jensen, O., Gips, B., Bergmann, T. O., & Bonnefond, M. (2014). Temporal coding organized by coupled alpha and gamma oscillations prioritize visual processing. *Trends in neurosciences*, *37*(7), 357-369.
- Jensen, O., & Mazaheri, A. (2010a). Shaping functional architecture by oscillatory alpha activity: gating by inhibition. *Frontiers in Human Neuroscience*, *4*.

- Jensen, O., & Mazaheri, A. (2010b). Shaping functional architecture by oscillatory alpha activity: gating by inhibition. *Frontiers in human neuroscience*, 4, 186.
- Jiang, H., van Gerven, M. A., & Jensen, O. (2015). Modality-specific alpha modulations facilitate long-term memory encoding in the presence of distracters. *Journal of cognitive neuroscience*, 27(3), 583-592.
- Jokisch, D., & Jensen, O. (2007). Modulation of gamma and alpha activity during a working memory task engaging the dorsal or ventral stream. *Journal of Neuroscience*, 27(12), 3244-3251.
- Klimesch, W. (1999). EEG alpha and theta oscillations reflect cognitive and memory performance: a review and analysis. *Brain Research Reviews*, 29(2), 169-195.
- Klimesch, W., Sauseng, P., & Hanslmayr, S. (2007). EEG alpha oscillations: the inhibition-timing hypothesis. *Brain research reviews*, 53(1), 63-88.
- Lamprecht, R., & LeDoux, J. (2004). Structural plasticity and memory. *Nature Reviews Neuroscience*, 5(1), 45-54.
- Lohnas, L. J., Duncan, K., Doyle, W. K., Thesen, T., Devinsky, O., & Davachi, L. (2018). Time-resolved neural reinstatement and pattern separation during memory decisions in human hippocampus. *Proceedings of the National Academy of Sciences*, 115(31), E7418-E7427.
- Markicevic, M., Fulcher, B. D., Lewis, C., Helmchen, F., Rudin, M., Zerbi, V., & Wenderoth, N. (2020). Cortical excitation: inhibition imbalance causes abnormal brain network dynamics as observed in neurodevelopmental disorders. *Cerebral cortex*, 30(9), 4922-4937.
- Martin, S. J., Grimwood, P. D., & Morris, R. G. (2000). Synaptic plasticity and memory: an evaluation of the hypothesis. *Annual review of neuroscience*, 23(1), 649-711.
- Pacheco Estefan, D., Sánchez-Fibla, M., Duff, A., Principe, A., Rocamora, R., Zhang, H., . . . Verschure, P. F. (2019). Coordinated representational reinstatement in the human hippocampus and lateral temporal cortex during episodic memory retrieval. *Nature communications*, 10(1), 1-13.
- Poldrack, R. A. (2006). Can cognitive processes be inferred from neuroimaging data? *Trends in cognitive sciences*, 10(2), 59-63.
- Popov, T., Gips, B., Kastner, S., & Jensen, O. (2019). Spatial specificity of alpha oscillations in the human visual system. *Human brain mapping*, 40(15), 4432-4440.
- Popov, T., Gips, B., Weisz, N., & Jensen, O. (2021). Brain areas associated with visual spatial attention display topographic organization during auditory spatial attention. *bioRxiv*.
- Rihs, T. A., Michel, C. M., & Thut, G. (2007). Mechanisms of selective inhibition in visual spatial attention are indexed by α -band EEG synchronization. *European Journal of Neuroscience*, 25(2), 603-610.
- Roux, F., & Uhlhaas, P. J. (2014). Working memory and neural oscillations: alpha-gamma versus theta-gamma codes for distinct WM information? *Trends in cognitive sciences*, 18(1), 16-25.
- Sauseng, P., Klimesch, W., Stadler, W., Schabus, M., Doppelmayr, M., Hanslmayr, S., . . . Birbaumer, N. (2005). A shift of visual spatial attention is selectively associated with human EEG alpha activity. *European Journal of Neuroscience*, 22(11), 2917-2926.
- Spitzer, B., & Haegens, S. (2017). Beyond the status quo: a role for beta oscillations in endogenous content (re) activation. *eneuro*, 4(4).
- Staudigl, T., Vollmar, C., Noachtar, S., & Hanslmayr, S. (2015). Temporal-pattern similarity analysis reveals the beneficial and detrimental effects of context reinstatement on human memory. *Journal of Neuroscience*, 35(13), 5373-5384.
- Wälti, M. J., Woolley, D. G., & Wenderoth, N. (2020). Assessing Rhythmic Visual Entrainment and Reinstatement of Brain Oscillations to Modulate Memory Performance. *Frontiers in Behavioral Neuroscience*, 14.
- Wilsch, A., Henry, M. J., Herrmann, B., Maess, B., & Obleser, J. (2015). Alpha oscillatory dynamics index temporal expectation benefits in working memory. *Cerebral cortex*, 25(7), 1938-1946.

- Yaffe, R. B., Shaikhouni, A., Arai, J., Inati, S. K., & Zaghoul, K. A. (2017). Cued memory retrieval exhibits reinstatement of high gamma power on a faster timescale in the left temporal lobe and prefrontal cortex. *Journal of Neuroscience*, 37(17), 4472-4480.
- Zhang, H., Fell, J., Staresina, B. P., Weber, B., Elger, C. E., & Axmacher, N. (2015). Gamma power reductions accompany stimulus-specific representations of dynamic events. *Current Biology*, 25(5), 635-640.

REVIEWER COMMENTS

Reviewer #1 (Remarks to the Author):

The authors adequately addressed my comments. I now recommend this article for publication.

Reviewer #2 (Remarks to the Author):

The authors have implemented a partial regression to help account for cue-related modifications that recur at the time of retrieval. I think this strengthens the findings overall.

Reviewer #3 (Remarks to the Author):

The authors present another thorough revision of their manuscript in response to the reviewer comments. Many of the points are now sufficiently addressed either by control analyses, or by including statements in the results and discussion sections highlighting where the findings need to be interpreted with caution. The change in wording from "strengthening of inhibitory connections" to "modification" also helps clarify the assumed mechanism underlying the observed effects.

The one aspect of the findings where I still have a major issue with is my previous point #2, concerning the timing of the observed "reactivation" effects. I had pointed out in this comment that an effect occurring as early as 50ms after presentation of the word is highly unlikely to reflect reinstatement of stored information recovered from the encoding period. The authors point out in their response that the alpha/beta effect peaks around 300ms (see their new Fig. 4Cii), consistent with a reactivation of encoding-related item information (e.g., contextual associations formed during encoding). They also changed the wording from "reinstatement" to "reactivation" throughout the manuscript, and they highlight that a recognition test still requires participants to recover encoding-related information in order to make an old/new judgment. I fully agree on the latter point, that old/new recognition requires the retrieval of encoding-related information not contained in the recognition probe itself. However, this recovery would then still depend on a hippocampal pattern completion process, such that the recognition probe needs to first be processed along the sensory-semantic word processing pathways, then eliciting the stored index in the hippocampus, and only then followed by the reactivation of the missing contextual information. Hence, such recognition-related context reactivation (no matter if it is called reinstatement or reactivation) would take a considerable amount of neural processing time, and it is very difficult to see how the brain would achieve such processing within 50ms after presentation of the recognition probe. The authors argue in their rebuttal that the effect peaks around 200-300ms (based on Fig. 4C), however, this refers to the encoding time, not the timing relative to the recognition probe. In the recognition time window (y-axis in this figure), it seems clear that the significant cluster starts right with the onset of the recognition probe, and peaks within 100ms. This timing, in my view, almost excludes an interpretation as "reinstatement" or "reactivation", which is at the core of the manuscript's narrative.

REVIEWER COMMENTS

Reviewer #3 (Remarks to the Author):

The authors present another thorough revision of their manuscript in response to the reviewer comments. Many of the points are now sufficiently addressed either by control analyses, or by including statements in the results and discussion sections highlighting where the findings need to be interpreted with caution. The change in wording from “strengthening of inhibitory connections” to “modification” also helps clarify the assumed mechanism underlying the observed effects.

The one aspect of the findings where I still have a major issue with is my previous point #2, concerning the timing of the observed “reactivation” effects. I had pointed out in this comment that an effect occurring as early as 50ms after presentation of the word is highly unlikely to reflect reinstatement of stored information recovered from the encoding period. The authors point out in their response that the alpha/beta effect peaks around 300ms (see their new Fig. 4Cii), consistent with a reactivation of encoding-related item information (e.g., contextual associations formed during encoding). They also changed the wording from “reinstatement” to “reactivation” throughout the manuscript, and they highlight that a recognition test still requires participants to recover encoding-related information in order to make an old/new judgment. I fully agree on the latter point, that old/new recognition requires the retrieval of encoding-related information not contained in the recognition probe itself. However, this recovery would then still depend on a hippocampal pattern completion process, such that the recognition probe needs to first be processed along the sensory-semantic word processing pathways, then eliciting the stored index in the hippocampus, and only then followed by the reactivation of the missing contextual information. Hence, such recognition-related context reactivation (no matter if it is called reinstatement or reactivation) would take a considerable amount of neural processing time, and it is very difficult to see how the brain would achieve such processing within 50ms after presentation of the recognition probe. The authors argue in their rebuttal that the effect peaks around 200-300ms (based on Fig. 4C), however, this refers to the encoding time, not the timing relative to the recognition probe. In the recognition time window (y-axis in this figure), it seems clear that the significant cluster starts right with the onset of the recognition probe, and peaks within 100ms. This timing, in my view, almost excludes an interpretation as “reinstatement” or “reactivation”, which is at the core of the manuscript’s narrative.

We thank all reviewers for their very constructive and valuable feedback throughout this thorough revision process.

The final remaining issue regards the cognitive interpretation of the ERS effects during retrieval given their short latency. As the reviewer points out, this effect starts within the first 100ms after stimulus presentation during retrieval and then continues until about 500ms (Figure 4Cii). As a side note, we would like to mention that interpretations of this effect in terms of cognitive processes rely on reverse inference based on the latencies observed in previous studies. Furthermore, the temporal resolution of our results is limited by the relatively slow dynamics of frequency-dependent power effects (as compared to spikes). Nevertheless, we agree that the early latency of the ERS effects needs to be considered more comprehensively and taken into account in the interpretation of the results.

We agree with the reviewer's description of the putative order of events during recognition memory: The stimulus is first processed along a hierarchy of sensory and semantic processing steps and is then transferred into the hippocampus, which, in turn, reactivates information about the encoding context. Thus, the observation that ERS effects of TBF items start very early during retrieval suggests that they do not (only) reflect reactivation of the encoding context, but rather correspond to the preceding sensory and semantic processing stages (even though later parts of the ERS effects may still reflect subsequent reactivation processes). Importantly, our results show that these ERS effects are functionally relevant for memory, since they are based on the contrast of forgotten vs. remembered TBF items.

We would like to note that our main conclusion – our main “narrative” – is not about the specific cognitive processes that occur during retrieval, since we can only infer them indirectly, but rather on the item-specific neural patterns that accompany the processing of stimuli during encoding and retrieval in the different experimental conditions. In other words, we aimed to measure the neural representations of individual words during encoding, and test which aspects of these representations – if any – reoccur during later retrieval of actively forgotten stimuli (as schematically depicted in Figure 1). Our data show that neural activity patterns that occurred during encoding reoccur in an item-specific manner, even for actively forgotten items. More specifically, memory traces of intentionally forgotten information have unique neural signatures in the LTC and in item-specific patterns of top-down connections from DLPFC to LTC.

In the previous versions of our manuscript, we aimed to express this reoccurrence of encoding-related neural activity patterns during retrieval by using the term “reinstatement” (in the original manuscript) or “reactivation” (in the revised manuscript). Both terms were only meant to refer to our finding that neural activity patterns that had occurred during encoding of individual items were also found during retrieval of these same items. In the previous revision of the manuscript, we already mentioned the putative cascade of cognitive processes during retrieval in the Discussion section, when we wrote *“During the cue, memory traces requires active retrieval of sensory representations. During encoding and retrieval, this is not needed as the word itself is presented. The ERS thus reflects the response to a sensory stimulus, combining both perceptual processes and memory-related item-specific information associated with the sensory representation.”* (lines 480-484 of current manuscript).

Thanks to the reviewer, we now understand that the terms “reinstatement” and “reactivation” may be misleading because they may be understood as implying an active memory process that brings back context information from the encoding section. We thoroughly revised the manuscript in order to make sure to remain neutral with regard to the specific cognitive processes that underlie the observed ERS effects. We thus consistently refer to the ERS effects that were actually measured, rather than to “reactivation” or “reinstatement”.

We also explicitly state that our results (or at least the early parts of the ERS effects during retrieval) likely reflect sensory-semantic processing stages of a given stimulus rather than reactivation of the encoding context. We took the freedom to use the described cascade of temporal processes provided by the reviewer to point out that *“More specifically, the recognition probe needs to first be processed along the sensory-semantic (word) processing pathways, elicit the stored index in the hippocampus, to then reactivate the missing contextual information.”* (lines 484-486) and *“During recognition memory, the recognition probe is first processed along the sensory-semantic (in our study, word) processing pathways*

after which missing contextual information is reactivated via memory processes in the hippocampus.” (lines 525-528)

We then continue with the proposed interpretation and write: *“The early ERS effects found in our study during retrieval likely reflect the initial sensory and semantic processing steps rather than an active memory process that reactivates information about the encoding context.”* (lines 528-530)

Similarly, we clearly state in the introduction that ERS effects could either occur due to sensory-semantic processing or due to recognition-memory related processes. The revised manuscript now reads: *“Indeed, several studies showed that episodic memory retrieval depends on the reoccurrence of item- and/or context-specific neural representations, a process that relies on close coordination between the hippocampus and the adjacent lateral temporal cortex⁵⁻⁹. This effect is typically more pronounced for remembered compared to forgotten information (^{10,11}; Figure 1A) and has been quantified via encoding-retrieval similarity (ERS). The increased ERS that was found for remembered events could come about due to processing along selectively modified sensory-semantic connections or due to activation of the memory context due to recognition-related processes. Note that while in some studies the terms reactivation or reinstatement refer specifically to recognition-related reactivation, we here are neutral about the process that causes the ERS effects (sensory-semantic processing or active recognition memory).”*

REVIEWERS' COMMENTS

Reviewer #3 (Remarks to the Author):

The authors fully addressed my one remaining concern regarding the interpretation of the early ERS effects as memory reinstatement. The language is now neutral throughout, and the possible interpretations of the ERS patterns are discussed with the necessary caution. All points are thus sufficiently addressed and I have no further comments.